# Review article: Comparison of local particle filters and new implementations

**Version 2.1, September 27, 2018**

Alban Farchi[1] and Marc Bocquet[1]

[1]CEREA, joint laboratory École des Ponts ParisTech and EDF R&D, Université Paris-Est, Champs-sur-Marne, France

*Correspondence to:* A. Farchi (alban.farchi@enpc.fr)

**Abstract.** Particle filtering is a generic weighted ensemble data assimilation method based on sequential importance sampling, suited for nonlinear and non-Gaussian filtering problems. Unless the number of ensemble members scales exponentially with the problem size, particle filter (PF) algorithms experience weight degeneracy. This phenomenon is a manifestation of the curse of dimensionality that prevents the use of PF methods for high-dimensional data assimilation. The use of local analyses to counteract the curse of dimensionality was suggested early in the development of PF algorithms. However, implementing localisation in the PF is a challenge because there is no simple and yet consistent way of gluing together locally updated particles across domains.

In this article, we review the ideas related to localisation and the PF in the geosciences. We introduce a generic and theoretical classification of local particle filter (LPF) algorithms, with an emphasis on the advantages and drawbacks of each category. Alongside with the classification, we suggest practical solutions to the difficulties of local particle filtering, that lead to new implementations and improvements in the design of LPF algorithms.

The LPF algorithms are systematically tested and compared using twin experiments with the one-dimensional Lorenz 40-variables model and with a two-dimensional barotropic vorticity model. The results illustrate the advantages of using the optimal transport theory to design the local analysis. With reasonable ensemble sizes, the best LPF algorithms yield data assimilation scores comparable to those of typical ensemble Kalman filter algorithms even for a mildly nonlinear system.

## 1 Introduction

The ensemble Kalman filter (EnKF, Evensen, 1994) and its variants are currently among the most popular data assimilation (DA) methods. Because EnKF-like methods are simple to implement, they have been successfully developed and applied to numerous dynamical systems in geophysics such as atmospheric and oceanographic models including in operational conditions (see for example Houtekamer et al., 2005; Sakov et al., 2012a).

The EnKF can be viewed as a subclass of sequential Monte Carlo (MC) methods whose analysis step relies on Gaussian distributions. However, observations can have non-Gaussian error distributions, an example being the case of bounded variables — which are frequent in ocean and land surface modeling or in atmospheric chemistry. Most geophysical dynamical models are nonlinear yielding non-Gaussian error distributions (Bocquet et al., 2010). Moreover, recent advances in numerical modeling enable the use of finer resolutions for the models: small scale processes that can increase nonlinearity have then to be resolved.

When the Gaussian assumption is not fulfilled, Kalman filtering is suboptimal. Iterative ensemble Kalman filter and smoother methods have been developed to overcome these limitations, mainly by including variational analysis in the algorithms (Zupanski, 2005; Sakov et al., 2012b; Bocquet and Sakov, 2014), or through heuristic iterations (Kalnay and Yang, 2010). Yet one cannot bypass the Gaussian representation of the conditional density with these latter methods. On the other hand, with particle filter (PF) methods (Gordon et al., 1993; Doucet et al., 2001; Arulampalam et al., 2002; Chen, 2003; van Leeuwen, 2009; Bocquet et al., 2010), all Gaussian and linear hypotheses have been relaxed, allowing a fully Bayesian analysis step. That is why the generic PF is a promising method.

Unfortunately, there is no successful application of it to a significantly high-dimensional DA problem. Unless the number of ensemble members scales exponentially with the problem size, PF methods experience weight degeneracy and yield poor estimates of the model state. This phenomenon is a symptom of the curse of dimensionality and is the main obstacle to an application of PF algorithms to most DA problems (Silverman, 1986; Kong et al., 1994; Snyder et al., 2008). Nevertheless, the PF has appealing properties – the method is elegant, simple and fast, and it allows for a Bayesian analysis. Part of the research on the PF is dedicated to their application to high-dimensional DA with a focus on four topics: importance sampling, resampling, hybridisation, and localisation.

Importance sampling is at the heart of PF methods where the goal is to construct a sample of the posterior density (the conditional density) given particles from the prior density using importance weights. The use of a proposal transition density is a way to reduce the variance of the importance weights, hence allowing the use of fewer particles. However, importance sampling with a proposal density can lead to more costly algorithms that are not necessarily rid of the curse of dimensionality (Chap. 4 of MacKay, 2003; Snyder et al., 2015). Proposal-density PF methods include the optimal importance particle filter (OIPF, Doucet et al., 2000), whose exact implementation is only available in simple DA problems (linear observation operator and additive Gaussian noise), the implicit particle filter (Chorin and Tu, 2009; Chorin et al., 2010; Morzfeld et al., 2012) which is an extension of the OIPF for DA problems using smoothing, the equivalent-weights particle filter (EWPF) and its implicit version (van Leeuwen, 2010; Zhu et al., 2016).

Resampling is the first improvement that was suggested in the bootstrap algorithm (Gordon et al., 1993) to avoid the collapse of a PF based on sequential importance sampling. Common resampling algorithms include the multinomial resampling and the stochastic universal (SU) sampling algorithms. The resampling step allows the algorithm to focus on particles that are more likely, but, as a drawback, it introduces sampling noise. Worse, it may lead to sample impoverishment hence failing to avoid the collapse of the PF if the model noise is insufficient (van Leeuwen, 2009; Bocquet et al., 2010). Therefore it is usual practice to add a regularisation step after the resampling (Musso et al., 2001). Using ideas from the optimal transport theory, Reich (2013)

designed a resampling algorithm that creates strong bindings between the prior ensemble members and the updated ensemble members.

Hybridising PFs with EnKFs seems a promising approach for the application of PF methods to high-dimensional DA, in which one can hope to take the best of both worlds: robustness of the EnKF and Bayesian analysis of the PF. The balance
between the EnKF and the PF analysis must be chosen carefully. Hybridisation especially suits the case where the number of significantly nonlinear degrees of freedom is small compared to the others. Hybrid filters have been applied for example to geophysical low-order models (Chustagulprom et al., 2016) and to Lagrangian DA (Apte and Jones, 2013; Slivinski et al., 2015).

In most geophysical systems, distant regions have (almost) independent evolution over short time scales. This idea was used
in the EnKF to implement localisation in the analysis (Houtekamer and Mitchell, 2001; Hamill et al., 2001; Evensen, 2003; Ott et al., 2004). In a PF context, localisation could be used to counteract the curse of dimensionality. Yet, if localisation of the EnKF is simple and leads to efficient algorithms (Hunt et al., 2007), implementing localisation in the PF is a challenge because there is no trivial way of gluing together locally updated particles across domains (van Leeuwen, 2009). The aim of this paper is to review and compare recent propositions of local particle filter (LPF) algorithms (Rebeschini and van Handel, 2015; Lee
and Majda, 2016; Penny and Miyoshi, 2016; Poterjoy, 2016; Robert and Künsch, 2017) and to suggest practical solutions to the difficulties of local particle filtering that lead to improvements in the design of LPF algorithms.

Section 2 provides some background on DA and particle filtering. Section 3 is dedicated to the curse of dimensionality with some theoretical elements and illustrations. The challenges of localisation in PF methods are then discussed in Sects. 4 and 7 from two different angles. For both approaches, we propose new implementations of LPF algorithms, which are tested in
Sects. 5, 6 and 8 with twin simulations of low-order models. Several of the LPFs are tested in Sect. 9 with twin simulations of a higher dimensional model. Conclusions are given in Sect. 10.

## 2   Background

### 2.1   The data assimilation filtering problem

We follow a state vector $\boldsymbol{x}_k \in \mathbb{R}^{N_x}$ at discrete times $t_k, k \in \mathbb{N}$, through independent observation vectors $\boldsymbol{y}_k \in \mathbb{R}^{N_y}$. The evo-
lution is assumed to be driven by a hidden Markov model whose initial distribution is $p(\boldsymbol{x}_0)$, whose transition distribution is $p(\boldsymbol{x}_{k+1}|\boldsymbol{x}_k)$, and whose observation distribution is $p(\boldsymbol{y}_k|\boldsymbol{x}_k)$.

The model can alternatively be described by

$$\boldsymbol{x}_{k+1} = \mathcal{M}_k(\boldsymbol{x}_k, \boldsymbol{w}_k), \tag{1}$$

$$\boldsymbol{y}_k = \mathcal{H}_k(\boldsymbol{x}_k, \boldsymbol{v}_k), \tag{2}$$

where the random vectors $\boldsymbol{w}_k$ and $\boldsymbol{v}_k$ follow the transition and observation distributions.

The components of the state vector $\boldsymbol{x}_k$ are called *state variables* or simply *variables* and the components of the observation vector $\boldsymbol{y}_k$ are called *observations*.

Let $\pi_{k|k}$ be the *analysis* (or *filtering*) density $\pi_{k|k} = p(\boldsymbol{x}_k | \boldsymbol{y}_{k:0})$, where $\boldsymbol{y}_{k:0}$ is the set $\{\boldsymbol{y}_l, \ l = 0 \dots k\}$ and let $\pi_{k+1|k}$ be the *prediction* (or *forecast*) density $\pi_{k+1|k} = p(\boldsymbol{x}_{k+1} | \boldsymbol{y}_{k:0})$ with $\pi_{0|-1}$ coinciding with $p(\boldsymbol{x}_0)$ by convention.

The prediction operator $P_k$ is defined by the Chapman–Kolmogorov equation:

$$P_k\left(\pi_{k|k}\right) \triangleq \pi_{k+1|k} = \int p(\boldsymbol{x}_{k+1} | \boldsymbol{x}_k) \pi_{k|k} \, \mathrm{d}\boldsymbol{x}_k, \tag{3}$$

and Bayes' theorem is used to define the correction operator $C_k$:

$$C_{k+1}\left(\pi_{k+1|k}\right) \triangleq \pi_{k+1|k+1} = \frac{p(\boldsymbol{y}_{k+1} | \boldsymbol{x}_{k+1}) \pi_{k+1|k}}{p(\boldsymbol{y}_{k+1} | \boldsymbol{y}_{k:0})}. \tag{4}$$

In this article, we consider the DA filtering problem that consists in estimating $\pi_{k|k}$ with given realisations of $\boldsymbol{y}_{k:0}$.

## 2.2 Particle filtering

The PF is a class of sequential MC methods that produces, from the realisations of $\boldsymbol{y}_{k:0}$, a set of weighted ensemble members (or *particles*) $\left(\boldsymbol{x}_k^i, w_k^i\right)$, $i = 1 \dots N_e$. The analysis density $\pi_{k|k}$ is estimated through the empirical density:

$$\pi_{k|k}^{N_e} = \sum_{i=1}^{N_e} w_k^i \, \delta_{\boldsymbol{x}_k^i}, \tag{5}$$

where the weights are normalised so that their sum is 1 and $\delta_{\boldsymbol{x}}$ is the Dirac distribution centered at $\boldsymbol{x}$.

Inserting the particle representation Eq. (5) in the Chapman–Kolmogorov equation yields

$$P_k\left(\pi_{k|k}^{N_e}\right) = \sum_{i=1}^{N_e} w_k^i \, p\left(\boldsymbol{x}_{k+1} | \boldsymbol{x}_k^i\right). \tag{6}$$

In order to recover a particle representation, the prediction operator $P_k$ must be followed by a sampling step $S^{N_e}$. In the bootstrap or sampling importance resampling (SIR) algorithm of Gordon et al. (1993), the sampling is performed as follows:

$$\boldsymbol{x}_{k+1}^i \sim p\left(\boldsymbol{x}_{k+1} | \boldsymbol{x}_k^i\right), \tag{7}$$

$$w_{k+1}^i \leftarrow w_k^i, \tag{8}$$

where $\boldsymbol{x} \sim p$ means that $\boldsymbol{x}$ is a realisation of a random vector distributed according to the probability density function (pdf) $p$. The empirical density $\pi_{k+1|k}^{N_e}$ is now an estimator of $\pi_{k+1|k}$.

Applying Bayes' theorem to $\pi_{k+1|k}^{N_e}$ gives a weight update that follows the principle of importance sampling:

$$w_{k+1}^i \leftarrow w_{k+1}^i \, p\left(\boldsymbol{y}_{k+1} | \boldsymbol{x}_{k+1}^i\right). \tag{9}$$

The weights are then renormalised so that they sum to 1.

Finally, an optional resampling step $R^{N_e}$ is added if needed (see Sect. 2.3). In terms of densities, the PF can be summarised by the recursion

$$\pi_{k+1|k+1}^{N_e} = R^{N_e} \circ C_{k+1} \circ S^{N_e} \circ P_k\left(\pi_{k|k}^{N_e}\right). \tag{10}$$

The additional sampling and resampling operators $S^{N_e}$ and $R^{N_e}$ are ensemble transformations that are required to propagate the particle representation of the density. Ideally, they should not alter the densities.

Under reasonable assumptions on the prediction and correction operators and on the sampling and resampling algorithms, it is possible to show that, in the limit $N_e \to \infty$, $\pi_{k|k}^{N_e}$ converges to $\pi_{k|k}$ for the weak topology on the set of probability measures over $\mathbb{R}^{N_x}$. This convergence result is one of the main reasons for the interest of the DA community in PF methods. More details about the convergence of PF algorithms can be found in Crisan and Doucet (2002).

Eventually, the focus of this article is on the analysis step, that is the correction and the resampling. Hence, *prior* or *forecast* (respectively *posterior*, *updated* or *analysis*) will refer to quantities before (respectively after) the analysis step.

## 2.3 Resampling

Without resampling, PF methods are subject to weight degeneracy: after a few assimilation cycles, one particle gets almost all the weight. The goal of resampling is to reduce the variance of the weights by reinitialising the ensemble. After this step, the ensemble is made of $N_e$ equally weighted particles.

In most resampling algorithms, highly probable particles are selected and duplicated while particles with low probability are discarded. It is desirable that the selection of particles has a low impact on the empirical density $\pi_{k|k}^{N_e}$. The most common resampling algorithms — multinomial resampling, SU sampling, residual resampling and Monte Carlo Metropolis–Hastings algorithm — are reviewed by van Leeuwen (2009). The multinomial resampling and the SU sampling algorithms, frequently mentionned in this paper, are described in Appendix E.

Resampling introduces sampling noise. On the other hand, not resampling means imparting computational time to highly improbable particles, that have a very low contribution to the empirical analysis density. Therefore, the choice of the resampling frequency is critical in the design of PF algorithms. Common criteria to decide if a resampling step is needed are based on measures of the degeneracy: for example the maximum of the weights or the effective ensemble size defined by Kong et al. (1994), i.e.

$$N_{\text{eff}} = \left( \sum_{i=1}^{N_e} \left( w_k^i \right)^2 \right)^{-1} \in [1, N_e]. \tag{11}$$

The correction and resampling steps of PF methods can be combined and embedded into the so-called *linear ensemble transform* (LET) framework (Bishop et al., 2001; Reich and Cotter, 2015) as follows. Let $\mathbf{E}_k$ be the *ensemble matrix*, that is the $N_x \times N_e$ matrix whose columns are the ensemble members $\boldsymbol{x}_k^i$. The update of the particles is then given by

$$\mathbf{E}_k \leftarrow \mathbf{E}_k \mathbf{T}, \tag{12}$$

where $\mathbf{T}$ is a $N_e \times N_e$ transformation matrix whose coefficients are uniquely determined during the resampling step. In the general LET framework, $\mathbf{T}$ has real coefficients and it is subject to the normalisation constraint

$$\sum_{i=1}^{N_e} [\mathbf{T}]^{i,j} = 1, \quad j = 1 \dots N_e, \tag{13}$$

such that the updated ensemble members can be interpreted as weighted averages of the prior ensemble members. The transformation is said to be first-order accurate if it preserves the ensemble mean (Acevedo et al., 2017), i.e. if

$$\sum_{j=1}^{N_{\mathrm{e}}} [\mathbf{T}]^{i,j} = N_{\mathrm{e}} w_k^i, \quad i = 1 \ldots N_{\mathrm{e}}. \tag{14}$$

In the "select and duplicate" resampling schemes, the coefficients of $\mathbf{T}$ are in $\{0, 1\}$ meaning that the updated particles are copies of the prior particles. The first-order condition Eq. (14) is then only satisfied on average over realisations of the resampling step. Yet it is sufficient to ensure the weak convergence of $\pi_{k|k}^{N_{\mathrm{e}}}$ almost surely in the case of the multinomial resampling (Crisan and Doucet, 2002).

If the coefficients of $\mathbf{T}$ are positive reals, the transformation can be understood as a resampling where the updated particles are composite copies of the prior particles. For example, in the ensemble transform particle filter (ETPF) algorithm of Reich (2013), the transformation is chosen such that it minimises the expected distance between the prior and the updated ensembles (seen as realisations of random vectors) among all possible first-order accurate transformations. This leads to a minimisation problem typical of the discrete optimal transport theory (Villani, 2009):

$$\min_{\mathbf{T} \in \mathcal{T}} \sum_{i,j=1}^{N_{\mathrm{e}}} [\mathbf{T}]^{i,j} \left\| \boldsymbol{x}_k^i - \boldsymbol{x}_k^j \right\|^2, \tag{15}$$

where $\mathcal{T}$ is the set of $N_{\mathrm{e}} \times N_{\mathrm{e}}$ transformation matrices satisfying Eqs. (13) and (14). In this way, the correlation between the prior and the updated ensembles is increased and $\pi_{k|k}^{N_{\mathrm{e}}}$ still converges toward $\pi_{k|k}$ for the weak topology. In the following, this resampling algorithm will be called *optimal ensemble coupling*.

## 2.4 Proposal-density particle filters

Let $q(\boldsymbol{x}_{k+1})$ be a density whose support is larger than that of $p(\boldsymbol{x}_{k+1}|\boldsymbol{x}_k)$ — i.e. $q(\boldsymbol{x}_{k+1}) > 0$ whenever $p(\boldsymbol{x}_{k+1}|\boldsymbol{x}_k) > 0$. The Chapman–Kolmogorov Eq. (3) can be written:

$$\pi_{k+1|k} = \int \frac{p(\boldsymbol{x}_{k+1}|\boldsymbol{x}_k)}{q(\boldsymbol{x}_{k+1})} q(\boldsymbol{x}_{k+1}) \pi_{k|k} \, \mathrm{d}\boldsymbol{x}_k. \tag{16}$$

In the importance sampling literature, $q$ is called the *proposal* density and can be used to perform the sampling step $S^{N_{\mathrm{e}}}$ described by Eqs. (7) and (8) in a more general way:

$$\boldsymbol{x}_{k+1}^i \sim q(\boldsymbol{x}_{k+1}), \tag{17}$$

$$w_{k+1}^i \leftarrow w_k^i \frac{p(\boldsymbol{x}_{k+1}^i|\boldsymbol{x}_k^i)}{q(\boldsymbol{x}_{k+1}^i)}. \tag{18}$$

Using the proposal density $q$ can lead to an improvement of the PF method if for example $q$ is easier to sample from than $p$ or if $q$ includes information about $\boldsymbol{x}_k$ or $\boldsymbol{y}_{k+1}$ in order to reduce the variance of the importance weights.

The SIR algorithm is recovered with the *standard* proposal $p(\boldsymbol{x}_{k+1}|\boldsymbol{x}_k)$, while the *optimal importance* proposal $p(\boldsymbol{x}_{k+1}|\boldsymbol{x}_k, \boldsymbol{y}_{k+1})$ yields the optimal importance sampling importance resampling (OISIR) algorithm (Doucet et al., 2000). Merging the prediction

and correction steps of the OISIR algorithm yields the weight update

$$w_{k+1}^i \leftarrow w_k^i \, p\left(\boldsymbol{y}_{k+1} | \boldsymbol{x}_k^i\right). \tag{19}$$

It is remarkable that this formula does not depend on $\boldsymbol{x}_{k+1}$ (Doucet et al., 2000). Hence the optimal importance proposal is optimal in the sense that it minimises the variance of the weights over realisations of $\boldsymbol{x}_{k+1}^i$ — namely 0. Moreover, it can be shown that it also minimises the variance of the weights over realisations of the whole trajectory $\boldsymbol{x}_{k+1:0}^i$ among proposal densities that depend on $\boldsymbol{x}_k$ and $\boldsymbol{y}_{k+1}$ (Snyder et al., 2015).

Although the optimal importance proposal has appealing properties, its computation is non-trivial. For the generic model with Gaussian additive noise described in Appendix A2, when the observation operator $\mathcal{H}$ is linear, the optimal importance proposal can be computed as a Kalman filter analysis as shown by Doucet et al. (2000). However, in the general case there is no analytic form and one must resort to more elaborate algorithms (Chorin and Tu, 2009; Chorin et al., 2010; Morzfeld et al., 2012).

## 3   The curse of dimensionality

### 3.1   The weight degeneracy of particle filters

The PF has been successfully applied to low-dimensional DA problems (Doucet et al., 2000). However, attempts to apply the SIR algorithm to medium- to high-dimensional geophysical models have led to weight degeneracy (e.g., van Leeuwen, 2003; Zhou et al., 2006).

Bocquet et al. (2010) demonstrated weight degeneracy in low-order models, for example in the Lorenz 1996 (L96, Lorenz and Emanuel, 1998) model in the standard configuration described in Appendix A3. They illustrated the empirical statistics of the maximum of the weights for several values of the system size. When the system size is small (10 to 20 variables) weights are balanced and values close to 1 are infrequent. However, when the system size grows (more than 40 variables) weights rapidly degenerate: values close to 1 become more frequent. Ultimately, the frequency of the maximum of the weights peaks to 1.

Similar results are produced when applying one importance sampling step to the Gaussian linear model described in Appendix A1. For this model, we illustrate the empirical statistics of the maximum of the weights in Fig. 1. Snyder et al. (2008) also computed the required number of particles in order to avoid degeneracy in simulations and found that it scales exponentially with the size of the problem.

This phenomenon, well known in the PF literature, is often referred to as *degeneracy*, *collapse* or *impoverishment* and is a symptom of the curse of dimensionality.

### 3.2   The equivalent state dimension

At first sight, it might seem surprising that, although MC methods have a convergence rate independent of the dimension, the curse of dimensionality applies to PF methods. Yet, the correction step $C_k$ is an importance sampling step between the prior

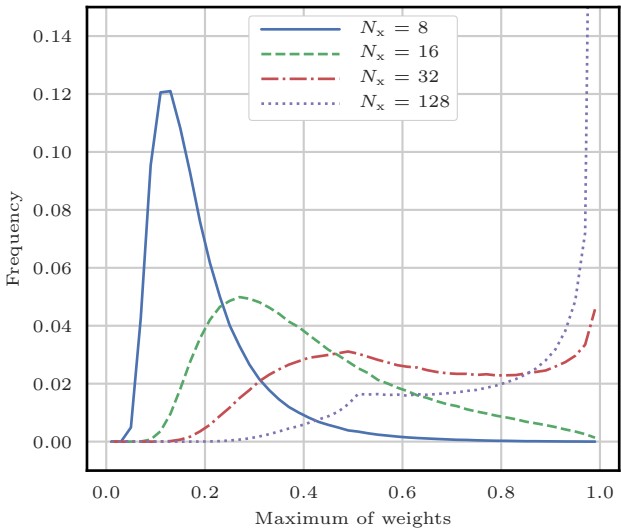

**Figure 1.** Empirical statistics of the maximum of the weights for one importance sampling step applied to the Gaussian linear model of Appendix A1. The model parameters are $p = 1$, $a = 1$, $h = 1$, $q = 1$, $\sigma = 1$, the ensemble size is $N_e = 128$ and the system size varies from $N_x = 8$ (well balanced case) to $N_x = 128$ (almost degenerate case).

and the analysis probability densities. The higher the number of observations $N_y$, the more singular these densities are to each other: random particles from the prior density have an exponentially small likelihood according to the analysis density. This is the main reason for the blow up of the number of particles required for a non-degenerate scenario (Rebeschini and van Handel, 2015).

A quantitative description of the behavior of weights for large values of $N_y$ can be found in Snyder et al. (2008). In this study, the authors first define:

$$\tau^2 = \mathrm{var}\left[\ln\left(p(\boldsymbol{y}_k|\boldsymbol{x}_k)\right)\right], \tag{20}$$

with the hypothesis that the observation noise is additive and each of its component is independent and identically distributed (iid). Then they derive the asymptotic relationship for only one analysis step:

$$\mathbb{E}\left[\frac{1}{\max_i w_k^i}\right] \underset{N_e \to \infty}{\sim} 1 + \frac{\sqrt{2\ln N_e}}{\tau}, \tag{21}$$

where $\mathbb{E}$ is the expectation over realisations of the prior ensemble members.

This result means that, in order to avoid the collapse of a PF method, the number of particles $N_e$ must be of order $\exp\left(\tau^2/2\right)$. In simple cases, as the ones considered in the previous sections, $\tau^2$ is proportional to $N_y$. The dependence of $\tau$ on $N_x$ is indirect in the sense that the derivation of Eq. (21) requires $N_x$ to be asymptotically large. In a sense, one can think of $\tau^2$ as an *equivalent state dimension*.

Snyder et al. (2008) then illustrate the validity of the asymptotic relationship Eq. (21) using simulations of the Gaussian linear model of Appendix A1 with a SIR algorithm, for which:

$$\tau^2 = N_{\text{y}} \frac{h^2 \left(q^2 + a^2 p^2\right)}{\sigma^2} \left(1 + \frac{3h^2}{2\sigma^2} \left(q^2 + a^2 p^2\right)\right). \tag{22}$$

Snyder et al. (2008) do not illustrate the validity of Eq. (21) in more general cases, mainly because the computation of $\tau$ is non-trivial. The effect of resampling is not investigated either, thought it is clear from simulations that resampling is not enough to avoid filter collapse. Finally, the effect of using proposal densities is the subject of another study by Snyder et al. (2015).

## 3.3 Mitigating the collapse using proposals

One objective of using proposal densities in PF methods is to reduce the variance of the importance weights as discussed in Sect. 2.4. If one uses the optimal importance proposal density $p\left(\boldsymbol{x}_{k+1} | \boldsymbol{x}_k, \boldsymbol{y}_{k+1}\right)$ to sample $\boldsymbol{x}_k$ in the prediction and sampling step $S^{N_{\text{e}}} \circ P_k$, the correction step $C_{k+1}$ consists in matching two identical densities, which leads to a weight update Eq. (19) that does not depend on the realisation of $\boldsymbol{x}_{k+1}$.

Yet, the OISIR algorithm still collapses even for low-order models such as the L96 model with 40 variables (Bocquet et al., 2010). In fact, the curse of dimensionality for any proposal-density PF does not primarily come from the correction step $C_k$, but from the recursion in the PF. In particular it stems from the fact that the algorithm does not correct the particles at earlier times to account for new observations (Snyder et al., 2015). This was a key motivation in the development of the guided SIR algorithm of van Leeuwen (2009), whose ideas were included in the practical implementations of the EWPF algorithm (van Leeuwen, 2010; Ades and van Leeuwen, 2015) as a relaxation step, with moderate success (Browne, 2016).

Snyder et al. (2015) illustrate the validity of Eq. (21) using simulations of the Gaussian linear model of Appendix A1 with an OISIR algorithm, for which:

$$\tau^2 = N_{\text{y}} \frac{a^2 p^2 h^2}{\sigma^2 + h^2 q^2} \left(1 + \frac{3a^2 h^2 p^2}{2\left(\sigma^2 + h^2 q^2\right)}\right), \tag{23}$$

and they found a good accuracy of Eq. (21) in the limit $N_{\text{e}} \ll \exp\left(\tau^2/2\right)$. This shows that the use of the optimal importance proposal reduces the number of particles required to avoid the collapse of a PF method. However, ultimately, proposal-density PFs cannot counteract the curse of dimensionality in this simple model and there is no reason to think that they could in more elaborate models (see chapter 29 of MacKay, 2003).

In a generic Gaussian linear model, the equivalent state dimension $\tau^2$ as in Eqs. (22) and (23) is directly proportional to the system size $N_{\text{x}}$ — equal to $N_{\text{y}}$ in this case. For more elaborate models, the relationship between $\tau^2$ and $N_{\text{x}}$ is likely to be more complex and may involve the effective number of degrees of freedom in the model.

## 3.4 Using localisation to avoid collapse

By considering the definition of $\tau^2$, Eq. (20), one can see that the curse of dimensionality is a consequence of the fact that the importance weights are influenced by all components of the observation vector $\boldsymbol{y}_k$. Yet, a particular state variable and

observation can be nearly independent, for example in spatially extended models if they are distant to each other. In this situation, the statistical properties of the ensemble at this state variable (i.e. the marginal density) should not evolve during the analysis step. Yet, this is not the case in PF methods, because of the use of (relatively) low ensemble sizes; even the ensemble mean can be significantly impacted. A good illustration of this phenomenon can be found in Fig. 2 of Poterjoy (2016). In this case, the PF overestimates the information available and equivalently underestimates the uncertainty in the analysis density (Snyder et al., 2008). As a consequence, spurious correlations appear between distant state variables.

This would not be the case in a PF algorithm that would be able to perform local analyses, that is when the influence of each observation is restricted to a spatial neighborhood of its location. The equivalent state dimension $\tau^2$ would then be defined using the maximum number of observations that influence a state variable, which could be kept relatively small even for high-dimensional systems.

In the EnKF literature, this idea is known as *domain localisation* or *local analysis* and was introduced to fix the same kind of issues (Houtekamer and Mitchell, 2001; Hamill et al., 2001; Evensen, 2003; Ott et al., 2004). Technical implementations of domain localisation in EnKF methods is as easy as implementing a global analysis and the local analyses can be carried out in parallel (Hunt et al., 2007). By contrast, the application of localisation techniques in PF methods is discussed in Snyder et al. (2008); van Leeuwen (2009); Bocquet et al. (2010) with an emphasis on two major difficulties.

The first issue is that the variation of the weights across local domains irredeemably breaks the structure of the global particles. There is no trivial way of recovering this global structure, i.e. gluing together the locally updated particles. Global particles are required for the prediction and sampling step $S^{N_e} \circ P_k$ in all PF algorithms, where the model $\mathcal{M}_k$ is applied to each individual ensemble member.

Second, if not carefully constructed, this gluing together could lead to balance problems and sharp gradients in the fields (van Leeuwen, 2009). In EnKF methods, these issues are mitigated by using smooth functions to taper the influence of the observations. The smooth dependency of the analysis ensemble on the observation precision reduces imbalance (Greybush et al., 2011). Yet, in most PF algorithms, there is no such smooth dependency. From now on, this issue will be called "imbalance" or "discontinuity" issue. The word "discontinuity" does not point to the discrete nature of the model field on the grid, but, inspired by the mathematical notion of continuity, to large unphysical gaps appearing in the discrete model field.

### 3.5 Two types of localisation

From now on, we will assume that our DA problem has a well-defined spatial structure:

- each state variable is attached to a location, the *grid point*;

- each observation is attached to a location, the *observation site* or simply the *site* (observations are assumed local);

- there is a distance function between locations.

The goal is to be able to define notions such as "the distance between an observation site and a grid point", "the distance between two grid points" or "the center of a group of grid points". In realistic models, these concepts need to be related to the underlying physical space.

In the following sections, we discuss algorithms that address the two issues of local particle filtering (gluing and imbalance) and lead to implementations of domain localisation in PF methods. We divide the solutions into two categories.

In the first approach, independent analyses are performed at each grid point by using only the observation sites that influence this grid point. This leads to algorithms that are easy to define, to implement and to parallelise. However, there is no obvious relationship between state variables, which could be problematic with respect to the imbalance issue. This approach is used for example by Rebeschini and van Handel (2015); Penny and Miyoshi (2016); Lee and Majda (2016); Chustagulprom et al. (2016). In this article, we call it *state–domain* (and later *state–block–domain*) localisation.

In the second approach, an analysis is performed at each observation site. When assimilating the observation at a site, we partition the state space: nearby grid points are updated while distant grid point remain unchanged. In this formalism, observations need to be assimilated sequentially, which makes the algorithms harder to define and to parallelise but may mitigate the imbalance issue. This approach is used for example by Poterjoy (2016). In this article, we call it *sequential–observation* localisation.

## 4   State–domain localisation for particle filters

From now on, the time subscript $k$ is systematically dropped for clarity and the conditioning with respect to prior quantities is implicit. The superscript $i \in \{1 \dots N_e\}$ is the member index, the subscript $n \in \{1 \dots N_x\}$ is the state variable or grid point index, the subscript $q \in \{1 \dots N_y\}$ is the observation or observation site index, the subscript $b \in \{1 \dots N_b\}$ is the block index (the concept of block is defined in Sect. 4.2).

### 4.1   Introducing localisation in particle filters

Localisation is generally introduced in PF methods by allowing the analysis weights to depend on the spatial position. In the (global) PF, the marginal of the analysis density for each state variable is

$$p(x_n) = \sum_{i=1}^{N_e} w^i \, \delta_{x_n^i}, \tag{24}$$

whose localised version is

$$p(x_n) = \sum_{i=1}^{N_e} w_n^i \, \delta_{x_n^i}. \tag{25}$$

The local weights $w_n^i$ depend on the spatial position through the grid point index $n$.

With local analysis weights, the marginals of the analysis density are uncoupled. This is the reason why localisation was introduced in the first place, but, as a drawback, the full analysis density is not known. The simplest fix is to approximate the full density as the product of its marginals:

$$p(\boldsymbol{x}) = \prod_{n=1}^{N_x} \sum_{i=1}^{N_e} w_n^i \, \delta_{x_n^i}, \tag{26}$$

which is a weighted sum of the $N_e^{N_x}$ possible combinations between all particles.

In summary, in LPF methods, we keep the generic MC structure described in Sect. 2.2. The prediction and sampling step is not modified. The correction step is adjusted to allow the analysis density to have the form given by Eq. (26). In particular, one has to define the local analysis weights $w_n^i$; this point will be discussed in Sect. 4.2.2. Finally, global particles are required for the next assimilation cycle and they are obtained as follows. A local resampling is first performed *independently for each grid*
*point*. The locally resampled particles are then assembled into global particles. The local resampling step is discussed in detail in Sect. 4.4.

## 4.2    Extension to state–block–domain localisation

The principle of localisation in the PF, and in particular Eq. (26), can be included into a more general state–block–domain (SBD) localisation formalism. The state space is divided into (local state) *blocks* with the additional constraint that the weights
should be constant over the blocks. The resampling then has to be performed *independently for each block*.

In the block particle filter algorithm of Rebeschini and van Handel (2015), the local weight of a block is computed using the observation sites that are located inside this block. However, in general nothing prevents one from using the observation sites inside a *local domain* potentially different from the block. This is the case in the LPF of Penny and Miyoshi (2016), in which the blocks have size 1 grid point while the size of the local domains is controlled by a localisation radius.
To summarise, LPF algorithms using the SBD localisation formalism, hereafter called LPF[x] algorithms[1], are characterised by

- the geometry of the blocks over which the weights are constant;

- the local domain of each block, which gathers all observation sites used to compute the local weight;

- the local resampling algorithm.

Most LPFs (e.g. those described in Rebeschini and van Handel, 2015; Penny and Miyoshi, 2016; Lee and Majda, 2016) in the literature can be seen to adopt this SBD formalism.

### 4.2.1    The local state blocks

Using parallelepipedic blocks is a standard geometric choice (Rebeschini and van Handel, 2015; Penny and Miyoshi, 2016). It is easy to conceive and to implement and it offers a potentially interesting degree of freedom: the *block shape*. Using larger
blocks decreases the proportion of block boundaries and hence the bias in the local analyses. On the other hand, it also means less freedom to counteract the curse of dimensionality.

In the clustered particle filter algorithms of Lee and Majda (2016), the blocks are centered around the observation sites. The potential gains of this method are unclear. Moreover, when the sites are regularly distributed over the space — which is the case in the numerical examples of Sects. 5 and 6 — there is no difference with the standard method.

---

[1]The x exponent emphasises the fact that we perform one analysis per (local state) block.

## 4.2.2 The local domains

The general idea of domain localisation in the EnKF is that the analysis at one grid point is computed using only the observation sites that lie within a local region around this grid point, hereafter called the *local domain*. For instance in two dimensions a common choice is to define the local domain of a grid point as a disk, centered at this grid point, and whose radius is a free parameter called the *localisation radius*. The same principle can be applied to the SBD localisation formalism: the local domain of a block will be a disk whose center coincides with that of the block and whose radius will be a free parameter.

The terminology adopted here (disk, radius...) fits two-dimensional spatial spaces. Yet, most geophysical models have a three-dimensional spatial structure, with typical uneven vertical scales usually much shorter than horizontal scales. For these models, the geometry of the local domains should be adapted accordingly.

Increasing the localisation radius allows one to take more observation sites into account hence reducing the bias in the local analysis. It is also a means to reduce the spatial inhomogeneity by making the weights smoother in space.

The smoothness of the local weights is an important property. Indeed, spatial discontinuities in the weights can lead to spatial discontinuities in the updated particles. Again lifting ideas from the local EnKF methods, the smoothness of the weights can be improved by tapering the influence of an observation site with respect to its distance to the block center as follows. For the (global) PF, assuming that the observation sites are independent, the unnormalised weights are computed according to

$$w^i = \prod_{q=1}^{N_y} p\left(y_q | \boldsymbol{x}^i\right). \tag{27}$$

Following Poterjoy (2016), it becomes for an LPF:

$$w_b^i = \prod_{q=1}^{N_y} \left\{ \alpha + G\left(\frac{d_{q,b}}{r}\right) \left(p\left(y_q | \boldsymbol{x}^i\right) - \alpha\right) \right\}, \tag{28}$$

where $\alpha$ is a constant that should be of the same order as the maximum value of $p\left(\boldsymbol{y} | \boldsymbol{x}\right)$, $d_{q,b}$ is the distance between the $q$-th observation site and the center of the $b$-th block, $r$ is the localisation radius and $G$ is the taper function: $G(0) = 1$ and $G(x) = 0$ if $x$ is larger than 1, with a smooth transition. A popular choice for $G$ is the piecewise rational function of Gaspari and Cohn (1999), hereafter called the Gaspari–Cohn function. If the observation error is an iid Gaussian additive noise with variance $\sigma^2$, one can use an alternative "Gaussian" formula for $w_b^i$ directly inspired from local EnKF methods:

$$\ln w_b^i = -\frac{1}{2\sigma^2} \sum_{q=1}^{N_y} G\left(\frac{d_{q,b}}{r}\right) \left(y_q - \mathcal{H}_q\left(\boldsymbol{x}^i\right)\right)^2. \tag{29}$$

Equations (28) and (29) differ. Still they are equivalent in the asymptotic limit $r \to 0$ and $\sigma \to \infty$.

## 4.2.3 Algorithm summary

Algorithm 1 describes the analysis step for a generic LPF[x]. The algorithm parameters are: the ensemble size $N_e$, the geometry of the blocks and the localisation radius $r$ used to compute the local weights with Eq. (28) or (29). $N_b$ is the number of blocks

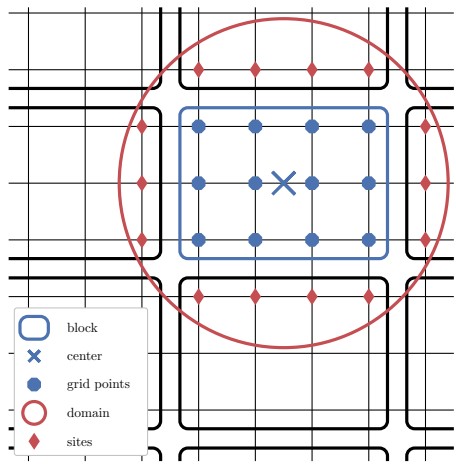

**Figure 2.** Example of geometry in the SBD localisation formalism for a two-dimensional space. The focus is on the block in the middle which gathers 12 grid points. The local domain is circumscribed by a circle around the block center with potential observation sites outside the block.

and $\mathbf{E}_{|b}$ is the restriction of the ensemble matrix $\mathbf{E}$ to the $b$-th block (i.e. the rows of $\mathbf{E}$ corresponding to grid points that are located within the $b$-th block). $\mathbf{E}_{|b}$ is a $N_\mathrm{x}/N_\mathrm{b} \times N_\mathrm{e}$ matrix.

In this algorithm, and in the rest of this article, the ensemble matrix $\mathbf{E}$ and the particles $\boldsymbol{x}^i$ (its columns) are used interchangeably. Note that in most cases, steps 3, 5 and 6 can be merged into one step.

An illustration of the definition of blocks and local domains is displayed in Fig. 2.

---

**Algorithm 1** Analysis step for a generic LPF$^\mathrm{x}$

---

**Require:** Prior (forecast) ensemble $\boldsymbol{x}^i$, $i = 1 \ldots N_\mathrm{e}$

1:  **for** $b = 1$ **to** $N_\mathrm{b}$ **do**
2:       Compute the local weights $w_b^i$ using Eq. (28) or (29)
3:       Resample the local ensemble $\mathbf{E}_{|b}$ with weights $w_b^i$ as $\mathbf{E}_{|b}^\mathrm{r}$
4:  **end for**
5:  Assemble the locally resampled ensembles $\mathbf{E}_{|b}^\mathrm{r}$ as $\mathbf{E}^\mathrm{r}$
6:  Update the ensemble: $\mathbf{E} \leftarrow \mathbf{E}^\mathrm{r}$
7:  **return** Updated (analysis) ensemble $\boldsymbol{x}^i$, $i = 1 \ldots N_\mathrm{e}$.

---

### 4.3 Beating the curse of dimensionality

The feasibility of PF methods using SBD localisation is discussed by Rebeschini and van Handel (2015) through the example

of their block particle filter algorithm. In this algorithm, the distinction between blocks and local domains does not exist. The

influence of each observation is not tapered and the resampling is performed independently for each block, regardless of the boundaries between blocks.

The main mathematical result is that, under reasonable hypotheses, the error on the analysis density for this algorithm can be bounded by the sum of a bias and a variance term. The bias term is related to the block boundaries and decreases exponentially with the diameter of the blocks, in number of grid points. It is due to the fact that the correction is not Bayesian any more since only a subset of observations is used to update each block. The exponential decrease is a demonstration of the *decay of correlations* property. The variance term is common to all MC methods and scales with $\exp(K)/\sqrt{N_e}$. For global MC methods, $K$ is the state dimension, whereas here $K$ is the number of grid points inside each block. This implies that LPF$^x$ algorithms can indeed beat the curse of dimensionality with reasonably large ensembles.

## 4.4   The local resampling

Resampling from the analysis density given by Eq. (26) does not cause any theoretical or technical issue. One just needs to apply any resampling algorithm (e.g. those described in Sect. 2.3) locally to each block using the local weights. Global particles are then obtained by assembling the locally resampled particles. By doing so, adjacent blocks are fully uncoupled — this is the same remark as when we constructed the analysis density Eq. (26) from its marginals Eq. (25). Once again, this is beneficial, since uncoupling is what counteracts the curse of dimensionality.

On the other hand, blind assembling is likely to lead to unphysical discontinuities in the updated particles, regardless of the spatial smoothness of the analysis weights. More precisely, one builds *composite particles*, that is when the $i$-th updated particle is composed of the $j$-th particle on one block and of the $k$-th particle on an adjacent block with $j \neq k$ — as shown by Fig. 3 in one dimension. There is no guarantee that the $j$-th and the $k$-th local particles are *close* and that assembling them will represent a physical state.

In order to mitigate the unphysical discontinuities, the analysis weights must be spatially smooth, as mentioned in Sect. 4.2.2. Moreover, the resampling scheme must have some "regularity", in order to preserve part of the spatial structure held in the prior particles. This is a challenge due to the stochastic nature of the resampling algorithms; potential solutions are presented hereafter.

### 4.4.1   Applying a smoothing by weights

A first solution is to smooth out potential unphysical discontinuities by averaging in space the locally resampled ensemble as follows. This method was introduced by Penny and Miyoshi (2016) in their LPF and called *smoothing by weights*.

Let $\mathbf{E}_b^r$ be the matrix of the ensemble computed by applying the resampling method to the global ensemble, weighted by the local weights $w_b^i$ of the $b$-th block. $\mathbf{E}_b^r$ is an $N_x \times N_e$ matrix different from the $N_x/N_b \times N_e$ matrix $\mathbf{E}_{|b}^r$ defined in Sect. 4.2.3.

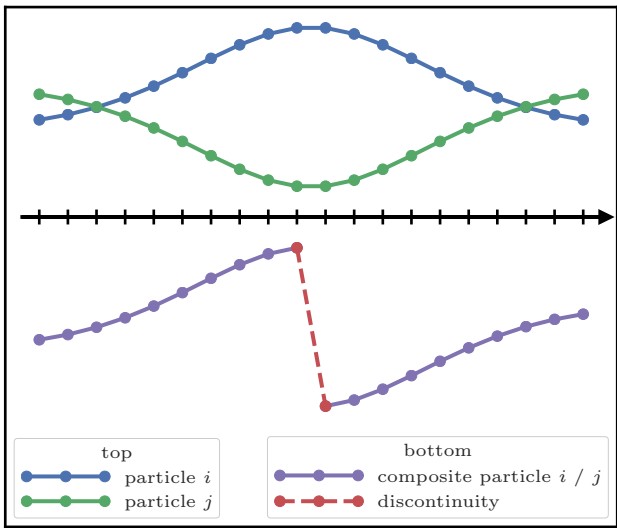

**Figure 3.** Example of one-dimensional concatenation of particle $i$ on the left and particle $j$ on the right. Top: the prior particles $i$ and $j$. Bottom: the composite particle, concatenation of $i$ and $j$. In this situation, a large unphysical discontinuity appears at the boundary.

We then define the smoothed ensemble matrix $\mathbf{E}^{\mathrm{s}}$ by

$$[\mathbf{E}^{\mathrm{s}}]^i_n = \frac{\sum\limits_{b=1}^{N_{\mathrm{b}}} G\left(\frac{d_{n,b}}{r_{\mathrm{s}}}\right) [\mathbf{E}^{\mathrm{r}}_b]^i_n}{\sum\limits_{b=1}^{N_{\mathrm{b}}} G\left(\frac{d_{n,b}}{r_{\mathrm{s}}}\right)}, \tag{30}$$

where $d_{n,b}$ is the distance between the $n$-th grid point and the center of the $b$-th block, $r_{\mathrm{s}}$ is the *smoothing radius*, a free parameter potentially different from $r$ and $G$ is a taper function, potentially different from the one used to compute the local weights.

If the resampling is performed using a "select and duplicate" algorithm (see Sect. 2.3), for example the SU sampling algorithm, then define $\phi_b$ as the resampling map for the $b$-th block, i.e. the map computed with the local weights $w^i_b$ such that $\phi_b(i)$ is the index of the $i$-th selected particle. $\mathbf{E}$ being the prior ensemble matrix, Eq. (30) becomes

$$[\mathbf{E}^{\mathrm{s}}]^i_n = \frac{\sum\limits_{b=1}^{N_{\mathrm{b}}} G\left(\frac{d_{n,b}}{r_{\mathrm{s}}}\right) [\mathbf{E}]^{\phi_b(i)}_n}{\sum\limits_{b=1}^{N_{\mathrm{b}}} G\left(\frac{d_{n,b}}{r_{\mathrm{s}}}\right)}. \tag{31}$$

Finally, the ensemble is updated as

$$\mathbf{E} \leftarrow \alpha_{\mathrm{s}} \mathbf{E}^{\mathrm{s}} + (1 - \alpha_{\mathrm{s}}) \mathbf{E}^{\mathrm{r}}, \tag{32}$$

where $\mathbf{E}^{\mathrm{r}}$ is the resampled ensemble matrix implicitly defined by step 5 of Algorithm 1 and $\alpha_{\mathrm{s}}$ is the *smoothing strength*, a free parameter in $[0, 1]$ that controls the intensity of the smoothing. When $\alpha_{\mathrm{s}} = 0$, no smoothing is performed and when $\alpha_{\mathrm{s}} = 1$, the analysis ensemble is totally replaced by the smoothed ensemble.

Algorithm 2 describes the analysis step for a generic LPF$^x$ with smoothing by weights. The original LPF of Penny and Miyoshi (2016) can be recovered if:

- blocks have size 1 grid point (hence there is no distinction between grid points and blocks);

- the local weights are computed using Eq. (29);

- $G$ is a top hat function;

- the resampling method is the SU sampling algorithm;

- $r_s$ is set to be equal to $r$;

- $\alpha_s$ is set to 0.5.

The method described here is a generalisation of their algorithm.

Note that when the resampling method is the SU sampling algorithm, the matrices $\mathbf{E}_b^r$ do not need to be explicitly computed. One just has to store in memory the resampling maps $\phi_b$, $b = 1 \ldots N_b$ and then use Eq. (31) to obtain the smoothed ensemble matrix $\mathbf{E}^s$.

The smoothing by weights step is an ad-hoc fix to reduce potential unphysical discontinuities after they have been introduced in the local resampling step. Its necessity hints that there is room for improvement in the design of the local resampling algorithms.

---

**Algorithm 2** Analysis step for a generic LPF$^x$ with smoothing by weights

---

**Require:** Prior ensemble $\boldsymbol{x}^i$, $i = 1 \ldots N_e$

1: **for** $b = 1$ **to** $N_b$ **do**

2:      Compute the local weights $w_b^i$ using Eq. (28) or (29)

3:      Resample the local ensemble $\mathbf{E}_{|b}$ with weights $w_b^i$ as $\mathbf{E}_{|b}^r$

4:      Resample the global ensemble $\mathbf{E}$ with weights $w_b^i$ as $\mathbf{E}_b^r$

5: **end for**

6: Assemble the locally resampled ensembles $\mathbf{E}_{|b}^r$ as $\mathbf{E}^r$

7: Compute the smoothed ensemble matrix $\mathbf{E}^s$ using Eq. (30)

8: Update the ensemble matrix $\mathbf{E}$ using Eq. (32)

9: **return** Updated ensemble $\boldsymbol{x}^i$, $i = 1 \ldots N_e$.

---

#### 4.4.2 Refining the sampling algorithms

In this section, we study several properties of the local resampling algorithm that might help dealing with the discontinuity issue: balance, adjustment and random numbers.

A "select and duplicate" sampling algorithm is said to be *balanced* if, for $i = 1 \ldots N_e$ the number of copies of the $i$-th particle selected by the algorithm does not differ by more than one unity from $w^i N_e$. For example, this is the case of the SU sampling but not the multinomial resampling algorithm.

A "select and duplicate" sampling algorithm is said to be *adjustment-minimising* if the indices of the particles selected by the algorithm are reordered to maximise the number of indices $i \in \{1 \ldots N_e\}$ such that the $i$-th updated particle is a copy of the $i$-th original particle. The SU sampling and the multinomial resampling algorithms can be simply modified to yield adjustment-minimising resampling algorithms.

While performing the resampling independently for each block, one can use the same random number(s) in the local resampling of each block.

Using the same random number(s) for the resampling of all blocks avoids a stochastic source of unphysical discontinuity. Choosing balanced and adjustment-minimising resampling algorithms is an attempt to include some kind of continuity in the map {local weights} $\mapsto$ {locally updated particles} by minimising the occurrences of composite particles. However, these properties cannot eliminate all sources of unphysical discontinuity. Indeed, ultimately composite particles will be built — if not then localisation would not be necessary — and there is no mechanism to reduce unphysical discontinuities in them. These properties have been first introduced in the "naive" local ensemble Kalman particle filter of Robert and Künsch (2017).

### 4.4.3   Using optimal transport in ensemble space

As mentioned in Sect. 2.3, using the optimal transport (OT) theory to design a resampling algorithm was first investigated in the ETPF algorithm of Reich (2013).

Applying optimal ensemble coupling to the SBD localisation frameworks results in a local LET resampling method, whose local transformation at each block $\mathbf{T}_b$ solves the discrete OT problem

$$\min_{\mathbf{T}_b \in \mathcal{T}_b} \sum_{i,j=1}^{N_e} [\mathbf{T}_b]^{i,j} c_b^{i,j}, \tag{33}$$

where $\mathcal{T}_b$ is the set of $N_e \times N_e$ transformations satisfying the normalisation constraint Eq. (13) and the local first-order accuracy constraint

$$\sum_{j=1}^{N_e} [\mathbf{T}_b]^{i,j} = N_e w_b^i, \quad i = 1 \ldots N_e. \tag{34}$$

In the ETPF, the coefficients $c^{i,j}$ were chosen as the squared $L^2$ distance between the whole $i$-th and $j$-th particles as in Eq. (15). Since we perform a local resampling step, it seems more appropriate to use a local criterion, such as

$$c_b^{i,j} = \sum_{n=1}^{N_x} \left( x_n^i - x_n^j \right)^2 G \left( \frac{d_{n,b}}{r_d} \right), \tag{35}$$

where $d_{n,b}$ is the distance between the $n$-th grid point and the center of the $b$-th block, $r_d$ is the *distance radius*, another free parameter and $G$ is a taper function, potentially different from the one used to compute the local weights.

To summarise, Algorithm 3 describes the analysis step for a generic LPF$^x$ that uses optimal ensemble coupling as local resampling algorithm. Localisation was first included in the ETPF algorithm by Cheng and Reich (2015), in a similar way as in the SBD localisation formalism. Hence Algorithm 3 can be seen as a generalisation of the local ETPF of Cheng and Reich (2015) that includes the concept of local state blocks.

---

**Algorithm 3** Analysis step for a generic LPF$^x$ using optimal ensemble coupling for the local resampling

---

**Require:** Prior ensemble $\boldsymbol{x}^i$, $i = 1 \ldots N_e$

1: **for** $b = 1$ **to** $N_b$ **do**

2:     Compute the local weights $w_b^i$ using Eq. (28) or (29)

3:     Compute the local coefficients $c_b^{i,j}$ with Eq. (35)

4:     Solve the minimisation problem Eq. (33) for $\mathbf{T}_b$

5:     Transform local ensemble: $\mathbf{E}_{|b}^t \leftarrow \mathbf{E}_{|b} \mathbf{T}_b$

6: **end for**

7: Assemble the locally transformed ensemble $\mathbf{E}_{|b}^t$ as $\mathbf{E}^t$

8: Update the ensemble as $\mathbf{E} \leftarrow \mathbf{E}^t$

9: **return** Updated ensemble $\boldsymbol{x}^i$, $i = 1 \ldots N_e$

---

On each block, the linear transformation establishes a strong connection between the prior and the updated ensembles. Moreover, there is no stochastic variation of the coupling at each block. This means that the spatial coherence can be (at least partially) transferred from the prior to the updated ensemble.

### 4.4.4 Using optimal transport in state space

In Sect. 4.4.3, the discrete OT theory was used to compute a linear transformation between the prior and the updated ensembles. 475 Following these ideas, we would like to use OT directly in state space. In more than one spatial dimension, the continuous OT problem is highly non-trivial and numerically challenging (Villani, 2009). Therefore, we will restrict ourselves to the case where blocks have size 1 grid point. Hence there is no distinction between blocks and grid points.

For each state variable $n$, we define the prior (marginal) pdf $p_n^f$ as the empirical density of the unweighted prior ensemble $\{x_n^i, i = 1 \ldots N_e\}$ and the analysis pdf $p_n^a$ as the empirical density of the prior ensemble, weighted by the analysis weights 480 $\{(x_n^i, w_n^i), i = 1 \ldots N_e\}$. We seek the map $T_n$ that solves the following OT problem:

$$\min_{T \in \mathcal{T}_n^{f \to a}} \int |x_n - T(x_n)|^2 \, \mathrm{d}x_n, \tag{36}$$

where $\mathcal{T}_n^{f \to a}$ is the set of maps $T$ that transport $p_n^f$ into $p_n^a$:

$$p_n^f = p_n^a \circ T \cdot \mathrm{Jac}(T), \tag{37}$$

with $\mathrm{Jac}(T)$ being the absolute value of the determinant of the Jacobian matrix of $T$.

In one dimension, this transport map is also known to be the *anamorphosis* from $p_n^\text{f}$ to $p_n^\text{a}$ and its computation is immediate:

$$T_n = (c_n^\text{a})^{-1} \circ c_n^\text{f}, \tag{38}$$

where $c_n^\text{f}$ and $c_n^\text{a}$ are the cumulative density function (cdf) of $p_n^\text{f}$ and $p_n^\text{a}$ respectively. Since $T_n$ maps the prior ensemble to an ensemble whose empirical density is $p_n^\text{a}$, the images of the prior ensemble members by $T_n$ are suitable candidates for updated ensemble members.

The computation of $T_n$ using Eq. (38) requires a continuous representation for the empirical densities $p_n^\text{f}$ and $p_n^\text{a}$. An appealing approach to obtain it is to use the kernel density estimation (KDE) theory (Silverman, 1986; Musso et al., 2001). In this context, the prior density can be written

$$p_n^\text{f}(x_n) = \alpha_n^\text{f} \sum_{i=1}^{N_\text{e}} K\left(\frac{x_n - x_n^i}{h\sigma_n^\text{f}}\right), \tag{39}$$

while the updated density is

$$p_n^\text{a}(x_n) = \alpha_n^\text{a} \sum_{i=1}^{N_\text{e}} w_n^i K\left(\frac{x_n - x_n^i}{h\sigma_n^\text{a}}\right). \tag{40}$$

$K$ is the regularisation kernel, $h$ is the bandwidth, a free parameter, $\sigma_n^\text{f}$ and $\sigma_n^\text{a}$ are the empirical standard deviation of respectively the unweighted ensemble $\{x_n^i,\ i=1\dots N_\text{e}\}$ and the weighted ensemble $\{(x_n^i, w_n^i),\ i=1\dots N_\text{e}\}$ and $\alpha_n^\text{f}$ and $\alpha_n^\text{a}$ are normalisation constants.

According to the KDE theory, when the underlying distribution is Gaussian, the optimal shape for $K$ is the Epanechnikov kernel (quadratic functions). Yet, there is no reason to think that this will also be the case for the prior density. Besides, the Epanechnikov kernel, having a finite support, generally leads to a poor representation of the distribution tails and it is a potential source of indetermination in the definition of the cdfs. That is why it is more common to use a Gaussian kernel for $K$. However, in this case, the computational cost associated to the cdf of the kernel (the error function) becomes significant. Hence, as an alternative, we choose to use the Student's t-distribution with two degrees of freedom. It is similar to a Gaussian but it has heavy tails and its cdf is fast to compute. It was also shown to be a better representation of the prior density than a Gaussian in an EnKF context (Bocquet et al., 2015).

To summarise, Algorithm 4 describes the analysis step for a generic LPF$^\text{x}$ that uses anamorphosis as local resampling algorithm.

The local resampling algorithm using anamorphosis is, as well as the algorithm using optimal ensemble coupling, a deterministic transformation. This means that unphysical discontinuities due to different random realisations over the grid points are avoided. As explained by Poterjoy (2016), in such an algorithm the updated ensemble members have the same *quantiles* as the prior ensemble members. The quantile property should be to some extent regular in space — for example if the spatial discretisation is fine enough — and this kind of regularity is transferred in the updated ensemble.

When defining the prior and the corrected densities with Eqs. (39) and (40), we introduce some regularisation whose magnitude is controlled through the bandwidth parameter $h$. Regularisation is necessary to obtain continuous pdfs. Yet, it introduces

**Algorithm 4** Analysis step for a generic LPF$^x$ using anamorphosis for the local resampling

---

**Require:** Prior ensemble $\boldsymbol{x}^i$, $i = 1 \dots N_e$

1: **for** $n = 1$ **to** $N_x$ **do**

2:      Compute the local weights $w_n^i$ using Eq. (28) or (29)

3:      Compute the empirical standard deviations $\sigma_n^f$ and $\sigma_n^a$

4:      Compute $c_n^f$ and $c_n^a$ by integrating Eqs. (39) and (40)

5:      **for** $i = 1$ **to** $N_e$ **do**

6:          Compute $p^i = c_n^f\left(x_n^i\right)$

7:          Solve $c_n^a\left(x_n^i\right) = p^i$ for the updated local particle $x_n^i$

8:      **end for**

9: **end for**

10: **return** Updated ensemble $\boldsymbol{x}^i$, $i = 1 \dots N_e$.

---

an additional bias in the analysis step. Typical values of $h$ should be around 1, with larger ensemble sizes $N_e$ requiring smaller values for $h$. More generally, regularisation is widely used in PF algorithms as a fix to avoid (or at least limit the impact of) weight degeneracy, though its implementation (see Sect. 5.2) is usually different from the method used in this section.

The refinements of the resampling algorithms suggested in Sect. 4.4.2 were designed to minimise the number of unphysical discontinuities in the local resampling step. The goal of the smoothing by weights step is to mitigate potential unphysical discontinuities after they have been introduced. On the other hand, the local resampling algorithms based on OT are designed to mitigate the unphysical discontinuities themselves. The main difference between the algorithm based on optimal ensemble coupling and the one based on anamorphosis is that the first one is formulated in the ensemble space whereas the second one

is formulated in the state space. That is to say in the first case we build an ensemble transformation $\mathbf{T}_b$ whereas in the second case we build a state transformation $T_n$.

     Due to computational considerations, the optimisation problem Eq. (36) was only considered in one dimension. Hence, contrary to the local resampling algorithm based on optimal ensemble coupling, the one based on anamorphosis is purely one-dimensional and can only be used with blocks of size 1 grid point.

The design of the resampling algorithm based on anamorphosis has been inspired from the kernel density distribution mapping (KDDM) step of the LPF algorithm of Poterjoy (2016) which will be introduced in Sect. 7.3. However, the use of OT has different purposes. In our algorithm, we use the anamorphosis transformation to sample particles from the analysis density, whereas the KDDM step of Poterjoy (2016) is designed to correct the posterior particles — they have already been transformed — with consistent high-order statistical moments.

## 4.5 Summary for the LPF$^{\times}$ algorithms

### 4.5.1 Highlights

In this section, we have constructed a generic SBD localisation framework, which we have used to define the LPF$^{\times}$s, our first category of LPF methods. The LPF$^{\times}$ algorithms are characterised by: the geometry of the blocks and domains (i.e. the definition of the local weights) and the resampling algorithm. As shown by Rebeschini and van Handel (2015), the LPF$^{\times}$ algorithms have potential to beat the curse of dimensionality. However, unphysical discontinuities are likely to arise after the assembling of locally resampled particles (van Leeuwen, 2009). In this section, we have proposed to mitigate these discontinuities by improving the design of the local resampling step. We distinguished four approaches:

1. A smoothing by weights step can be applied after the local resampling step in order to reduce potential unphysical discontinuities. Our method is a generalisation of the original smoothing designed by Penny and Miyoshi (2016) that includes spatial tapering, a smoothing strength and that is suited to the use of state blocks.

2. Simple properties of the local resampling algorithms can be used in order to minimise the occurences of unphysical discontinuity as shown by Robert and Künsch (2017).

3. Using the principles of discrete OT, we have proposed a resampling algorithm based on a local version of the ETPF of Reich (2013). This algorithm is similar to the PF–EnKF part of the PF–EnKF hybrid derived by Chustagulprom et al. (2016) but it includes a more general transport cost and it is suited to the use of blocks and any resampling algorithm. By construction, the distance between the prior and the analysis local ensembles is minimised.

4. Combining the continuous OT problem with the KDE theory, we have derived a new local resampling algorithm based on anamorphosis. We have shown how it helps mitigate the unphysical discontinuities.

In Sect. 4.5.2, we discuss the numerical complexity and in Sect. 4.5.4 the asymptotic limits of the proposed LPF$^{\times}$ algorithms. In Sect. 4.5.3, we propose guidelines that should inform our choice of the key parameters when implementing these algorithms.

### 4.5.2 Numerical complexity

We define the auxiliary quantities $N_{\mathrm{b}}^{\ell}(R)$, $N_{\mathrm{x}}^{\ell}(R)$ and $N_{\mathrm{y}}^{\ell}(R)$ by

$$N_{\mathrm{x}}^{\ell}(R) = \max_{b \in \{1\ldots N_{\mathrm{b}}\}} \mathrm{Card}\{n \in \{1\ldots N_{\mathrm{x}}\} \backslash d_{n,b} \leq R\}, \tag{41}$$

$$N_{\mathrm{b}}^{\ell}(R) = \max_{n \in \{1\ldots N_{\mathrm{x}}\}} \mathrm{Card}\{b \in \{1\ldots N_{\mathrm{b}}\} \backslash d_{n,b} \leq R\}, \tag{42}$$

$$N_{\mathrm{y}}^{\ell}(R) = \max_{q \in \{1\ldots N_{\mathrm{b}}\}} \mathrm{Card}\{q \in \{1\ldots N_{\mathrm{y}}\} \backslash d_{q,b} \leq R\}. \tag{43}$$

$N_{\mathrm{y}}^{\ell}(R)$ is the maximum number of observation sites in a local domain of radius $R$. $N_{\mathrm{b}}^{\ell}(R)$ and $N_{\mathrm{x}}^{\ell}(R)$ are the corresponding quantities for the neighborhood grid points and blocks. In a $d$-dimensional spatial space, these quantities are at most proportional to $R^d$.

The complexity of the LPF$^x$ analysis is the sum of the complexity of computing all local weights and the complexity of the resampling. Using Eq. (28) or (29), we conclude that the complexity of computing the local weights is $\mathcal{O}\left(N_e T_{\mathcal{H}} + N_b N_e N_y^\ell(r)\right)$, which depends on the localisation radius $r$ and on the complexity $T_{\mathcal{H}}$ of applying the observation operator $\mathcal{H}$ to a vector. In the following paragraphs we detail the complity of each resampling algorithm.

When using the multinomial resampling of the SU sampling algorithm for the local resampling, the total complexity of the resampling step is $\mathcal{O}\left(N_x N_e\right)$.

When using optimal ensemble coupling, the resampling step is computationally more expensive because it requires to solve one optimisation problem for each block. The minimisation coefficients Eq. (35) are computed with complexity $\mathcal{O}\left(N_e^2 N_x^\ell(r_d)\right)$, which depends on the distance radius $r_d$. The discrete OT problem Eq. (33) is a particular case of the minimum-cost flow problem and can be solved quite efficiently using the algorithm of Pele and Werman (2009) with complexity $\mathcal{O}\left(N_e^2 \ln N_e\right)$. Applying the transformation to the block has complexity $\mathcal{O}\left(N_x N_b^{-1} N_e^2\right)$. Finally, the total complexity of the resampling step is $\mathcal{O}\left(N_b N_e^2 N_x^\ell(r_d) + N_b N_e^2 \ln N_e + N_x N_e^2\right)$.

When using optimal transport in state space, every one-dimensional anamorphosis is computed with complexity $\mathcal{O}\left(N_p\right)$ where $N_p$ is the one-dimensional resolution for each state variable. Therefore the total complexity of the resampling step is $\mathcal{O}\left(N_x N_e N_p\right)$.

When using the smoothing by weights step with the multinomial resampling or the SU sampling algorithm, the smoothed ensemble Eq. (31) is computed with complexity $\mathcal{O}\left(N_x N_e N_b^\ell(r_s)\right)$, which depends on the smoothing radius $r_s$ and the updated ensemble Eq. (32) is computed with complexity $\mathcal{O}\left(N_x N_e\right)$. Therefore, the total complexity of the resampling and the smoothing steps is $\mathcal{O}\left(N_x N_e N_b^\ell(r_s)\right)$.

For comparison, the more costly operation in the local analysis of a local EnKF algorithm is to compute the singular value decomposition of a $N_y^\ell(r) \times N_e$ matrix, which has complexity $\mathcal{O}\left(N_y^\ell(r) N_e^2\right)$ assuming that $N_e \leq N_y^\ell(r)$. The total complexity for a local EnKF algorithm depends on the specific implementation but should be at least $\mathcal{O}\left(N_b N_y^\ell(r) N_e^2\right)$.

In this complexity analysis, the influence of the parameters $r$, $r_d$ and $r_s$ is explicitly shown because a practitioner must be aware of the numerical cost of increasing these parameters. Since the resampling is performed independently for each block, this algorithmic step (which is the most costly step in practice) can be carried out in parallel, allowing a theoretical gain up to a factor $N_b$.

### 4.5.3 Choice of key parameters

The localisation radius $r$ controls the number of observation sites in the local domains $N_y^\ell(r)$ and the impact of the curse of dimensionality. To avoid immediate weight degeneracy, $r$ should therefore be relatively small — smaller than what would be required for an EnKF using domain localisation for example. This is especially true for realistic models with two or more spatial dimensions in which $N_y^\ell(r)$ grows as $r^2$ or more. In this case, it can happen that the localisation radius $r$ have to be too small for the method to follow the truth trajectory (either because too much information is ignored, or because there is too much spatial variation in the local weights) which would mean that localisation alone would not be enough to make PF methods operational.

For a local EnKF algorithm, gathering grid points into blocks is an approximation that reduces the numerical cost of the analysis steps by reducing the number of local analyses to perform. For an LPF$^x$ algorithm, the local analyses should in general be faster (see the complexity analysis in Sect. 4.5.2). In this case, using larger blocks is a way to decrease the proportion of block borders, which are potential spots for unphysical discontinuities. However, increasing the size of the blocks reduces the number of degrees of freedom to counteract the curse of dimensionality. It also introduces an additional bias in the local weight update, Eq. (28) or (29), since the local weights are computed relatively to the block centers. This issue was identified by Rebeschini and van Handel (2015) as a source of spatial inhomogeneity of the error. For these reasons, the blocks should be small (no more than a few grid points). Only large ensembles could potentially benefit from larger blocks.

More discussion regarding the choice of the localisation radius $r$ and the number of blocks $N_{\mathrm{b}}$, but also regarding the choice of other parameters (the smoothing radius $r_{\mathrm{s}}$, the smoothing strength $\alpha_{\mathrm{s}}$, the distance radius $r_{\mathrm{d}}$ and the regularisation bandwidth $h$) can be found in Sect. 5.

### 4.5.4 Asymptotic limit

An essential property of PF algorithms is that they are asymptotically Bayesian: as stated in Sect. 2.2, under reasonable assumptions the empirical analysis density converges to the true analysis density for the weak topology on the set of probability measures over $\mathbb{R}^{N_{\mathrm{x}}}$ in the limit $N_{\mathrm{e}} \to \infty$. In this section, we study under which conditions the LPF$^x$ analysis can be equivalent to a (global) PF analysis and therefore be asymptotically Bayesian.

In the limit of very large localisation radius, $r \to \infty$, the local weights Eq. (28) and (29) are equal to the (global) weights of the (global) PF. However, this does not imply that the LPF$^x$ analysis is equivalent to a PF analysis because the resampling is performed independently for each block. Yet we can distinguish the following cases in the limit $r \to \infty$:

– When using independent multinomial resampling or SU sampling for the local resampling, if one uses the same random number for all blocks (this property is always true if $N_{\mathrm{b}} = 1$), then the LPF$^x$ analysis is equivalent to the analysis of the PF.

– When using the smoothing by weigths step with the multinomial resampling or the SU sampling, if one uses the same random number for all blocks then the smoothed ensemble Eq. (31) is equal to the (locally) resampled ensemble and the smoothing has no effect: we are back to the first case.

– When using optimal ensemble coupling for the local resampling, in the limit $r_{\mathrm{d}} \to \infty$, the LPF$^x$ analysis is equivalent to the analysis of the (global) ETPF.

For other cases, we cannot give a firm conclusion:

– When using independent multinomial resampling or SU sampling for the local resampling with different random number for all blocks, then the updated particles are distributed according to the product of the marginal analysis density Eq. (26), which is in general different from the analysis density even in the limit $r \to \infty$.

- For the same reason, when using anamorphosis for the local resampling, we could not find a proof that the LPF$^{\text{x}}$ analysis is asymptotically Bayesian, even in the limit $h \to 0$ and $r \to \infty$.

- When using the smoothing by weigths step with the multinomial resampling or the SU sampling, in the limit $r \to \infty$ and $r_{\text{s}} \to \infty$ the smoothed ensemble Eq. (31) can be different from the updated ensemble of the global PF because the resampling is performed independently for each block.

## 5 Numerical illustration of LPF$^{\text{x}}$ algorithms with the Lorenz-96 model

### 5.1 Model specifications

In this section, we illustrate the performance of LPF$^{\text{x}}$s with twin simulations of the L96 model in the standard (mildly nonlinear) configuration described in Appendix A3. For this series of experiments, as for all experiments in this paper, the synthetic truth is computed without model error. This is usually a stringent constraint for the PF methods for which accounting for model error is a means for regularisation. But on the other hand it allows for a fair comparison with the EnKF and it avoids the issue of defining a realistic model noise.

The distance between the truth and the analysis is measured with the average analysis root mean square error, hereafter simply called the RMSE. To ensure the convergence of the statistical indicators, the runs are at least $5 \times 10^4 \Delta t$ long with an additional $10^3 \Delta t$ spin-up period. An advantage of using PF methods is that it should asymptotically yield sharp though reliable ensembles. This may not be entirely reflected in the RMSE. However, not only does the RMSE offer a clear ranking of the algorithms but it is an indicator that measures the adequacy to the primary goal of data assimilation, i.e. mean state estimation. Moreover, for a sufficiently cycled DA problem, it seems likely that good RMSE scores can only be achieved with ensembles of good quality in the light of most other indicators. Nonetheless, in addition to the RMSE, rank histograms meant to assess the quality of the ensembles are computed and reported in Appendix D for a selection of experiments.

For the localisation, we assume that the grid points are positioned on an axis with a regular spacing of 1 unit of length and periodic boundary conditions consistent with the system size. Therefore, the local domain centered on the $n$-th grid point is composed of the points $\{n - \lfloor r \rfloor \dots n + \lfloor r \rfloor\}$, where $\lfloor r \rfloor$ is the integer part of the localisation radius and the $N_{\text{b}}$ blocks consist of $N_{\text{x}}/N_{\text{b}}$ consecutive grid points.

This filtering problem has been widely used to asses the performance of DA algorithms. In this configuration, nonlinearities in the model are rather mild and representative of synoptic scale meteorology and the error distributions are close to Gaussian. As a reference, the evolution of the RMSE as a function of the ensemble size $N_{\text{e}}$ is shown in Fig. 4 for the ensemble transform Kalman filter (ETKF) and its local version (LETKF). For each value of $N_{\text{e}}$, the multiplicative inflation parameter and the localisation radius (for the LETKF) are optimally tuned to yield the lowest RMSE. In most of the following figures related to the L96 test series, we draw a baseline at $0.2$, roughly the RMSE of the LETKF with $N_{\text{e}} = 10$ particles. Note that slightly lower RMSE scores can be achieved with larger ensembles.

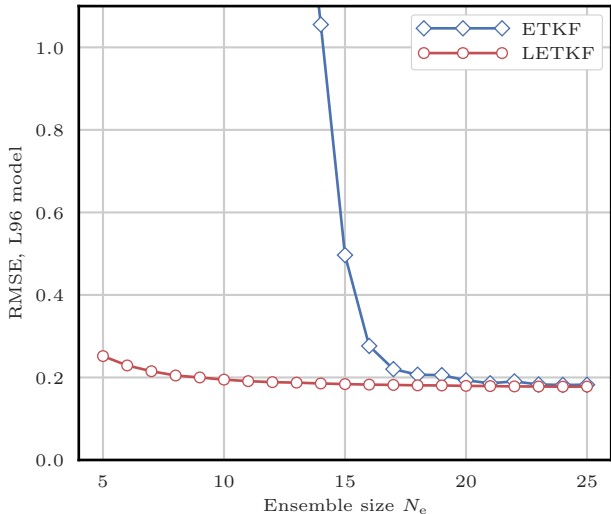

**Figure 4.** RMSE as a function of the ensemble size $N_e$ for the ETKF and the LETKF.

## 5.2 Perfect model and regularisation

The application of PF algorithms to this chaotic model without error leads to a fast collapse. Even with stochastic models that account for some model error, PF algorithms experience weight degeneracy when the model noise is too low. Therefore, PF practitioners commonly include some additional jitter to mitigate the collapse (e.g. Pham, 2001). As described by Musso et al. (2001), jitter can be added in two different ways.

### 5.2.1 Pre-regularisation

First, the prediction and sampling step, Eq. (7), can be performed using a stochastic extension of the model:

$$\boldsymbol{x}_{k+1}^i - \mathcal{M}\left(\boldsymbol{x}_k^i\right) = \boldsymbol{w}_k \sim \mathcal{N}\left(\boldsymbol{0}, q^2 \mathbf{I}\right), \tag{44}$$

where $\mathcal{M}$ is the model associated to the integration scheme of the ordinary differential equations (ODEs), $\mathcal{N}\left(\boldsymbol{v}, \boldsymbol{\Sigma}\right)$ is the normal distribution with mean $\boldsymbol{v}$ and covariance matrix $\boldsymbol{\Sigma}$ and $q$ is a tunable parameter. This jitter is meant to compensate for the deterministic nature of the given model. In this case, the truth could be seen as a trajectory of the perturbed model Eq. (44) with a realisation of the noise that is identically zero. In the literature, this method is called pre–regularisation (Le Gland et al., 1998) because the jitter is added before the correction step.

### 5.2.2 Post-regularisation

Second, a regularisation step can be added after a full analysis cycle:

$$\boldsymbol{x}_{k+1}^i \leftarrow \boldsymbol{x}_{k+1}^i + \boldsymbol{u}, \quad \boldsymbol{u} \sim \mathcal{N}\left(\boldsymbol{0}, s^2 \mathbf{I}\right), \tag{45}$$

where $s$ is a tunable parameter. As opposed to the first method, it can be seen as a jitter before integration: the noise is integrated by the model before the next analysis step, while smoothing potential unphysical discontinuities. In some ways this method is similar to additive inflation in EnKF algorithms. It is called post–regularisation (Musso and Oudjane, 1998; Oudjane and Musso, 1999) because the jitter is added after the correction step.

### 5.2.3 Numerical complexity and asymptotic limit

Both regularisation steps have numerical complexity $\mathcal{O}\left(N_{\mathrm{x}} N_{\mathrm{e}} T_{\mathrm{r}}\right)$, with $T_{\mathrm{r}}$ being the complexity of drawing one random number according to the univariate standard normal law $\mathcal{N}\left(0, 1\right)$.

The exact LPF is recovered in the limit $q \to 0$ and $s \to 0$.

### 5.2.4 Standard S(IR)$^{\mathrm{x}}$R algorithm

With optimally tuned jitter for the standard L96 model, the bootstrap PF algorithm requires about $200$ particles to give on average more information than the observations.[2] With $10^3$ particles, its RMSE is around $0.6$ and with $10^4$ it is around $0.4$.

We define the standard S(IR)$^{\mathrm{x}}$R algorithm — sampling, importance, resampling, regularisation, the $x$ exponent meaning that steps in parentheses are performed locally for each block — as the LPF$^{\mathrm{x}}$ (Algorithm 1) with the following characteristics:

- grid points are gathered into $N_{\mathrm{b}}$ blocks of $N_{\mathrm{x}}/N_{\mathrm{b}}$ connected grid points;

- jitter is added after the integration using Eq. (44) with a standard deviation controlled by $q$;

- the local weights are computed using the Gaussian tapering of observation influence given by Eq. (29), where $G$ is the Gaspari–Cohn function;

- the local resampling is performed independently for each block with the adjustment-minimising SU sampling algorithm;

- jitter is added at the end of each assimilation cycle using Eq. (45) with a standard deviation controlled by $s$.

The standard deviation of the jitter after integration ($q$) and before integration ($s$) shall be called "integration jitter" and "regularisation jitter", respectively. The S(IR)$^{\mathrm{x}}$R algorithm has five parameters: $(N_{\mathrm{e}}, N_{\mathrm{b}}, r, q, s)$. All algorithms tested in this section are variants of this standard algorithm and are named S($\alpha\beta$)$^{\mathrm{x}}\gamma\delta$ with the conventions detailed in Table 1. Table 2 lists all LPF$^{\mathrm{x}}$ algorithms tested in this section and reports their characteristics according to the convention of Table 1.

### 5.3 Tuning the localisation radius

We first check that, in this standard configuration, localisation is working by testing the S(IR)$^{\mathrm{x}}$R algorithm with $N_{\mathrm{b}} = 40$ blocks of size 1 grid point. We take $N_{\mathrm{e}} = 10$ particles, $q = 0$ (perfect model) and several values for the regularisation jitter $s$. The evolution of the RMSE as a function of the localisation radius $r$ is shown in Fig. 5. With SBD localisation, the LPF yields an RMSE around $0.45$ in a regime where the bootstrap PF algorithm is degenerate. The compromise between bias (small values

---

[2]We have proven in this case that the RMSE, when computed between the observations $\boldsymbol{y}_k$ and truth $\boldsymbol{x}_k$, has an expected value of $0.98$.

**Table 1.** Nomenclature conventions for the $S(\alpha\beta)^x\gamma\delta$ algorithms. Capital letters refer to the main algorithmic ingredients: "I" for importance, "R" for reampling or regularisation, "T" for transport, "S" for smoothing. Subscripts are used to distinguish the methods in two different ways. Lower case subscripts refer to explicit concepts used in the method: "ng" for non-Gaussian, "su" for stochastic universal, "s" for state space and "c" for colour; while upper case subscripts refer to the work that inspired the method: "PM" for Penny and Miyoshi (2016) and "R" for Reich (2013). For simplicity, some subscripts are omitted: "g" for Gaussian, "amsu" for adjustment-minimising stochastic universal and "w" for white. Finally, note that we used the subscript "d" (for deterministic) to indicate that the same random numbers are used for the resampling over all blocks.

| | |
|---|---|
| $\alpha$ | Local importance weights (Sect. 4.2.2) |
| $I_{ng}$ | Eq. (28) [non-Gaussian] |
| $I$ | Eq. (29) [Gaussian] |
| $\beta$ | Local resampling algorithm (Sect. 4.4) |
| $R_{su}$ | SU sampling algorithm |
| $R_d$ | adjustment-minimising SU sampling algorithm with the same random numbers over all blocks |
| $R$ | adjustment-minimising SU sampling algorithm |
| $T_R$ | optimal transport in ensemble space |
| $T_s$ | optimal transport in state space |
| $\gamma$ | Smoothing-by-weights (Sect. 4.4.1) |
| $S_{PM}$ | enabled |
| $-$ | disabled |
| $\delta$ | Regularisation method (Sects. 5.2 and 5.8) |
| $R$ | white noise method |
| $R_c$ | coloured noise method |

of $r$, too much information is ignored or there is too much spatial variation in the local weights) and variance (large values of $r$, the weights are degenerate) reaches an optimum around $r = 3$ grid points. As expected, the local domains are quite small (5 observation sites) in order to efficiently counteract the curse of dimensionality.

### 5.4 Tuning the jitter

To evaluate the efficiency of the jitter, we experiment with the $S(IR)^xR$ algorithm with $N_e = 10$ particles, $N_b = 40$ blocks of size 1 grid point and a localisation radius $r = 3$ grid points. The evolution of the RMSE as a function of the integration jitter $q$ is shown in Fig. 6 and as a function of the regularisation jitter $s$ in Fig. 7.

**Table 2.** List of all LPF$^x$ algorithms tested in this article. For each algorithm, the main characteristics are reported with appropriate references. The last column indicate the section in which benchmarks based on the L96 model can be found.

| Algorithm | Local importance weights (Sect. 4.2.2) Eq. (28) [non-Gaussian] | Eq. (29) [Gaussian] | Local resampling algorithm (Sect. 4.4) | Subsection | Smoothing by weights (Sect. 4.4.1) – [disabled] | Eq. (30) [enabled] | Eq. (31) [enabled] | Regularisation method (Sects. 5.2 and 5.8) Eq. (45) [white] | Eq. (49) [colour] | L96 benchmark sections |
|---|---|---|---|---|---|---|---|---|---|---|
| $S(IR)^{\times}R$ | | ✓ | adjustment-minimising SU sampling | 4.4.2 | ✓ | | | ✓ | | 5.3 to 5.12 |
| $S(I_{ng}R)^{\times}R$ | ✓ | | adjustment-minimising SU sampling | 4.4.2 | ✓ | | | ✓ | | 5.6 |
| $S(IR_{su})^{\times}R$ | | ✓ | SU sampling | – | ✓ | | | ✓ | | 5.7 |
| $S(IR_{d})^{\times}R$ | | ✓ | adjustment-minimising SU sampling with the same random numbers | 4.4.2 | ✓ | | | ✓ | | 5.7 |
| $S(IR)^{\times}R_c$ | | ✓ | adjustment-minimising SU sampling | 4.4.2 | ✓ | | | | ✓ | 5.8 to 5.11 |
| $S(IR)^{\times}S_{PM}R$ | | ✓ | adjustment-minimising SU sampling | 4.4.2 | | | ✓ | ✓ | | 5.9 |
| $S(IR)^{\times}S_{PM}R_c$ | | ✓ | adjustment-minimising SU sampling | 4.4.2 | | | ✓ | | ✓ | 5.9, 5.12 |
| $S(IT_R)^{\times}R$ | | ✓ | optimal ensemble coupling | 4.4.3 | ✓ | | | ✓ | | 5.10, 5.12 |
| $S(IT_R)^{\times}R_c$ | | ✓ | optimal ensemble coupling | 4.4.3 | ✓ | | | | ✓ | 5.10 |
| $S(IT_R)^{\times}S_{PM}R$ | | ✓ | optimal ensemble coupling | 4.4.3 | | ✓ | | ✓ | | 5.10 |
| $S(IT_R)^{\times}S_{PM}R_c$ | | ✓ | optimal ensemble coupling | 4.4.3 | | ✓ | | | ✓ | 5.10 |
| $S(IT_s)^{\times}R$ | | ✓ | anamorphosis | 4.4.4 | ✓ | | | ✓ | | 5.11, 5.12 |
| $S(IT_s)^{\times}R_c$ | | ✓ | anamorphosis | 4.4.4 | ✓ | | | | ✓ | 5.11 |
| $S(IT_s)^{\times}S_{PM}R$ | | ✓ | anamorphosis | 4.4.4 | | ✓ | | ✓ | | 5.11 |

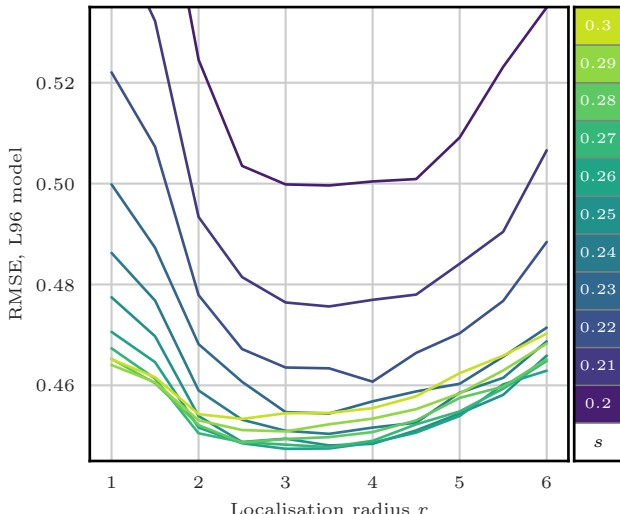

**Figure 5.** RMSE as a function of the localisation radius $r$ for the S(IR)$^\times$R algorithm with $N_\text{e} = 10$ particles, $N_\text{b} = 40$ blocks of size 1 grid point and no integration jitter ($q = 0$). For each $r$, several values for the regularisation jitter $s$ are tested as shown by the colour scale.

From these results, we can identify two regimes:

- with low regularisation jitter ($s < 0.15$), the filter stability is ensured by the integration jitter, with optimal values around $q = 1.25$;

- with low integration jitter ($q < 0.5$), the stability is ensured by the regularisation jitter, with optimal values around $s = 0.26$.

As expected, adding jitter before integration (i.e. with $s$) yields significantly better results. This indicates that the model integration indeed smoothes the jitter out and removes unphysical discontinuities for the correction step. We observed the same tendency for most LPFs tested in this article.

In the rest of this section, we take zero integration jitter ($q = 0$) and the localisation radius $r$ and the regularisation jitter $s$ are systematically tuned to yield the lowest RMSE score.

### 5.5 Increasing the size of the blocks

To illustrate the influence of the size of the blocks, we compare the RMSEs obtained by the S(IR)$^\times$R algorithm with various fixed number of blocks $N_\text{b}$. The evolution of the RMSE as a function of the ensemble size $N_\text{e}$ is shown in Fig. 8. For small ensemble sizes, using larger blocks is inefficient, because of the need for degrees of freedom to counteract the curse of dimensionality. Only very large ensembles benefit from using large blocks as a consequence of the reduction of proportion of block boundaries, potential spots for unphysical discontinuities.

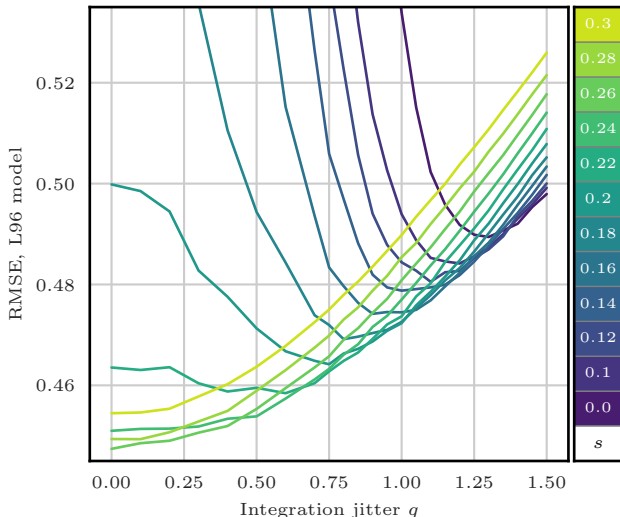

**Figure 6.** RMSE as a function of the integration jitter $q$ for the S(IR)$^x$R algorithm with $N_e = 10$ particles, $N_b = 40$ blocks of size 1 grid point and a localisation radius $r = 3$ grid points. For each $q$, several values for the regularisation jitter $s$ are tested as shown by the colour scale.

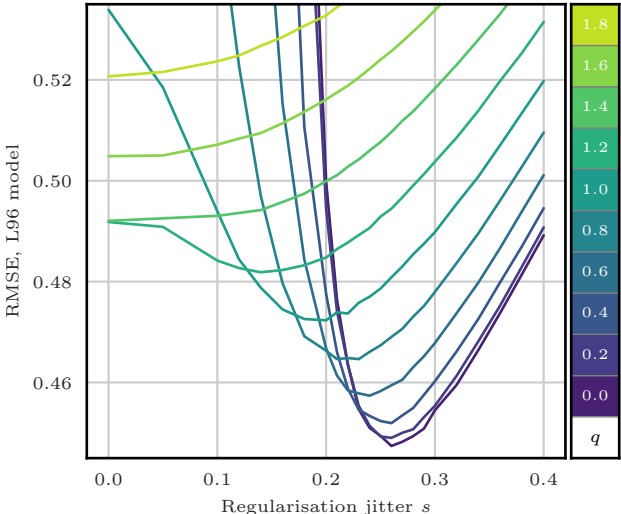

**Figure 7.** RMSE as a function of the regularisation jitter $s$ for the S(IR)$^x$R algorithm with $N_e = 10$ particles, $N_b = 40$ blocks of size 1 grid point and a localisation radius $r = 3$ grid points. For each $s$, several values for the integration jitter $q$ are tested as shown by the colour scale.

From now on, unless specified otherwise, we systematically test our algorithms with $N_b = 40, 20, 10$ blocks of respectively $1, 2, 4$ grid points and we keep the best RMSE score.

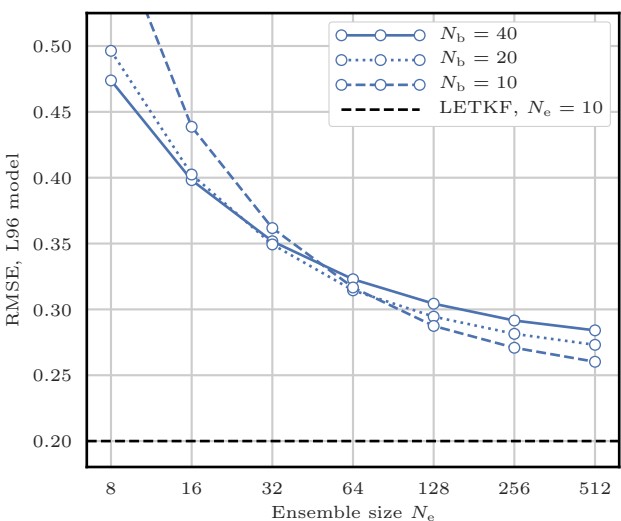

**Figure 8.** RMSE as a function of the ensemble size $N_e$ for the S(IR)$^x$R algorithm with respectively $N_b = 40, 20, 10$ blocks of size 1, 2, 4 grid points.

## 5.6 Choice of the local weights

To illustrate the influence of the definition of the local weights, we compare the RMSEs of the S(IR)$^x$R and the S($I_{ng}$R)$^x$R algorithms. These two variants only differ by their definition of the local importance weights: the S(IR)$^x$R algorithm uses the Gaussian tapering of observation influence defined by Eq. (29) while the S($I_{ng}$R)$^x$R algorithm uses the non-Gaussian tapering given by Eq. (28).

Figure 9 shows the evolution of the RMSE as a function of the ensemble size $N_e$. The Gaussian version of the definition of
735 the weights always yields better results. This is probably a consequence of the fact that, in this configuration nonlinearities are mild and the error distributions are close to Gaussian. In the following, we always use Eq. (29) to define the local weights.

## 5.7 Refining the stochastic universal sampling

In this section, we test the refinements of the sampling algorithms proposed in Sect. 4.4.2. To do this we compare, the RMSEs of the S(IR)$^x$R algorithms with those of:

– the S(IR$_d$)$^x$R algorithm, for which the same random numbers are used for the resampling of each block;

– the S(IR$_{su}$)$^x$R algorithm, which uses the SU sampling algorithm without the adjustment-minimising property.

Figure 10 shows the evolution of the RMSE as a function of the ensemble size $N_e$. The S(IR$_{su}$)$^x$R, the only algorithm that does not satisfy the adjustment-minimising property, yields higher RMSEs. This shows that the adjustment-minimising property is indeed an efficient way of reducing the number of unphysical discontinuities introduced during the resampling step.

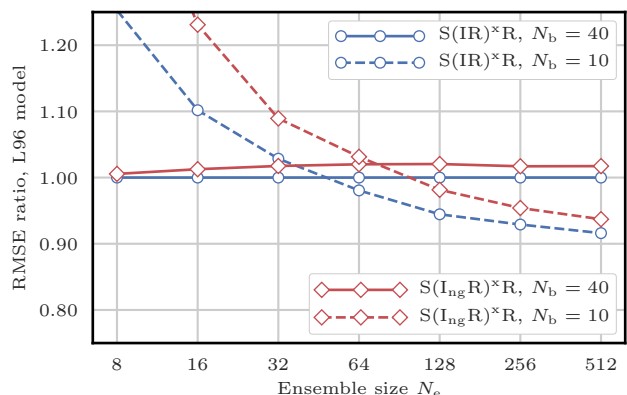

**Figure 9.** RMSE as a function of the ensemble size $N_e$ for the S(IR)$^\times$R and the S(I$_{ng}$R)$^\times$R algorithms with respectively $N_b = 40$, 10 blocks of size 1, 4 grid points. The scores are displayed in units of the RMSE of the S(IR)$^\times$R algorithm with $N_b = 40$ blocks of size 1 grid point.

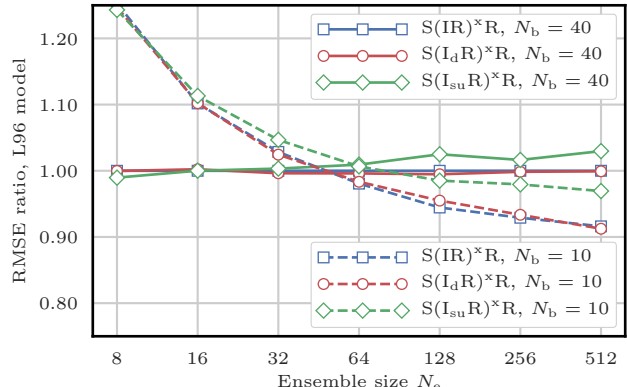

**Figure 10.** RMSE as a function of the ensemble size $N_e$ for the S(IR)$^\times$R, the S(IR$_d$)$^\times$R and the the S(IR$_{su}$)$^\times$R algorithms with respectively $N_b = 40$, 10 blocks of size 1, 4 grid points. The scores are displayed in units of the RMSE of the S(IR)$^\times$R algorithm with $N_b = 40$ blocks of size 1 grid point.

However, using the same random number for the resampling of each block does not produce significantly lower RMSEs. This method is insufficient to reduce the number of unphysical discontinuities introduced when assembling the locally updated particles. This is probably a consequence of the fact that the SU sampling algorithm only uses one random number to compute the resampling map. It also suggests that the specific realisation of this random number has a weak influence on long-term statistical properties.

In the following, when using the SU sampling algorithm, we always choose its adjustment-minimising form but we do not enforce the same random numbers over different blocks.

### 5.8 Colourising the regularisation

#### 5.8.1 Colourisation for global PFs

According to Eqs. (44) and (45), the regularisation jitters are white noises. In realistic models, different state variables may take their values in disjoint intervals (e.g., the temperature takes values around $300$ K and the wind speed can take its values between $-10$ and $10\,\mathrm{m\,s^{-1}}$) which makes these jittering methods inadequate.

It is hence a common procedure in ensemble DA to scale the regularisation jitter with statistical properties of the ensemble. In a (global) PF context, practitioners often "colourise" the Gaussian regularisation jitter with the empirical covariances of the ensemble as described by Musso et al. (2001). Since the regularisation jitter is added after the resampling step, it is scaled with the weighted ensemble before resampling in order to mitigate the effect of resampling noise.

More precisely, the regularisation jitter has zero mean and $N_\mathrm{x} \times N_\mathrm{x}$ covariance matrix given by

$$[\mathbf{\Sigma}]_{n,m} = \frac{\hat{h}}{1 - \sum\limits_{i=1}^{N_\mathrm{e}} (w^i)^2} \sum_{i=1}^{N_\mathrm{e}} w^i \left(x_n^i - \overline{x}_n\right)\left(x_m^i - \overline{x}_m\right), \tag{46}$$

where $\hat{h}$ is the bandwidth, a free parameter, and $\overline{x}_n$ is the ensemble mean for the $n$-th state variable:

$$\overline{x}_n = \frac{1}{N_e} \sum_{i=1}^{N_\mathrm{e}} x_n^i. \tag{47}$$

In practice, the $N_\mathrm{x} \times N_\mathrm{e}$ anomaly matrix $\mathbf{X}$ is defined by

$$[\mathbf{X}]_{n,i} = \sqrt{\frac{\hat{h} w^i}{1 - \sum\limits_{i=1}^{N_\mathrm{e}} (w^i)^2}} \left(x_n^i - \overline{x}_n\right), \tag{48}$$

and the regularisation is added as

$$\mathbf{E} \leftarrow \mathbf{E} + \mathbf{X}\mathbf{Z}, \tag{49}$$

where $\mathbf{E}$ is the ensemble matrix and $\mathbf{Z}$ is a $N_\mathrm{e} \times N_\mathrm{e}$ random matrix whose coefficients are distributed according to a normal law such that $\mathbf{X}\mathbf{Z}$ is a sample from the Gaussian distribution with zero mean and covariance matrix $\mathbf{\Sigma}$. In this case, the regularisation fits in the LET framework with a random transformation matrix.

Colourisation could be added as well to the integration jitter. However in this case, scaling the noise with the ensemble is less justified than for the regularisation jitter. Indeed, the integration noise is inherent to the perturbed model that is used to evolve each ensemble member independently. Hence PF practitioners often take a time–independent Gaussian integration noise whose covariance matrix does not depend on the ensemble but includes some off-diagonal terms based on the distance between grid points (e.g., Ades and van Leeuwen, 2015). However, as we mentioned in Sect. 5.4, we do not use integration jitter for the rest of this article.

### 5.8.2 Colourisation for LPFs

The $40$ variables of the L96 model in its standard configuration are statistically homogeneous with short-range correlations. This is the main reason of the efficiency of the white noise jitter in the $S(IR)^xR$ algorithm and its variants tested so far. We still want to investigate the potential gains of using coloured jitters in $LPF^x$s.

In the analysis step of $LPF^x$ algorithms, at each grid point there is a different set a local weights $w_n^i$, therefore it is not possible to compute the covariance of the regularisation jitter with Eq. (46). We propose two different ways of circumventing this obstacle.

A first approach could be to scale the regularisation with the locally resampled ensemble, since in this case all weights are equal. This is the approach followed by Reich (2013) and Chustagulprom et al. (2016) under the name "particle rejuvenation". However, this approach systematically leads to higher RMSEs for the $S(IR)^xR$ algorithm (not shown here). This can be potentially explained by two factors. First, the resampling could introduce noise in the computation of the anomaly matrix $\mathbf{X}$. Second, the fact that the resampling is performed independently for each block perturbs the propagation of multivariate properties (such as sample covariance) over different blocks.

In a second approach, the anomaly matrix $\mathbf{X}$ is defined by the weighted ensemble before resampling, i.e. using the local weights $w_n^i$, as follows:

$$[\mathbf{X}]_{n,i} = \sqrt{\frac{\hat{h}w_n^i}{1 - \sum\limits_{i=1}^{N_e}(w_n^i)^2}}\left(x_n^i - \overline{x}_n\right). \tag{50}$$

In this case, the Gaussian regularisation jitter has covariance matrix:

$$[\mathbf{\Sigma}]_{n,m} = \sum_{i=1}^{N_e}\frac{\hat{h}\sqrt{w_n^i w_m^i}\left(x_n^i - \overline{x}_n\right)\left(x_m^i - \overline{x}_m\right)}{\sqrt{\left(1 - \sum\limits_{i=1}^{N_e}(w_n^i)^2\right)\left(1 - \sum\limits_{i=1}^{N_e}(w_m^i)^2\right)}}, \tag{51}$$

which is a generalisation of Eq. (46). This method can as well be seen as a generalisation of the adaptative inflation used by Penny and Miyoshi (2016). For their adaptative inflation, Penny and Miyoshi (2016) only computed the diagonal of the matrix $\mathbf{X}$ and fixed the bandwidth parameter $\hat{h}$ to 1. Our approach yields lowest RMSEs in all tested cases, which is most probably due to the tuning of the bandwidth parameter $\hat{h}$.

### 5.8.3 Numerical complexity and asymptotic limit

The coloured regularisation step has complexity $\mathcal{O}\left(N_x N_e^2\right)$. It is slightly more costly than using the white noise regularisation step due to the matrix product Eq. (49).

The exact LPF is recovered in the limit $\hat{h} \to 0$.

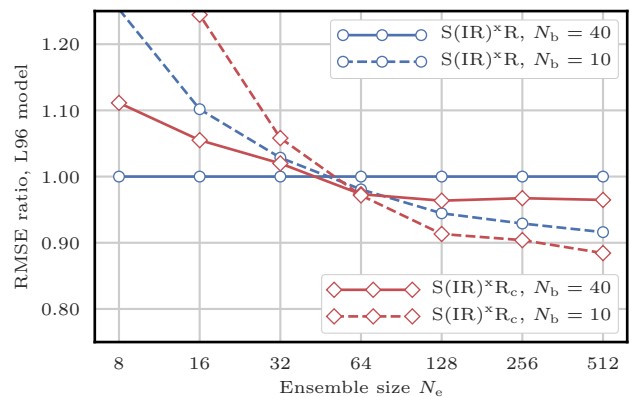

**Figure 11.** RMSE as a function of the ensemble size $N_e$ for the S(IR)$^x$R and the S(IR)$^x$R$_c$ algorithms with respectively $N_b = 40$, 10 blocks of size 1, 4 grid points. The scores are displayed in units of the RMSE of the S(IR)$^x$R algorithm with $N_b = 40$ blocks of size 1 grid point.

### 5.8.4 Illustrations

We experiment with the S(IR)$^x$R$_c$ algorithm, in which the regularisation jitter is colourised as described by Eqs. (49) and (50). In this algorithm, the parameter $s$ (regularisation jitter standard deviation) is replaced by the bandwidth parameter $\hat{h}$, hereafter simply called regularisation jitter. The evolution of the RMSE as a function of $\hat{h}$ for the S(IR)$^x$R$_c$ algorithm (not shown here) is very similar to the evolution of the RMSE as a function of $s$ for the S(IR)$^x$R algorithm. In the following, when using the coloured regularisation jitter method, $\hat{h}$ is systematically tuned to yield the lowest RMSE score.

Figure 11 shows the evolution of the RMSE as a function of the ensemble size $N_e$ for the S(IR)$^x$R and the S(IR)$^x$R$_c$ algorithms. These two variants only differ by the regularisation method. The S(IR)$^x$R algorithm uses white regularisation jitter while the S(IR)$^x$R$_c$ algorithm uses coloured regularisation jitter. For small ensembles, the S(IR)$^x$R$_c$ algorithm yields higher RMSEs, whereas it shows slightly better RMSEs for larger ensembles. Depending on the block size, the transition between both regimes happens around $N_e = 32$ to 64 particles. The higher RMSEs of the S(IR)$^x$R$_c$ algorithm for small ensembles

may have two potential explanations. First, even if the L96 model in its standard configuration is characterised by short-range correlations, the covariance matrix $\Sigma$ is a high-dimensional object that is poorly represented with a weighted ensemble. Second, the analysis distribution for small ensemble may be too different from a Gaussian for the coloured regularisation jitter method to yield better results even though in this mildly nonlinear configuration the densities are close to Gaussian.

### 5.9 Applying a smoothing by weights

In this section, we look for the potential benefits of adding a smoothing by weights step as presented in Sect. 4.4.1, by testing the S(IR)$^x$S$_{PM}$R and the S(IR)$^x$S$_{PM}$R$_c$ algorithms. These algorithms only differ from the S(IR)$^x$R and the S(IR)$^x$R$_c$ algorithms by the fact that they add a smoothing by weights step as specified in Algorithm 2.

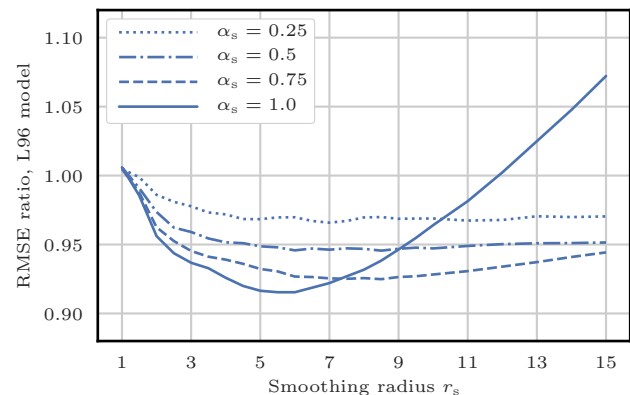

**Figure 12.** RMSE as a function of the smoothing radius $r_s$ for the S(IR)$^x$S$_{PM}$R algorithms with $N_e = 16$ particles and $N_b = 40$ blocks of size 1 grid point for several values of the smoothing strength $\alpha_s$. The scores are displayed in units of the RMSE of the S(IR)$^x$R algorithm with $N_e = 16$ particles and $N_b = 40$ blocks of size 1 grid point.

Alongside with the smoothing by weights step come two additional tuning parameters: the smoothing strength $\alpha_s$ and the smoothing radius $r_s$. We first investigate the influence of theses parameters. Figure 12 shows the evolution of the RMSE as a
function of the smoothing radius $r_s$ for the S(IR)$^x$S$_{PM}$R with $N_e = 10$ particles, $N_b = 40$ blocks of size 1 grid point for several values of the smoothing strength $\alpha_s$. As before, the localisation radius $r$ and the regularisation jitter $s$ are optimally tuned.

At a fixed smoothing strength $\alpha_s > 0$, starting from $r_s = 1$ grid point (no smoothing), the RMSE decreases when $r_s$ increases. It reaches a minimum and then increases again. In this example, the optimal smoothing radius $r_s$ lies between 5 and 6 grid points for a smoothing strength $\alpha_s = 1$, with corresponding optimal localisation radius $r$ between 2 and 3 grid points and
optimal regularisation jitter $s$ around $0.45$ (not shown here). For comparison, the optimal tuning parameters for the S(IR)$^x$R algorithm in the same configuration were $r$ between 4 and 5 grid points and $s$ around $0.2$.

Based on extensive tests of the S(IR)$^x$S$_{PM}$R and the S(IR)$^x$S$_{PM}$R$_c$ algorithms with $N_e$ ranging from 8 to 128 particles (not shown here), we conclude that:

– in general $\alpha_s = 1$ is optimal, or at least only slightly suboptimal;

– optimal values for $r$ and $s$ are larger with the smoothing by weights step than without it;

– optimal values for $r$ and $r_s$ are not related and must be tuned separately.

In the following, when using the smoothing by weights, we take $\alpha_s = 1$ and $r_s$ is tuned to yield the lowest RMSE score — alongside with the tuning of the localisation radius $r$ and the regularisation jitter $s$ or $\hat{h}$. Figure 13 shows the evolution of the RMSE as a function of the ensemble size $N_e$ for the S(IR)$^x$S$_{PM}$R and the S(IR)$^x$S$_{PM}$R$_c$ algorithms. The S(IR)$^x$S$_{PM}$R
algorithm yields systematically lower RMSEs than the standard S(IR)$^x$R. However, as the ensemble size $N_e$ grows, the gain in RMSE score becomes very small. With $N_e = 512$ particles, there is almost no difference between both scores. In this case, the optimal smoothing radius $r_s$ is around 5 grid points, much smaller than the optimal localisation radius $r$ around 15 grid points,

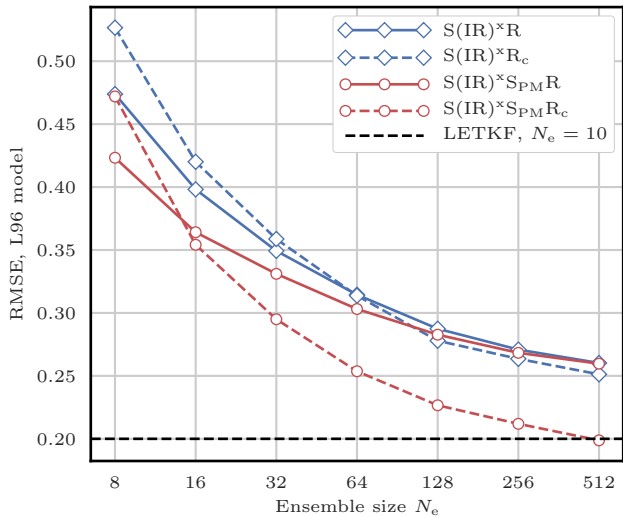

**Figure 13.** RMSE as a function of the ensemble size $N_e$ for S(IR)$^x$R, the S(IR)$^x$R$_c$, the S(IR)$^x$S$_{PM}$R and the S(IR)$^x$S$_{PM}$R algorithms.

such that the smoothing by weights step does not modify much the analysis ensemble. The S(IR)$^x$S$_{PM}$R$_c$ algorithm also yields lower RMSEs than the S(IR)$^x$R$_c$ algorithm. Yet, in this case, the gain in RMSE is still significant for large ensembles and with
$N_e = 512$ particles, the RMSEs are even comparable to those of the EnKF.

From these results, we conclude that the smoothing by weights step is an efficient way of mitigating the unphysical discontinuities that were introduced when assembling the locally updated particles, especially when combined with the coloured noise regularisation jitter method.

### 5.10    Using optimal transport in ensemble space

In this section, we evaluate the efficiency of using the optimal transport in ensemble space as a way to mitigate the unphysical discontinuities of the local resampling step by experimenting the S(IT$_R$)$^x$R and the S(IT$_R$)$^x$R$_c$ algorithms. These algorithms only differ from the S(IR)$^x$R and the S(IR)$^x$R$_c$ algorithms by the fact that they use optimal ensemble coupling for the local resampling as described by Algorithm 3.

For each block, the local linear transformation is computed by solving the minimisation problem Eq. (33), which can be
seen as a particular case of the minimum–cost flow problem. We choose to compute its numerical solution using the network simplex algorithm implemented by the graph library LEMON (Dezső et al., 2011). As described in Sect.4.4.3, this method is characterised by an additional tuning parameter: the distance radius $r_d$. We have investigated the influence of the parameters $N_b$ and $r_d$ by performing extensive tests of the S(IT$_R$)$^x$R and the S(IT$_R$)$^x$R$_c$ algorithms with $N_e$ ranging from $8$ to $128$ particles (not shown here) and draw the following conclusions.

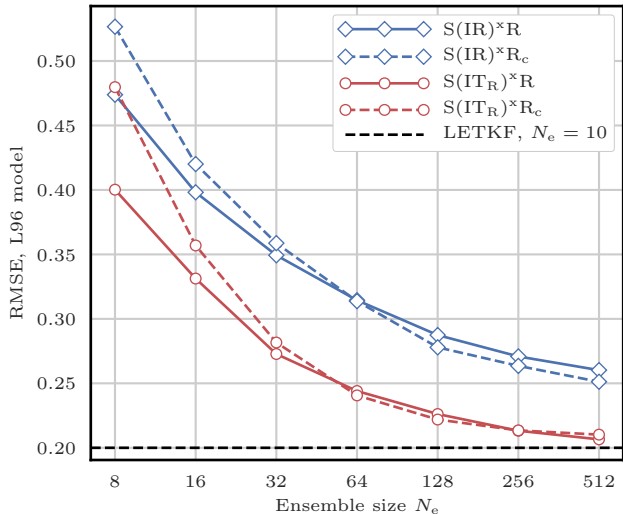

**Figure 14.** RMSE as a function of the ensemble size $N_e$ for the $S(IR)^x R$, the $S(IR)^x R_c$, the $S(IT_R)^x R$ and the $S(IT_R)^x R_c$ algorithms.

Optimal values for the distance radius $r_d$ are much smaller than the localisation radius, most of the time even smaller than 2 grid points. Using $r_d = 1$ grid point yields RMSEs that are only very slightly suboptimal. Moreover, in all situations using $N_b = 20$ blocks of size 2 grid points systematically yields higher RMSEs than using $N_b = 40$ blocks of size 1 grid point.

In the following, when using the optimal ensemble coupling algorithm, we take $r_d = 1$ grid point and $N_b = 40$ blocks of size 1 grid point. Figure 14 shows the evolution of the RMSE as a function of the ensemble size $N_e$ for the $S(IT_R)^x R$ and the $S(IT_R)^x R_c$ algorithms. Using optimal ensemble coupling for the local resampling step always yields significantly lower RMSEs than using the SU sampling algorithm. Yet in this case, using the coloured noise regularisation jitter method does not improve the RMSEs for very large ensembles.

We have also performed extensive tests with $N_e$ ranging from 8 to 128 particles on the $S(IT_R)^x S_{PM} R$ and the $S(IT_R)^x S_{PM} R_c$ algorithms in which the optimal ensemble coupling resampling method is combined with the smoothing by weights method (not shown here). Our implementations of these algorithms are numerically more costly. For small ensembles ($N_e \leq 32$ particles), the RMSEs of the $S(IT_R)^x S_{PM} R$ and the $S(IT_R)^x S_{PM} R_c$ algorithms are barely smaller than those of the $S(IT_R)^x R$ and the $S(IT_R)^x R_c$ algorithms. With larger ensembles, we could not find a configuration where using the smoothing by weights yields better RMSEs.

The facts that neither the use of larger blocks, nor the smoothing by weights does significantly improve the RMSE score when using optimal ensemble coupling indicate that this local resampling method is indeed an efficient way of mitigating the unphysical discontinuities inherent to assembling the locally updated particles.

## 5.11 Using continuous optimal transport

In this section, we test the efficiency of using the optimal transport in state space as a way to mitigate the unphysical disconti-nuities of the local resampling step by experimenting the $S(IT_s)^xR$ and the $S(IT_s)^xR_c$ algorithms. These algorithms only differ from the $S(IR)^xR$ and the $S(IR)^xR_c$ algorithms by the fact that they use anamorphosis for the local resampling as described by Algorithm 4.

As mentioned in Sect. 4.4.4, the local resampling algorithm based on anamorphosis uses blocks of size 1 grid point. Hence, when using the $S(IT_s)^xR$ and the $S(IT_s)^xR_c$ algorithms, we take $N_b = 40$ blocks of size 1 grid point. The definition of the state transformation map $T$ is based on the prior and corrected densities given by Eqs. (39) and (40) using the Student's t-distribution with two degrees of freedom for the regularisation kernel $K$. It is characterised by an additional tuning parameter: $h$, hereafter called *regularisation bandwidth* — different from the regularisation jitter $\hat{h}$. We have investigated the influence of the regularisation bandwidth $h$ by performing extensive tests of the $S(IT_s)^xR$ and the $S(IT_s)^xR_c$ algorithms with $N_e$ ranging from 8 to 128 particles (not shown here). For small ensembles ($N_e \leq 16$ particles), optimal values for $h$ lie between 2 and 3, the RMSE score obtained with $h = 1$ being very slightly suboptimal. For larger ensembles, we did not find any significant difference between $h = 1$ and larger values.

In the following, when using the anamorphosis resampling algorithm, we take the standard value $h = 1$. Figure 15 shows the evolution of the RMSE as a function of the ensemble size $N_e$ for the $S(IT_s)^xR$ and the $S(IT_s)^xR_c$ algorithms. These algorithms yield RMSEs even lower than the algorithms using optimal ensemble coupling. However in this case, using the coloured noise regularisation jitter method always yields significantly higher RMSEs than using the white noise regularisation method. It is probably a consequence of the fact that some coloured regularisation is already introduced in the nonlinear transformation process through the kernel representation of the densities with Eqs. (39) and (40). It may also be a consequence of the fact that the algorithms using anamorphosis for the local resampling step cannot be written as a local LET algorithm, contrary to the algorithms using the SU sampling or the optimal ensemble coupling algorithms.

We have also performed extensive tests with $N_e$ ranging from 8 to 128 particles on the $S(IT_s)^xS_{PM}R$ algorithm, in which the anamorphosis resampling method is combined with the smoothing by weights method (not shown here). As for the $S(IT_R)^xS_{PM}R$ and the $S(IT_R)^xS_{PM}R_c$ algorithms, our implementation is significantly numerically more costly and adding the smoothing by weights step only yields minor RMSE improvements.

These latter remarks, alongside with significantly lower RMSE for the $S(IT_s)^xR$ algorithm than for the $S(IR)^xR$ indicate that the local resampling method based on anamorphosis is, as well as the method based on optimal ensemble coupling, an efficient way of mitigating the unphysical discontinuities inherent to assembling the locally updated particles.

## 5.12 Summary

To summarise, Fig. 16 shows the evolution of the RMSE as a function of the ensemble size $N_e$ for the main LPF$^x$s tested in this section. For small ensembles ($N_e \leq 32$ particles), the algorithms using OT-based resampling methods clearly yield lower RMSEs than the other algorithms. For large ensemble ($N_e \geq 128$ particles), combining the smoothing by weights with the

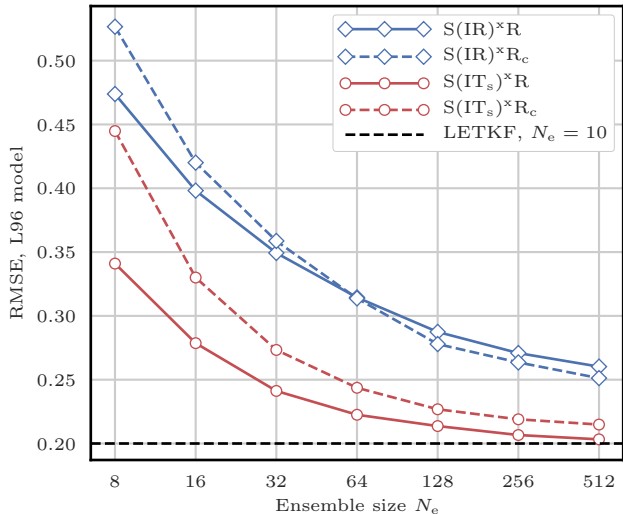

**Figure 15.** RMSE as a function of the ensemble size $N_e$ for the S(IR)$^x$R, the S(IR)$^x$R$_c$, the S(IT$_s$)$^x$R and the S(IT$_s$)$^x$R$_c$ algorithms.

coloured noise regularisation jitter methods yields equally good scores as the algorithms using OT. For $N_e = 512$ particles (the largest ensemble size tested with the L96 model), the best RMSE scores obtained with LPF$^x$s become comparable to those of the EnKF.

In this standard, mildly nonlinear configuration where error distributions are close to Gaussian, the EnKF performs very well and the LPF$^x$ algorithms tested in this section do not clearly yield lower RMSE scores than the ETKF and the LETKF. There are several potential reasons for this. First, the ETKF and the LETKF rely on more information than the LPF$^x$s because they use Gaussian error distributions, which is a good approximation in this configuration. Second, the values of the optimal localisation radius $r$ for the LPF$^x$s are in most cases smaller than the value of the optimal localisation radius $r$ for the LETKF, because localisation has to counteract the curse of dimensionality. This means that, in this case, localisation introduces more bias in the PF than in the EnKF. Third, using a non-zero regularisation jitter is necessary to avoid the collapse of the LPF$^x$s without model error. This method introduces an additional bias in the LPF$^x$ analysis. In practice, we have found in this case that the values of the optimal regularisation jitter for the LPF$^x$s are rather large whereas the optimal inflation factor in the ETKF and the LETKF is small.

Note that our objective is not to design LPF algorithms that beat the EnKF in all situations, but rather to incrementally improve the PF. However, specific configurations in which the EnKF fails and the PF succeeds can easily be conceived by increasing nonlinearities. Such a configuration is studied in Appendix C.

As a complement to this RMSE test series, rank histograms for several LPFs are computed, reported and discussed in Appendix D.

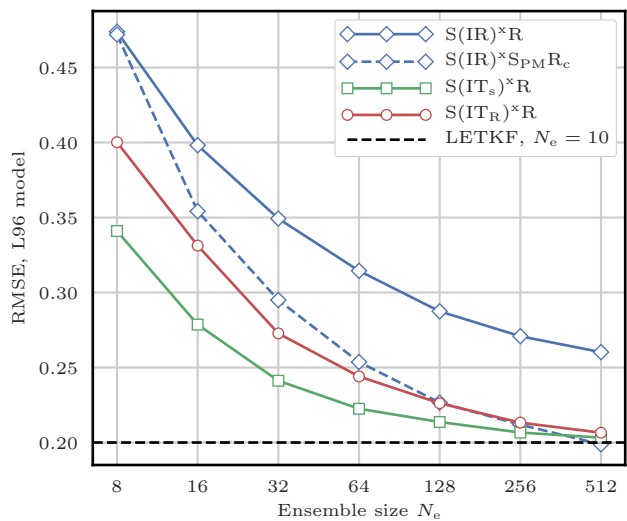

**Figure 16.** RMSE as a function of the ensemble size $N_e$ for the main LPF$^x$s tested in this section.

## 6 Numerical illustration of the LPF$^x$ algorithms with a barotropic vorticity model

### 6.1 Model specifications

In this section, we illustrate the performance of LPF$^x$s with twin simulations of the barotropic vorticity (BV) model in the coarse resolution (CR) configuration described in Appendix A4.1. Using this configuration yields a DA problem of size $N_x = 1024$ and $N_y = 256$. As mentioned in Appendix A4.1, the spatial resolution is enough to capture the dynamics of a few vortices and the model integration is not too expensive, such that we can perform extensive tests with small to moderate ensemble sizes.

As with the L96 model, the distance between the truth and the analysis is measured with the average analysis RMSE. The
935 runs are $9 \times 10^3 \Delta t$ long with an additional $10^3 \Delta t$ spin-up period, more than enough to ensure the convergence of the statistical indicators.

For the localisation, we use the underlying physical space with the Euclidean distance. The geometry of the blocks and domain are constructed as described by Fig. 2. Specifically, blocks are rectangles and local domains are disks, with the difference that the doubly periodic boundary conditions are taken into account.

### 6.2 Scores for the EnKF and the PF

As a reference, we first compute the RMSEs of the EnKF with this model. Figure 17 shows the evolution of the RMSE as a function of the ensemble size $N_e$ for the ETKF and the LETKF. For each value of $N_e$, the inflation parameter and the localisation radius (only for the LETKF) are optimally tuned to yield the lowest RMSE.

The ETKF requires at least $N_e = 12$ ensemble members to avoid divergence. The best RMSEs are approximately 20 times
945 smaller than the observation standard deviation ($\sigma = 0.3$). Even with only $N_e = 8$ ensemble members, the LETKF yields

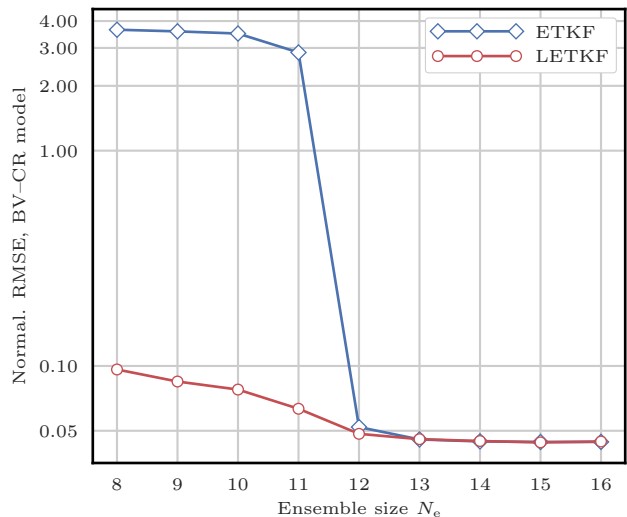

**Figure 17.** RMSE as a function of the ensemble size $N_e$ for the ETKF and the LETKF. The scores are displayed in units of the observation standard deviation $\sigma$.

RMSEs at least 10 times smaller than the observation standard deviation, showing that in this case localisation is working as expected. In this configuration, the observation sites are uniformly distributed over the spatial domain. This constrains the posterior pdfs to be close to Gaussian, which explains the success of the EnKF in this DA problem.

With $N_e \leq 1024$ particles, we could not find a combination of tuning parameters with which the bootstrap filter or the ETPF yield RMSEs significantly lower than 1. In the following figures related to this BV test series, we draw a baseline at $\sigma/20$, which is roughly the RMSE of the ETKF and the LETKF with $N_e = 12$ particles. Note that slightly lower RMSE scores can be achieved with larger ensembles.

### 6.3 Scores for the LPF$^x$ algorithms

In this section, we test the LPF$^x$ algorihtms with $N_e$ ranging from 8 to 128 particles. The nomenclature for the algorithms is the same as in Sect. 5. In particular, all algorithms tested in this Section are in the list reported in Table 2.

For each ensemble size $N_e$ we use similar parameter tuning methods as for the L96 model as follows:

– we take zero integration jitter: $q = 0$;

– the localisation radius $r$ is systematically tuned to yield the lowest RMSE score;

– the regularisation jitter $s$ (or $\hat{h}$ when using the coloured noise regularisation jitter method) is systematically tuned as well;

– for the algorithms using the SU sampling algorithm (i.e. the S(IR)$^x \ast \ast$ variants) we test four values for the number of blocks $N_b$, and we keep the best RMSE score:

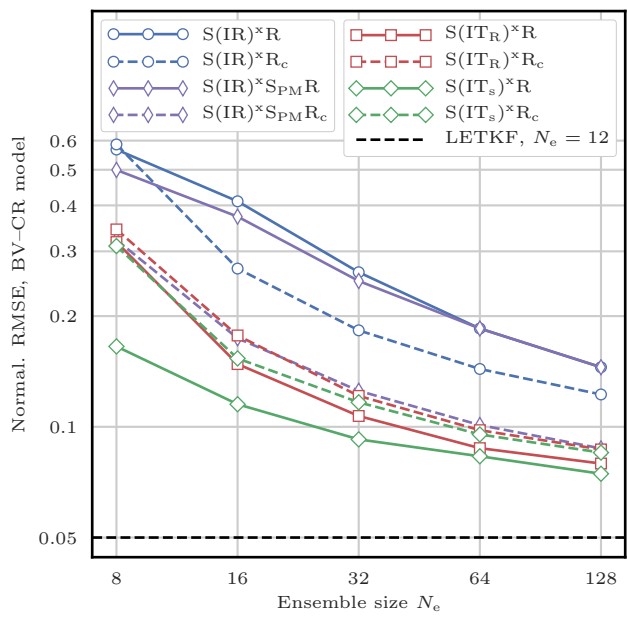

**Figure 18.** RMSE as a function of the ensemble size $N_e$ for the LPF$^x$s. The scores are displayed in units of the observation standard deviation $\sigma$.

- blocks of shape $1 \times 1$ grid point;
- 256 blocks of shape $2 \times 2$ grid points;
- 64 blocks of shape $4 \times 4$ grid points;
- 16 blocks of shape $8 \times 8$ grid points;

– for the algorithms using optimal ensemble coupling or anamorphosis (i.e. the S(IT$_*$)$^x$∗ variants) we only test blocks of shape $1 \times 1$ grid point;

– when using the smoothing by weights method, we take the smoothing strength $\alpha_s = 1$ and the smoothing radius $r_s$ is optimally tuned to yield the lowest RMSE score;

– when using the optimal ensemble coupling for the local resampling step, the distance radius $r_d$ is optimally tuned to yield the lowest RMSE score;

– when using the anamorphosis for the local resampling step, we take the regularisation bandwidth $h = 1$.

Figure 18 shows the evolution of the RMSE as a function of the ensemble size $N_e$ for the LPF$^x$s. Most of the conclusions drawn with the L96 model remain true with the BV model. The best RMSE scores are obtained with algorithms using OT-based resampling methods. Combining the smoothing by weights with the coloured noise regularisation jitter methods yields almost equally good scores as the algorithms using OT. Yet, some differences can be pointed out.

With such a large model, we expected the coloured noise regularisation jitter method to be much more effective than the white noise method because the colourisation reduces potential spatial discontinuities in the jitter. We observe indeed that the S(IR)$^x$R$_c$ and the S(IR)$^x$S$_{PM}$R$_c$ algorithms yield significantly lower RMSEs than the S(IR)$^x$R and the S(IR)$^x$S$_{PM}$R algorithms. Yet, the S(IT$_R$)$^x$R$_c$ and the S(IT$_s$)$^x$R$_c$ algorithms are clearly outperformed by both the S(IT$_R$)$^x$R and the S(IT$_s$)$^x$R algorithms in terms of RMSEs. This suggests that there is room for improvement in the design of regularisation jitter methods for PF algorithms.

Due to relatively high computational times, we restricted our study to reasonable ensemble sizes, $N_e \leq 128$ particles. In this configuration, the RMSE scores of LPF$^x$s are not yet comparable with those of the EnKF (see Fig. 18).

Finally, it should be noted that for the S(IT$_R$)$^x$R and the S(IT$_R$)$^x$R$_c$ algorithms with $N_e \geq 32$ particles, optimal values for the distance radius $r_d$ lie between 3 and 6 grid points (not shown here) contrary to the results obtained with the L96 model, for which $r_d = 1$ grid point could be considered optimal. More generally for all LPF$^x$s, the optimal values for the localisation radius $r$ (not shown here) are significantly larger (in number of grid points) for the BV model than for the L96 model.

# 7    Sequential–observation localisation for particle filters

In the SBD localisation formalism, each block of grid points is updated using the local domain of observation sites that may influence these grid points. In the sequential–observation (SO) localisation formalism, we use a different approach. Observations are assimilated sequentially and assimilating the observation at a site should only update nearby grid points. LPF algorithms using the SO localisation formalism will be called LPF$^y$ algorithms[3].

In this section, we set $q \in \{1 \dots N_y\}$ and we describe how to assimilate the observation $y_q$. In Sect. 7.1, we introduce the state space partitioning. The resulting decompositions of the conditional density are discussed in Sect. 7.2. Finally, practical algorithms using these principles are derived in Sects. 7.3 and 7.4.

These algorithms are designed to assimilate one observation at a time. Hence, a full assimilation cycle requires $N_y$ sequential iterations of these algorithms, during which the ensemble is gradually updated: the updated ensemble after assimilating $y_q$ will be the prior ensemble to assimilate $y_{q+1}$.

## 7.1    Partitioning the state space

Following Robert and Künsch (2017) the state space $\mathbb{R}^{N_x}$ is divided into three regions:

1. the first region $U$ covers all grid points that directly influence $y_q$ — if $\mathcal{H}$ is linear, it is all columns of $\mathcal{H}$ that have non-zero entries on row $q$;

2. the second region $V$ gathers all grid points that are deemed correlated to those in $U$;

3. the third region $W$ contains all remaining grid points.

---

[3]The y exponent emphasises the fact that we perform one analysis per observation.

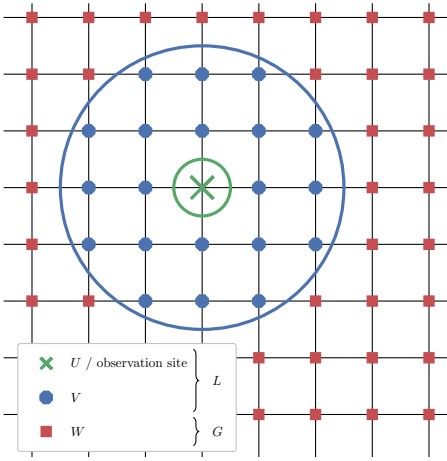

**Figure 19.** Example of the $UVW$ partition for a two-dimensional space. The site of observation $y_q$ lies in the middle. The local regions $U$ and $V$ are circumscribed by the thick green and blue circles and respectively contain 1 and 20 grid points. The global region $W$ contains all remaining grid points. In the case of the $LG$ partition, the local region $L$ gathers all 21 grid points in $U$ and $V$.

The meaning of "correlated" is to be understood as a prior hypothesis, where we define a valid tapering matrix $\mathbf{C}$ that represents the decay of correlations. Non-zero elements of $\mathbf{C}$ should be located near the main diagonal and reflect the intensity of the correlation. A popular choice for $\mathbf{C}$ is the one obtained using the Gaspari–Cohn function $G$:

$$[\mathbf{C}]_{m,n} = G\left(\frac{d_{m,n}}{r}\right), \tag{52}$$

where $d_{m,n}$ is the distance between the $m$-th and $n$-th grid points and $r$ is the localisation radius, a free parameter similar to the localisation radius defined in the SBD localisation formalism (see Sect. 4.2.2).

The $UVW$ partition of the state space is a generalisation of the original $LG$ partition introduced by Bengtsson et al. (2003) in which $U$ and $V$ are gathered into one region $L$, the *local* domain of $y_q$, and $W$ is called $G$ (for *global*). Figure 19 illustrates this $UVW$ partition. We emphasise that both the LG and the UVW state partitions depends on the site of observation $y_q$. They are fundamentally different from the (local state) block decomposition of Sect. 4.2.1 and therefore they shall simply be called "partition" to avoid confusion.

## 7.2 The conditional density

For any region $A$ of the physical space, let $\boldsymbol{x}_A$ be the restriction of vector $\boldsymbol{x}$ to $A$, i.e. the state variables of $\boldsymbol{x}$ whose grid points are located within $A$.

### 7.2.1 With the $LG$ partition

Without loss of generality, the conditional density is decomposed into:

$$p(\boldsymbol{x}|y_q) = p(\boldsymbol{x}_L, \boldsymbol{x}_G|y_q) = p(\boldsymbol{x}_L|\boldsymbol{x}_G, y_q)\,p(\boldsymbol{x}_G|y_q). \tag{53}$$

In a localisation context, it seems reasonable to assume that $\boldsymbol{x}_G$ and $y_q$ are independent, that is:

$$p\left(\boldsymbol{x}_G|y_q\right) = p\left(\boldsymbol{x}_G\right), \tag{54}$$

and the conditional pdf of the $L$ region can be written:

$$p\left(\boldsymbol{x}_L|\boldsymbol{x}_G, y_q\right) = \frac{p\left(y_q|\boldsymbol{x}_G, \boldsymbol{x}_L\right) p\left(\boldsymbol{x}_G, \boldsymbol{x}_L\right)}{p\left(\boldsymbol{x}_G, y_q\right)}, \tag{55}$$

$$= \frac{p\left(y_q|\boldsymbol{x}_L\right) p\left(\boldsymbol{x}_G, \boldsymbol{x}_L\right)}{p\left(\boldsymbol{x}_G, y_q\right)}. \tag{56}$$

This yields an assimilation method for $y_q$ described by Algorithm 5.

---

**Algorithm 5** Single analysis step for a generic LPF$^{\mathrm{y}}$ algorithm using the $LG$ partition

---

**Require:** Prior ensemble $\boldsymbol{x}^i$, $i = 1 \ldots N_{\mathrm{e}}$ and observation $y_q$

1: Build the $LG$ partition as described in Sect. 7.1

2: **for** $i = 1$ **to** $N_{\mathrm{e}}$ **do**

3:      Do not update $\boldsymbol{x}_G^i$

4:      Update $\boldsymbol{x}_L^i$ conditionally to $y_q$ and $\boldsymbol{x}_G^i$ as stated by Eq. (56)

5: **end for**

6: **return** Updated ensemble $\boldsymbol{x}^i$, $i = 1 \ldots N_{\mathrm{e}}$

---

### 7.2.2   With the $UVW$ partition

With the $UVW$ partition, the conditional density is factored as

$$p\left(\boldsymbol{x}|y_q\right) = p\left(\boldsymbol{x}_U, \boldsymbol{x}_V, \boldsymbol{x}_W|y_q\right), \tag{57}$$

$$= \frac{p\left(\boldsymbol{x}_U, \boldsymbol{x}_V, \boldsymbol{x}_W, y_q\right)}{p\left(y_q\right)}, \tag{58}$$

$$= \frac{p\left(y_q|\boldsymbol{x}\right) p\left(\boldsymbol{x}_V|\boldsymbol{x}_U, \boldsymbol{x}_W\right) p\left(\boldsymbol{x}_U, \boldsymbol{x}_W\right)}{p\left(y_q\right)}, \tag{59}$$

$$= \frac{p\left(y_q|\boldsymbol{x}_U\right) p\left(\boldsymbol{x}_V|\boldsymbol{x}_U, \boldsymbol{x}_W\right) p\left(\boldsymbol{x}_U, \boldsymbol{x}_W\right)}{p\left(y_q\right)}. \tag{60}$$

If one assumes that the $U$ and $W$ regions are not only uncorrelated but also independent, then one can make the additional factorisation:

$$p\left(\boldsymbol{x}_U, \boldsymbol{x}_W\right) = p\left(\boldsymbol{x}_U\right) p\left(\boldsymbol{x}_W\right). \tag{61}$$

Finally, the conditional density is

$$p\left(\boldsymbol{x}|y_q\right) = p\left(\boldsymbol{x}_U|y_q\right) p\left(\boldsymbol{x}_V|\boldsymbol{x}_U, \boldsymbol{x}_W\right) p\left(\boldsymbol{x}_W\right). \tag{62}$$

The assimilation method for $y_q$ is now described by Algorithm 6.

**Algorithm 6** Single analysis step for a generic LPF$^y$ algorithm using the $UVW$ partition

---

**Require:** Prior ensemble $\boldsymbol{x}^i$, $i = 1 \ldots N_e$ and observation $y_q$

1: Build the $UVW$ partition as described in Sect. 7.1
2: **for** $i = 1$ **to** $N_e$ **do**
3:     Do not update $\boldsymbol{x}_W^i$
4:     Update $\boldsymbol{x}_U^i$ conditionally to $y_q$
5:     Update $\boldsymbol{x}_V^i$ conditionally to $\boldsymbol{x}_W^i$ and (the updated) $\boldsymbol{x}_U^i$
6: **end for**
7: **return** Updated ensemble $\boldsymbol{x}^i$, $i = 1 \ldots N_e$

---

### 7.2.3 The partition and the particle filter

So far, the SO formalism looks elegant. The resulting assimilation schemes avoid the discontinuity issue inherent to the SBD formalism by using conditional updates of the ensemble.

However, this kind of update seems hopeless in a pure PF context. Indeed the factors $p(\boldsymbol{x}_G, \boldsymbol{x}_L)$ and $p(\boldsymbol{x}_V | \boldsymbol{x}_U, \boldsymbol{x}_W)$ in Eqs. (56) and (60) will be non-zero only if the updated particles are copies of the prior particles, which spoils the entire purpose of localising the assimilation. Hence potential solutions need to make approximations of the conditional density.

### 7.2.4 The multivariate rank histogram filter

Similar principles were used to design the multivariate rank histogram filter (MRHF) of Metref et al. (2014), with the main difference that the state space is entirely partitioned as follows. Assuming that $y_q$ only depends on $x_1$, the conditional density can be written:

$$p(\boldsymbol{x} | y_q) = p(x_1 | y_q) \, p(x_2 | x_1) \ldots p(x_{n+1} | x_n \ldots x_1) \ldots \tag{63}$$

In the MRHF analysis, the state variables are updated sequentially according to the conditional density $p(x_{n+1} | x_n \ldots x_1)$. Zero factors in $p(x_{n+1} | x_n \ldots x_1)$ are avoided by using a kernel representation *for the conditioning* on $x_n \ldots x_1$, in a similar way as in Eqs. (39) and (40) with top hat functions for the regularisation kernel $K$. The resulting one-dimensional density along $x_{n+1}$ is represented using histograms and the ensemble members are transformed using the same anamorphosis procedure as the one described in Sect. 4.4.4.

The MRHF could be used as a potential implementation of the SO localisation formalism. However, assimilating one observation requires the computation of $N_x$ different anamorphosis transformations.

### 7.2.5 Implementing the SO formalism

In the following sections, we introduce two different algorithms that implement the SO formalism (with the $UVW$ partition) to assimilate one observation. Both algorithms are based on an "importance, resampling, propagation" scheme as follows. Global

unnormalised importance weights are first computed as

$$w^i = p\left(y_q | \boldsymbol{x}^i\right). \tag{64}$$

Using these weights, we compute a resampling in the $U$ region (essentially at the observation site). The update is then propagated to the $V$ region using a dedicated propagation algorithm.

## 7.3 A hybrid algorithm for the propagation

The first algorithm that we introduce to implement the SO formalism using the "importance, resampling, propagation" scheme is the LPF of Poterjoy (2016) (hereafter Poterjoy's LPF). In this algorithm, the update is propagated using a hybrid scheme that mixes a (global) PF update with the prior ensemble.

### 7.3.1 Step 1: importance and resampling

Using the global unnormalised importance weights Eq. (64), we compute a resampling map $\phi$, using for example the SU sampling algorithm.

### 7.3.2 Step 2: update and propagation

The resampling map $\phi$ is used to update the ensemble in the $U$ region and the update is propagated to all grid points as

$$x_n^i = \overline{x}_n + \omega_n^{\mathrm{a}}\left(x_n^{\phi(i)} - \overline{x}_n\right) + \omega_n^{\mathrm{f}}\left(x_n^i - \overline{x}_n\right), \tag{65}$$

where $\overline{x}_n$ is the ensemble mean at the $n$-th grid point, $\omega^{\mathrm{a}}$ is the weight of the PF update and $\omega^{\mathrm{f}}$ is the weight of the prior. If the resampling algorithm is adjustment-minimising, the number of updates that need to be propagated is minimal. Finally, the $\omega^*$ (either $\omega^{\mathrm{f}}$ or $\omega^{\mathrm{a}}$) weights are chosen such that the updated ensemble yields correct statistics at the first- and second-orders.

At the observation site, $\omega^{\mathrm{a}} = 1$ and $\omega^{\mathrm{f}} = 0$, such that the update on the $U$ region is the PF update and is Bayesian. Far from the observation site, $\omega^{\mathrm{a}} = 0$ and $\omega^{\mathrm{f}} = 1$, such that there is no update on the $W$ region. Hence, the $i$-th updated particle is a composite particle between the $i$-th prior particle (in $W$) and the hypothetical $i$-th updated particle (in $U$) that would be obtained with a PF update. In between (in $V$) discontinuities are avoided by using a smooth transition for the $\omega^*$ weights, which involves the localisation radius $r$. A single analysis step according to Poterjoy's LPF is summarised by Algorithm 7.

The formulas for the $\omega^*$ weights are summarised in Appendix B. Their detailed derivation can be found in Poterjoy (2016), where $\omega^{\mathrm{a}}$ and $\omega^{\mathrm{f}}$ are called $r_1$ and $r_2$. Poterjoy (2016) included in his algorithm a weight inflation parameter that can be ignored to understand how the algorithm works. Moreover, the $N_{\mathrm{y}}$ sequential assimilations are followed by an optional KDDM step. As explained in Sect. 4.4.4, we found the KDDM step to be better suited for the local resampling step of LPF$^{\mathrm{x}}$ algorithms. Therefore, we have not included it in our presentation of Poterjoy's LPF.

**Algorithm 7** Single analysis step for Poterjoy's LPF algorithm

---

**Require:** Prior ensemble $\boldsymbol{x}^i$, $i = 1 \ldots N_e$ and observation $y_q$

1: Compute the analysis weights using Eq. (64)

2: Compute the resampling map $\phi$

3: **for** $n = 1$ **to** $N_x$ **do**

4:     Compute the weights $\omega_n^f$ and $\omega_n^a$

5:     **for** $i = 1$ **to** $N_e$ **do**

6:         Update $x_n^i$ using Eq. (65)

7:     **end for**

8: **end for**

9: **return** Updated ensemble $\boldsymbol{x}^i$, $i = 1 \ldots N_e$

---

## 7.4 A second-order algorithm for the propagation

The second algorithm that we introduce to implement the SO formalism using the "importance, resampling, propagation" scheme is based on the ensemble Kalman particle filter (EnKPF), a Gaussian mixture hybrid ensemble filter designed by Robert and Künsch (2017). In this algorithm, the updated is propagated using second-order moments.

### 7.4.1 Preliminary: the covariance matrix

Since the update is propagated using second-order moments, one first needs to compute the covariance matrix of the prior ensemble:

$$\boldsymbol{\Sigma}^f = \mathrm{cov}\left(\boldsymbol{x}\right). \tag{66}$$

In a localisation context, it seems reasonable to use a tapered representation of the covariance. Therefore, we use the covariance matrix $\boldsymbol{\Sigma}$ defined by

$$\boldsymbol{\Sigma} = \mathbf{C} \circ \boldsymbol{\Sigma}^f, \tag{67}$$

where $\mathbf{C}$ is the valid tapering matrix mentioned in section 7.1 (defined using the localisation radius $r$) and $\circ$ means the Schur product for matrices.

### 7.4.2 Step 1: importance and resampling

Using the global unnormalised importance weights Eq. (64), we resample the ensemble in the $U$ region and compute the update $\Delta \boldsymbol{x}_U^i$. For this resampling step, any resampling algorithm can be used:

- an adjustment-minimising resampling algorithm can be used to minimise the number of updates $\Delta \boldsymbol{x}_U^i$ that need to be propagated;

– the resampling algorithms based on OT in ensemble space or in state space, as derived in Sects. 4.4.3 and 4.4.4 can be used; as for the LPF$^\mathrm{x}$ methods, we expect them to create strong correlations between the prior and the updated ensembles.

### 7.4.3   Step 2: propagation

For each particle the update of $V$, $\Delta\boldsymbol{x}_V^i$, depends on the update on $U$, $\Delta\boldsymbol{x}_U^i$, through the linear regression:

$$\Delta\boldsymbol{x}_V^i = \boldsymbol{\Sigma}_{VU}\boldsymbol{\Sigma}_U^{-1}\Delta\boldsymbol{x}_U^i, \tag{68}$$

where $\boldsymbol{\Sigma}_{VU}$ and $\boldsymbol{\Sigma}_U$ are submatrices of $\boldsymbol{\Sigma}$. The full derivation of Eq. (68) is available in Robert and Künsch (2017). Note that $\boldsymbol{\Sigma}$ is a $N_\mathrm{x} \times N_\mathrm{x}$ matrix but only the submatrices $\boldsymbol{\Sigma}_{VU}$ and $\boldsymbol{\Sigma}_U$ need to be computed.

A single analysis step according to this second-order algorithm is summarised by Algorithm 8 in a generic context, with any resampling algorithm.

---

**Algorithm 8** Single analysis step for a generic LPF$^\mathrm{y}$ using the second-order propagation algorithm

---

**Require:** Prior ensemble $\boldsymbol{x}^i$, $i = 1\ldots N_\mathrm{e}$, observation $y_q$

1: Build the $UVW$ partition as described in Sect. 7.1

2: Compute the prior covariance submatrices $\boldsymbol{\Sigma}_{VU}$ and $\boldsymbol{\Sigma}_U$

3: Compute the analysis weights using Eq. (64)

4: Resample the ensemble on region $U$

5: Compute the associated updates $\Delta\boldsymbol{x}_U^i, i = 1\ldots N_\mathrm{e}$

6: **for** $i = 1$ **to** $N_\mathrm{e}$ **do**

7:    Compute the update $\Delta\boldsymbol{x}_V^i$ using Eq. (68)

8:    Apply the update $\Delta\boldsymbol{x}_V^i$ on region $V$

9: **end for**

10: **return** Updated ensemble $\boldsymbol{x}^i$, $i = 1\ldots N_\mathrm{e}$

---

### 7.5   Summary for the LPF$^\mathrm{y}$ algorithms

### 7.5.1   Highlights

In this section, we have introduced a generic SO localisation framework, which we have used to define the LPF$^\mathrm{y}$s, our second category of LPF methods. We have presented two algorithms, both based on an "importance, resampling, propagation" scheme:

1. The first algorithm is the LPF of Poterjoy (2016). It uses a hybrid scheme between a (global) PF update and the prior ensemble to propagate the update from the observation site to all grid points.

2. The second algorithm was inspired by the EnKPF of Robert and Künsch (2017). It uses tapered second-order moments to propagate the update.

Both algorithms derived in this section include some spatial smoothness in the construction of the updated particles. In Poterjoy's LPF, the smoothness comes from the definition of the $\omega^*$ weights. In the second-order propagation scheme, the smoothness comes from the prior correlations. Therefore, we expect the unphysical discontinuities to be less critical with these algorithms than with the LPF$^x$ algorithms, which is why the partition was introduced in the first place.

### 7.5.2 Numerical complexity

Let $N_U$ and $N_V$ be the maximum number of grid points in $U$ and $V$ respectively and let $N_{UV} = N_U + N_V$. The complexity of assimilating one observation using Poterjoy's LPF is:

– $\mathcal{O}(N_e)$ to compute the analysis weights Eq. (64) and the resampling map $\phi$;

– $\mathcal{O}(N_e N_{UV})$ to compute the $\omega^*$ weights and to propagate the update to the $U$ and $V$ regions.

The complexity of assimilating one observation using the second-order propagation algorithm is the sum of the complexity of computing the update on the $U$ region, on the $V$ region and of applying these updates to the ensemble. The complexity of computing the update on the $U$ region is:

– $\mathcal{O}(N_e N_U)$ when using the adjustment-minimising SU sampling algorithm;

– $\mathcal{O}\left(N_e^2 N_x^\ell(r_d) + N_e^3 + N_e^2 N_U\right)$ when using the optimal ensemble coupling derived in Sect. 4.4.3 with a distance radius $r_d$;

– $\mathcal{O}(N_U N_e N_p)$ when using the anamorphosis derived in Sect. 4.4.4 with a fixed one-dimensional resolution of $N_p$ points.

Using Eq. (68), the complexity of computing the update on the $V$ region is:

– $\mathcal{O}\left(N_U^3\right)$ to compute $\boldsymbol{\Sigma}_U^{-1}$;

– $\mathcal{O}\left(N_e N_U^2 + N_e N_V N_U\right)$ to apply $\boldsymbol{\Sigma}_{VU}\boldsymbol{\Sigma}_U^{-1}$ to all $\Delta\boldsymbol{x}_U^i$, $i = 1 \ldots N_e$.

Finally, the complexity of applying the update on the $U$ and $V$ region is $\mathcal{O}(N_e N_{UV})$.

With LPF$^y$ algorithms, observations are assimilated sequentially, which means that these algorithms are to be applied $N_y$ times per assimilation cycle. This also means that the LPF$^y$ algorithms are by construction non parallel. This issue was discussed by Robert and Künsch (2017): some level of parallelisation could be introduced in the algorithms, but only between observation sites for which the $U$ and $V$ regions are disjoint. That is to say, one can assimilate in parallel the observation at several sites as long as their domain of influence (in which an update is needed) do not overlap. This would require a preliminary geometric step to determine in which order observation sites are to be assimilated. This step would need to be performed again whenever the localisation radius $r$ is changed. Moreover, when $r$ is large enough, all $U$ and $V$ regions may overlap, and parallelisation is not possible.

### 7.5.3 Asymptotic limit

By definition of the $\omega^*$ weights, the single analysis step for Poterjoy's LPF is equivalent to the analysis step of the (global) PF for a single observation in the limit $r \to \infty$. This is not the case for the algorithm based on the second-order propagation scheme. Indeed, using second-order moments to propagate the update introduces a bias in the analysis. On the other hand, second-order methods are in general less sensitive to the curse of dimensionality. Therefore, we expect the algorithm based on the second-order propagation scheme to be able to handle larger values for the localisation radius $r$ than the LPF$^x$s.

### 7.6 Gathering observation sites into blocks

The LPF$^y$s can be extended to the case where observation sites are compounded into small blocks as follows:

- the unnormalised importance weights Eq. (64) are modified such that they account for all sites inside the block;

- any distance that needs to be computed relative to the site of observation $y_q$ (for example for the $\omega^*$ weights for Poterjoy's LPF) is now computed relatively to the block center;

- in the algorithm based on the second-order propagation scheme, the $UVW$ partition is modified: the $U$ region has to cover all grid points that directly influence every site inside the block.

Gathering observation sites into blocks reduces the number of sequential assimilations from $N_y$ to the number of blocks, hence reducing the computational time per cycle. However, it introduces an additional bias in the analysis. Therefore, we do not use this method in the numerical examples of Sects. 8 and 9.

## 8 Numerical illustration of the LPF$^y$ algorithms

### 8.1 Experimental setup

In this section, we illustrate the performance of the LPF$^y$ algorithms using twin simulations with the L96 and the BV models. The model specifications for this test series are the same as for the LPF$^x$ test series: the L96 model is used in the standard configuration described in Appendix A3 and the BV model is used in the CR configuration described in Appendix A4.1. In a manner consistent with Sects. 5 and 6, the LPF$^y$ algorithms are named S(I$\alpha$P$_\beta$)$^y\gamma$ — sampling, importance, resampling, propagation, regularisation, the $y$ exponent meaning that steps in parentheses are performed locally for each observation — with the conventions detailed in Table 3. Table 4 lists all LPF$^y$ algorithms tested in this section and reports their characteristics according to the convention of Table 3.

#### 8.1.1 Regularisation jitter

For the same reasons as with LPF$^x$s, jittering the LPF$^y$s is necessary to avoid a fast collapse. As we eventually did for the LPF$^x$s, the model is not perturbed (no integration jitter) and regularisation noise is added at the end of each assimilation cycle,

**Table 3.** Nomenclature conventions for the $S(I\alpha P_\beta)^y \gamma$ algorithms. Capital letters refer to the main algorithmic ingredients: "I" for importance, "R" for resampling or regularisation, "T" for transport, "P" for propagation. Subscripts are used to distinguish the methods in two different ways. Lower case subscripts refer to explicit concepts used in the method: "s" for state space, "c" for color; while upper case subscripts refer to the work that inspired the method: "P" for Poterjoy (2016) and "RK" for Robert and Künsch (2017). For simplicity, some subscripts are omitted: "amsu" for adjustment-minimising stochastic universal and "w" for white.

| | |
|---|---|
| $\alpha$ | Local resampling algorithm |
| R | adjustment-minimising SU sampling algorithm |
| $T_R$ | optimal transport in ensemble space (Sect. 4.4.3) |
| $T_s$ | optimal transport in state space (Sect. 4.4.4) |
| $\beta$ | Propagation method |
| P | Poterjoy's LPF (Algorithm 7) |
| RK | Second-order propagation (Algorithm 8) |
| $\gamma$ | Regularisation method (Sects. 5.2 and 5.8) |
| R | white noise method |
| $R_c$ | coloured noise method |

**Table 4.** List of all LPF$^y$ algorithms tested in this article. For each algorithm, the main characteristics are reported with appropriate references.

| Algorithm | Resampling algorithm (Sect. 4.4) | Subsection | Propagation algorithm (Sects. 7.3 and 7.4) | | Regularisation method (Sects. 5.2 and 5.8) | |
|---|---|---|---|---|---|---|
| | | | Algorithm 7 [Poterjoy's LPF] | Algorithm 8 [Second-order] | Eq. (45) [white] | Eq. (49) [colour] |
| $S(IRP_P)^y R$ | adjustment-minimising SU sampling | 4.4.2 | ✓ | | ✓ | |
| $S(IRP_P)^y R_c$ | adjustment-minimising SU sampling | 4.4.2 | ✓ | | | ✓ |
| $S(IRP_{RK})^y R$ | adjustment-minimising SU sampling | 4.4.2 | | ✓ | ✓ | |
| $S(IRP_{RK})^y R_c$ | adjustment-minimising SU sampling | 4.4.2 | | ✓ | | ✓ |
| $S(IT_R P_{RK})^y R$ | optimal ensemble coupling | 4.4.3 | | ✓ | ✓ | |
| $S(IT_R P_{RK})^y R_c$ | optimal ensemble coupling | 4.4.3 | | ✓ | | ✓ |
| $S(IT_s P_{RK})^y R$ | anamorphosis | 4.4.4 | | ✓ | ✓ | |
| $S(IT_s P_{RK})^y R_c$ | anamorphosis | 4.4.4 | | ✓ | | ✓ |

either using the white noise method described by Eq. (45) or using the coloured noise method described in Sect. 5.8. With this latter method, the local weights required for the computation of the covariance matrix of the regularisation noise are computed with Eq. (29).

### 8.1.2 The S(IRP$_\text{P}$)$^\text{y}$R algorithm and its variant

With the regularisation method described in Sect. 8.1.1, the S(IRP$_\text{P}$)$^\text{y}$R has 3 parameters:

- the ensemble size $N_\text{e}$;

- the localisation radius $r$ used to compute the $\omega^*$ weights (step 4 of Algorithm 7) as defined by Eqs. (B1) to (B4);

- the standard deviation $s$ of the regularisation jitter, hereafter simply called "regularisation jitter" to be consistent with the LPF$^\text{x}$s.

For each value of the ensemble size $N_\text{e}$, the localisation radius $r$ and the regularisation jitter $s$ are systematically tuned to yield the lowest RMSE score.

As mentioned in Sect. 7.3.2, the original algorithm designed by Poterjoy (2016) included another tuning parameter, the weight inflation, which serves the same purpose as the regularisation jitter. Based on extensive tests in the L96 model with 8 to 128 particles (not shown here), we have found that using weight inflation instead of regularisation jitter always yields higher RMSEs. Therefore, we have not included weight inflation in the S(IRP$_\text{P}$)$^\text{y}$R algorithm.

In the S(IRP$_\text{P}$)$^\text{y}$R$_\text{c}$ algorithm, the regularisation jitter parameter $s$ is replaced by $\hat{h}$ according to the coloured noise regularisation jitter method. The parameter tuning method is unchanged.

### 8.1.3 The S(IRP$_\text{RK}$)$^\text{y}$R algorithm and its variants

With the regularisation method described in Sect. 8.1.1, the S(IRP$_\text{RK}$)$^\text{y}$R has 3 parameters:

- the ensemble size $N_\text{e}$;

- the localisation radius $r$ used to define the valid tapering matrix $\mathbf{C}$ required for the computation of the prior covariance submatrices (step 2 of Algorithm 8) as defined by Eq. (67);

- the regularisation jitter $s$.

For each value of the ensemble size $N_\text{e}$, the localisation radius $r$ and the regularisation jitter $s$ are systematically tuned to yield the lowest RMSE score.

When using optimal ensemble coupling for the local resampling (step 4 of Algorithm 8), the local minimisation coefficients are computed using Eq. (35). This gives an additional tuning parameter, the distance radius $r_\text{d}$, which is also systematically tuned to yield the lowest RMSE score. When using anamorphosis for the local resampling step, the cdfs of the state variables in the region $U$ are computed in the same way as for LPF$^\text{x}$ algorithms, with a regularisation bandwidth $h = 1$. Finally, when using the coloured noise regularisation jitter method, the parameter $s$ is replaced by $\hat{h}$ and the tuning method stays the same.

### 8.2 RMSE scores for the L96 model

The evolution of the RMSE as a function of the ensemble size $N_\text{e}$ for the LPF$^\text{y}$ algorithms with the L96 model is shown in Fig. 20. The RMSEs obtained with the S(IRP$_\text{P}$)$^\text{y}$R algorithm are comparable to those obtained with the S(IR)$^\text{x}$R algorithm.

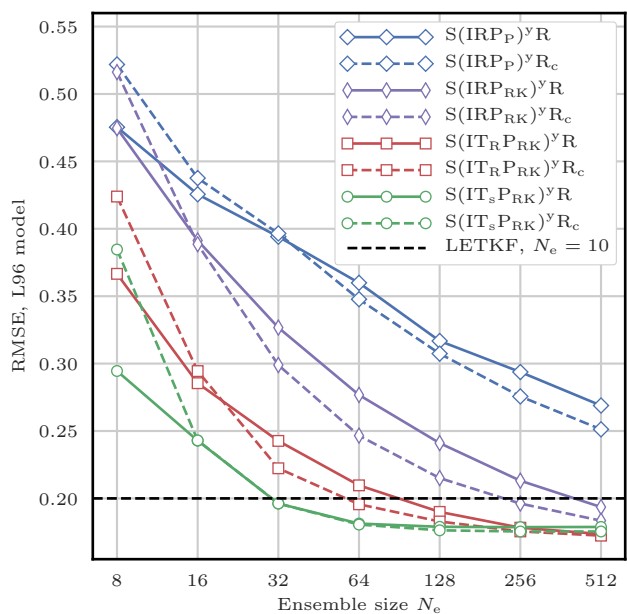

**Figure 20.** RMSE as a function of the ensemble size $N_e$ for the LPF$^y$s.

When using the second-order propagation method, the RMSEs are, as expected, significantly lower. The algorithm is less
sensitive to the curse of dimensionality than the LPF$^x$ algorithms: optimal values of the localisation radius $r$ are significantly
larger and less regularisation jitter $s$ is required. Similarly to the LPF$^x$s, combining the second-order propagation method with
OT-based resampling algorithms (optimal ensemble coupling or anamorphosis) yields important gains in RMSE scores as a
consequence of the minimisation of the update in the region $U$ that needs to be propagated to the region $V$. With a reasonable
number of particles (e.g. 64 for the S(IT$_s$P$_{RK}$)$^y$R algorithm), the scores are significantly lower than those obtained with the
1220 reference EnKF implementation (the ETKF). Finally, we observe that using the coloured noise regularisation jitter method
improves the RMSEs for large ensembles when the local resampling step is performed with the SU sampling algorithm, in a
similar way as for the LPF$^x$s. However when the local resampling step is performed with optimal ensemble coupling or with
anamorphosis, the coloured noise regularisation jitter method barely improves the RMSEs.

### 8.3 RMSE scores for the BV model

The evolution of the RMSE as a function of the ensemble size $N_e$ for the LPF$^y$ algorithms with the BV model is shown in
Fig. 21. Most of the conclusions drawn with the L96 model remain true with the BV model. However, in this case, as the
ensemble size $N_e$ grows, the RMSE decreases significantly more slowly for the S(IRP$_P$)$^y$R and the S(IRP$_P$)$^y$R$_c$ algorithms
than for the other algorithms. Finally, with an ensemble size $N_e \geq 64$ particles, the S(IT$_s$P$_{RK}$)$^y$R and the S(IT$_s$P$_{RK}$)$^y$R$_c$
algorithms yield RMSEs almost equivalent to those of the reference LETKF implementation.

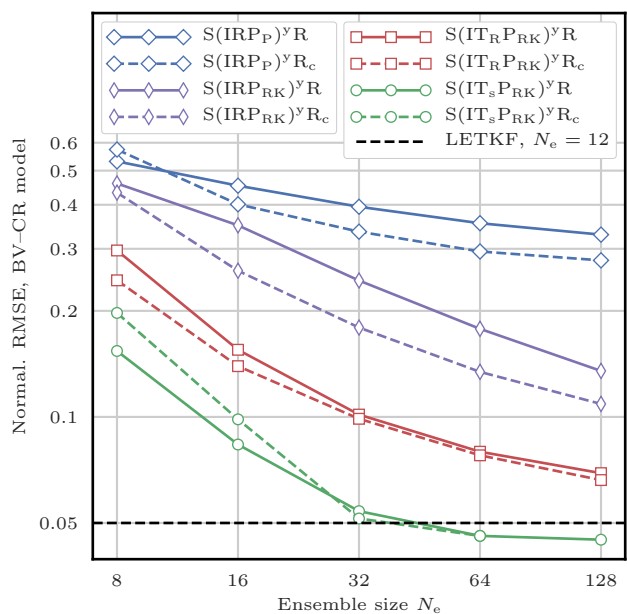

**Figure 21.** RMSE as a function of the ensemble size $N_e$ for the LPF$^y$s. The scores are displayed in units of the observation standard deviation $\sigma$.

## 9  Numerical illustration with a high-dimensional barotropic vorticity model

### 9.1  Experimental setup

In this section, we illustrate the performance of a selection of LPF$^x$s and LPF$^y$s using twin simulations of the BV model in the high resolution (HR) configuration described in Appendix A4.2. Using this configuration yields a higher dimensional DA problem ($N_x = 65536$ and $N_y = 4096$) for which the analysis step is too costly to perform exhaustive tests. Therefore, in this section, we take $N_e = 32$ ensemble members and we monitor the time evolution of the analysis RMSE during $501$ assimilation steps.

As with the CR configuration, all geometrical considerations (blocks and domains, $UVW$ partition...) use the Euclidean distance of the underlying physical space.

### 9.2  Algorithm specifications

For this test series, the selection of algorithms is listed in Table 5. Each algorithm uses the same initial ensemble obtained as follows:

$$\boldsymbol{x}_0^i = \boldsymbol{x}_0 + 0.5 \times \boldsymbol{u} + \boldsymbol{u}^i, \quad i = 1 \dots N_e, \tag{69}$$

with $\boldsymbol{u}$ and the $\boldsymbol{u}^i$ are random vectors whose coefficients are distributed according to a normal law. Such an ensemble is not very close to the truth (in terms of RMSE) and its spread is large enough to reflect the lack of initial information. The LPFs use zero integration jitter and $N_b = N_x$ blocks of size 1 grid point. Approximate optimal values for the localisation radius $r$ and the regularisation jitter ($s$ or $\hat{h}$ depending on the potential colourisation of the noise) are found using several twin experiments with a few hundred assimilation cycles (not shown here). The localisation radius $r$ and the multiplicative inflation for the LETKF are found in a similar manner. When using OT in state space, we only test a few values for the regularisation bandwidth $h$. When using the smoothing by weights, we take the smoothing strength $\alpha_s = 1$ and the smoothing radius $r_s$ is set to be equal to the localisation radius $r$.

### 9.3 RMSE time series

Figure 22 shows the evolution of the instantaneous analysis RMSE for the selected algorithms. Approximate optimal values for the tuning parameters, alongside with average analysis RMSE computed over the final 300 assimilation steps and wall-clock computational times are reported in Table 5. In terms of RMSE scores, the ranking of the methods is unchanged and most of the conclusions for this test series are the same as with the CR configuration.

Thanks to the uniformly distributed observation network, the posterior pdfs are close to Gaussian and therefore the LETKF algorithm can efficiently reconstruct a good approximation of the true state. As expected with this high-dimensional DA problem, the algorithms using a second-order truncation (the LETKF and the $S(I*P_{RK})^y R$ algorithms) are more robust. Optimal values of the localisation radius are qualitatively large, which allows for a better reconstruction of the system dynamics.

For the $S(IR)^x R$ and the $S(IRP_P)^x R$ algorithms, the optimal localisation radius $r$ needs to be very small to counteract the curse of dimensionality. With such small values for $r$, the local domain of each grid point contains only 4 to 13 observation sites. This is empirically barely enough to reconstruct the true state with an RMSE score lower than the observation standard deviation $\sigma$. As in the previous test series, using OT-based local resampling methods or the smoothing by weights step yields significantly lower RMSEs. The RMSEs of the $S(IT_s)^x R$ and the $S(IR)^x S_{PM} R_c$ algorithms, though not as good as that of the LETKF algorithm, show that the true state is reconstructed with an acceptable accuracy. The RMSEs of the $S(IT_s P_{RK})^y R$ and the LETKF algorithms are almost comparable. Depending on the algorithm, the conditioning to the initial ensemble more or less quickly vanishes.

Without parallelisation, we observe that the $N_x$ local analyses of the LPF$^x$s are almost always faster than both the $N_x$ local analyses of the LETKF and the $N_y$ sequential assimilations of the LPF$^y$s. The second-order propagation algorithm is slower because of the linear algebra involved in the method. Poterjoy's propagation algorithm is slower because computing the $\omega^*$ weights is numerically expensive. The LETKF is slower because of the matrices inversion in ensemble space. Finally, the $S(IR)^x S_{PM} R_c$ algorithm is even slower because, in this two-dimensional model, the smoothing by weights step is numerically very expensive.

The difference between the LPF$^x$s and the LPF$^y$s is even more visible on our 24-core platform. The LPF$^y$s are not parallel, that is why they are more than 70 times slower than the fastest LPF$^x$s.

**Table 5.** Characteristics of the algorithms tested with the BV model in the HR configuration (Fig. 22). The LPF$^x$s use zero integration jitter ($q = 0$) and $N_b = N_x$ blocks of size 1 grid point. The LPF$^y$s also use zero integration jitter ($q = 0$). For the LETKF, the optimal multiplicative inflation is reported in the regularisation jitter column. For the S(IR)$^x$S$_{PM}$R$_c$ algorithm, the optimal regularisation jitter bandwidth $\hat{h}$ is reported in the regularisation jitter column as well. The average RMSE is computed over the final 300 assimilation steps and given in units of the observation standard deviation $\sigma$. The wall-clock computational time is the average time spent per analysis step. The simulations are performed on a *single* core of a double Intel Xeon E5-2680 platform (for a total of 24 cores). For comparison, the average time spent per forecast ($\Delta t = 0.5$) for the 32-member ensemble is 0.94 s. The $^*$ asterisk indicates that the local analyses can be carried out in parallel, allowing a theoretical gain in computational time of up to a factor 65536. For these algorithms, the average time spent per analysis step for the *parallelised* runs on this 24-core platform, as well as the acceleration factor, are reported in the last column.

| Algorithm | Loc. radius $r$ [in units of $L$] | Reg. jitter $s$ | Other parameters | Average RMSE [in units of $\sigma$] | 1-core wall-clock time [in s] | 24-core wall-clock time [in s] |
|---|---|---|---|---|---|---|
| S(IRP$_P$)$^y$R | 0.03 | 0.70 | – | 0.90 | 122.18 | – |
| S(IR)$^x$R | 0.02 | 0.55 | – | 0.78 | 7.58$^*$ | 0.54 ($\times$14.04) |
| S(IRP$_{RK}$)$^y$R | 0.07 | 0.25 | – | 0.46 | 52.97 | – |
| S(IR)$^x$S$_{PM}$R$_c$ | 0.05 | 1.0 | $\alpha_s = 1$, $r_s = r$ | 0.38 | 226.20$^*$ | 12.50 ($\times$18.10) |
| S(IT$_s$)$^x$R | 0.08 | 0.11 | $h = 3$ | 0.33 | 13.94$^*$ | 0.86 ($\times$16.21) |
| S(IT$_s$P$_{RK}$)$^y$R | 0.20 | 0.01 | $h = 1$ | 0.13 | 64.79 | – |
| LETKF | 0.35 | 1.04 | – | 0.10 | 103.90$^*$ | 5.09 ($\times$20.41) |

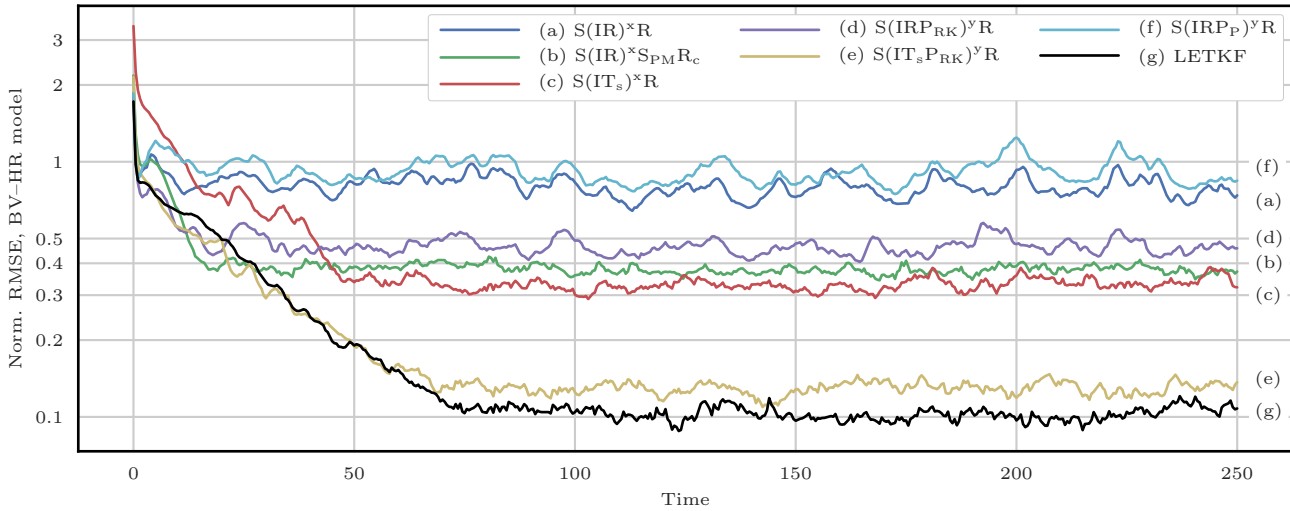

**Figure 22.** Instantaneous analysis RMSE for the selection of algorithms detailed in Table 5. The scores are displayed in units of the observation standard deviation $\sigma$.

## 10 Conclusions

The curse of dimensionality is a rather well-understood phenomenon in the statistical literature and it is the reason why PF methods fail when applied to high-dimensional DA problems. We have recalled the main results related to weight degeneracy of PFs, and why the use of localisation can be used as a fix. Yet, implementing localisation in PF analysis raises two major issues: the gluing of locally updated particles and potential physical imbalance in the updated particles. Adequate solutions to these issues are not obvious, witness the few but dissimilar LPF algorithms developed in the geophysical literature. In this article we have proposed a theoretical classification of LPF algorithms into two categories. For each category, we have presented the challenges of local particle filtering and we have reviewed the ideas that lead to practical implementation of LPFs. Some of them, already in the literature, have been detailed and sometimes generalised while others are new in this field and yield improvements in the design of LPF methods.

With the LPF$^x$ methods, the analysis is localised by allowing the analysis weights to vary over the grid points. We have shown that this yields an analysis pdf from which only the marginals are known. The local resampling step is mandatory to reconstruct global particles, that are obtained by assembling the locally updated particles. The quality of the updated ensemble directly depends on the regularity of the local resampling. This is related to unphysical discontinuities in the assembled particles. Therefore we have presented practical methods to improve the local resampling step by reducing the unphysical discontinuities.

In the LPF$^y$ methods, localisation is introduced more generally in the conditional density for one observation by the means of a state partition. The goal of the partition is to build a framework for local particle filtering without the discontinuity issue inherent to LPF$^x$s. However, this framework is irreconcilable with algorithms based on pure "importance, resampling" methods. We have shown how two hybrid methods could yet be used as an implementation of this framework. Besides, we have emphasised the fact that with these methods, observations are by construction assimilated sequentially, which is a great disadvantage when the number of observations in the DA problem is high.

With localisation, a bias is introduced in the LPF analyses. We have shown that, depending on the localisation parametrisation, some methods can yield an analysis step equivalent to that of global PF methods which are known to be asymptotically Bayesian.

We have implemented and systematically tested the LPF algorithms with twin simulations of the L96 model and the BV model. A few observations could be made from these experiments. With these models, implementing localisation is simple and works as expected: the LPFs yield acceptable RMSE scores even with small ensembles, in regimes where global PF algorithms are degenerate. In terms of RMSEs, there is no clear advantage of using Poterjoy's propagation method (designed to avoid unphysical discontinuities) over the (simpler) LPF$^x$ algorithms, which have a lower computational cost. As expected, algorithms based on the second-order propagation method are less sensitive to the curse of dimensionality and yields the lowest RMSE scores. We have shown that using OT-based local resampling methods always yields important gains in RMSE scores. For the LPF$^x$s, it is a consequence of mitigating the unphysical discontinuities introduced in the local resampling step. For the LPF$^y$s, it is a consequence of the minimisation of the update at the observation site that needs to be propagated to nearby grid points.

The successful application of the LPFs to DA problems with a perfect model is largely due to the use of regularisation jitter. Using regularisation jitter introduces an additional bias in the analysis alongside with an extra tuning parameter. For our numerical experiments, we have introduced two jittering methods: either using regularisation noise with fixed statistical properties (white noise) or by scaling the noise with the ensemble anomalies (coloured noise). We have discussed the relative performance of each method and concluded that there is room for improvement in the design of regularisation jitter methods for PFs.

In conclusion, introducing localisation in the particle filter is a relatively young topic that can benefit from more theoretical and practical developpements.

First, the resampling step is the main ingredient in the success, or not, of an LPF algorithm. The approaches based on optimal transport offer an elegent and quite efficient framework to deal with the discontinuity issue inherent to local resampling. However, the algorithms derived in this article could be improved. For example, it would be desirable to avoid the systematic reduction to one-dimensional problems when using optimal transport in state space. Besides, other frameworks for local resampling based on other theories could be conceived.

Second, the design of the regularisation jitter methods can be largely improved. Regularisation jitter is mandatory when the model is perfect. Even with stochastic models, it can be beneficial for example when the magnitude of the model noise is too small for the LPFs to perform well. Ideally, the regularisation jitter methods should be adaptive and built concurrently with the localisation method.

Third, with the localisation framework presented in this article, one cannot directly assimilate non-local observations. The ability to assimilate non-local observations becomes increasingly important with the prominence of satellite observations.

Finally, our numerical illustration with the BV model in the HR configuration is successful and shows that the LPF algorithms have the potential to work with high-dimensional systems. Nevertheless further research is needed to see if the LPFs can be used with realistic models. Such an application would require an adequate definition of the model noise and the observation error covariance matrix. Even if the local resampling methods have been designed to minimise the unphysical discontinuities, this will have to be carefully checked because this is a critical point in the success of the LPF. Last, the regularisation jitter method has to be chosen and tuned in adequation with the model noise. In particular, the magnitude of the jitter will almost certainly depend on the state variable.

*Acknowledgements.* The authors thank the Editor, Olivier Talagrand, and three reviewers, Stephen G. Penny and two anonymous reviewers, for their useful comments, suggestions and thorough reading of the manuscript. The authors are grateful to Patrick Raanes for enlightening debates and to Sebastian Reich for suggestions. CEREA is a member of Institut Pierre–Simon Laplace (IPSL).

## Appendix A: Numerical models

### A1 The Gaussian linear model

The Gaussian linear model is the simplest model with size $N_x$ whose prior distribution is

$$\boldsymbol{x}_0 \sim \mathcal{N}\left(\mathbf{0}, p^2 \mathbf{I}\right), \tag{A1}$$

whose transition distribution is

$$\boldsymbol{x}_{k+1} - a\boldsymbol{x}_k = \boldsymbol{w}_k \sim \mathcal{N}\left(\mathbf{0}, q^2 \mathbf{I}\right), \tag{A2}$$

and whose observation distribution is

$$\boldsymbol{y}_k - h\boldsymbol{x}_k = \boldsymbol{v}_k \sim \mathcal{N}\left(\mathbf{0}, \sigma^2 \mathbf{I}\right). \tag{A3}$$

### A2 Generic model with Gaussian additive noise

The Gaussian linear model can be generalised to include nonlinearity in the model $\mathcal{M}$ and in the observation operator $\mathcal{H}$. In this case, the transition distribution is:

$$\boldsymbol{x}_{k+1} - \mathcal{M}\left(\boldsymbol{x}_k\right) = \boldsymbol{w}_k \sim \mathcal{N}\left(\mathbf{0}, \mathbf{Q}\right), \tag{A4}$$

and the observation distribution is:

$$\boldsymbol{y}_k - \mathcal{H}\left(\boldsymbol{x}_k\right) = \boldsymbol{v}_k \sim \mathcal{N}\left(\mathbf{0}, \mathbf{R}\right), \tag{A5}$$

where $\mathbf{Q}$ and $\mathbf{R}$ are the covariance matrices of the additive model and observation errors.

### A3 The Lorenz 1996 model

The Lorenz 1996 model (Lorenz and Emanuel, 1998) is a low-order one-dimensional discrete chaotic model whose evolution is given by the following set of ODEs:

$$\frac{\mathrm{d}x_n}{\mathrm{d}t} = \left(x_{n+1} - x_{n-2}\right)x_{n-1} - x_n + F, \quad n = 1 \ldots N_x, \tag{A6}$$

where the indices are to be understood with periodic boundary conditions: $x_{-1} = x_{N_x - 1}$, $x_0 = x_{N_x}$, $x_1 = x_{N_x + 1}$ and where the system size $N_x$ can take arbitrary values. These ODEs are integrated using a fourth-order Runge–Kutta method with a time step of $0.05$ time unit.

In the standard configuration, $N_x = 40$ and $F = 8$ which yields a chaotic dynamics with a doubling time around $0.42$ time unit. The observations are given by

$$\boldsymbol{y}_k = \boldsymbol{x}_k + \boldsymbol{v}_k, \quad \boldsymbol{v}_k \sim \mathcal{N}\left(\mathbf{0}, \mathbf{I}\right), \tag{A7}$$

and the time interval between consecutive observations is $\Delta t = 0.05$ time unit, which represents $6$ h of real time and corresponds to a model autocorrelation around $0.967$.

## A4   The barotropic vorticity model

The barotropic vorticity model describes the evolution of the vorticity field of a two-dimensional incompressible homogeneous fluid in the $x_1 - x_2$ plane. The time evolution of the unknown vorticity field $\zeta$ is governed by the scalar equation

$$\frac{\partial \zeta}{\partial t} + \mathrm{J}\left(\psi, \zeta\right) = -\xi\zeta + \nu\Delta\zeta + F, \tag{A8}$$

and $\zeta$ is related to the stream function $\psi$ through

$$\Delta\psi = \zeta. \tag{A9}$$

In these equations, $\mathrm{J}\left(\psi, \zeta\right)$ is the advection of the vorticity by the stream, defined as

$$\mathrm{J}\left(\psi, \zeta\right) = \frac{\partial\psi}{\partial x_1}\frac{\partial\zeta}{\partial x_2} - \frac{\partial\psi}{\partial x_2}\frac{\partial\zeta}{\partial x_1}, \tag{A10}$$

$\xi \in \mathbb{R}^+$ is the friction coefficient, $\nu \in \mathbb{R}^+$ is the diffusion coefficient and $F$ is the forcing term, that may depend on $x_1$, $x_2$ and $t$. The system is characterised by homogeneous two-dimensional turbulence. The friction extracts energy at large scale, the diffusion dissipates vorticity at small scale and the forcing injects energy in the system. The number of degrees of freedom in this model can be roughly considered to be proportional to the number of vortices (Snyder, 2012, personal communication).

The equations are solved with $P^2$ grid points regularly distributed over the simulation domain $[0, L]^2$ with doubly periodic boundary conditions. Our time integration method is based on a semi-Lagrangian solver with a constant time step $\delta t$ as follows:

1. At time $t$, solve Eq. (A9) for $\psi$.

2. At time $t$, compute the advection velocity with second-order centered finite differences of the field $\psi$.

3. The advection of $\zeta$ during $t$ and $t + \delta t$ is computed by applying a semi-Lagrangian method to the left-hand side of Eq. (A8). The solver cannot be more precise than first-order in time, since the value of $\psi$ is not updated during this step. Therefore, our semi-Lagrangian solver uses the first-order forward Euler time integration method. The interpolation method used is the cubic convolution interpolation algorithm, which yields a third-order precision with respect to the spatial discretisation. In this step, the right-hand side of Eq. (A8) is ignored.

4. Integrate $\zeta$ from $t$ to $t + \delta t$ by solving Eq. (A8) with an implicit first-order time integration scheme, in which the advection term is the one computed in the previous step.

For the numerical experiments of this article, the spatial discretisation is fine enough such that the spatial interpolation error in the semi-Lagrangian step is negligible when compared to the time integration error. As a consequence, the overall integration method is first-order in time. For the DA experiments with this model, we define and use two configurations.

### A4.1 Coarse resolution configuration

The coarse resolution configuration is based on the following set of physical parameters:

$$L = 1, \tag{A11}$$

$$\xi = 10^{-2}, \tag{A12}$$

$$\nu = 5 \times 10^{-5}, \tag{A13}$$

the deterministic forcing is given by

$$F(x_1, x_2) = 0.25 \sin(4\pi x_1) \sin(4\pi x_2), \tag{A14}$$

and the space-time discretisation is

$$\delta t = 0.1, \tag{A15}$$

$$\delta x = \frac{L}{P} = \frac{1}{32}, \tag{A16}$$

which yields $N_{\mathrm{x}} = (L/\delta x)^2 = 1024$. The spatial discretisation is enough to allow a reasonable description of a few (typically five to ten) vortices inside the domain. The temporal discretisation is empirically enough to ensure the stability of the integration method and allows a fast computation of the trajectory. The physical parameters are chosen to yield a proper time evolution of the vorticity $\zeta$.

The initial true vorticity field for the DA twin experiments is the vorticity obtained after a run of $100$ time units starting from a random, spatially correlated field. The system is partially observed on a regular square mesh with one observation site every 2 grid points in each direction, i.e. $N_{\mathrm{y}} = 256$ observation sites for $N_{\mathrm{x}} = 1024$ grid points. At every cycle $k$, the observation at site $(q_1, q_2) \in \{1 \dots P/2\}^2$ is given by

$$y_{q_1, q_2} = \zeta_{2q_1-1, 2q_2-1} + v_{q_1, q_2}, \tag{A17}$$

$$v_{q_1, q_2} \sim \mathcal{N}(0, \sigma^2), \tag{A18}$$

with $\sigma = 0.3$, about one tenth of the typical vorticity variability. The time interval between consecutive observations is $\Delta t = 0.5$ time unit, which was chosen to match approximately the model autocorrelation of $0.967$ of the L96 model in the standard configuration.

We have checked that the vorticity flow remains stationary over the total simulation time of our DA twin experiments chosen to be $10^4 \Delta t$. Due to the forcing $F$, the flow remains uniformally and stationarily turbulent during the whole simulation. Compared to other experiments with the barotropic vorticity model (e.g. van Leeuwen and Ades, 2013; Ades and van Leeuwen, 2015; Browne, 2016), $\Delta t$ is smaller and $\sigma$ is larger, but the number of vortices is approximately the same with much fewer details.

## A4.2 High resolution configuration

For the high resolution configuration, the physical parameters are

$$L = 1, \tag{A19}$$

$$\xi = 5 \times 10^{-5}, \tag{A20}$$

$$\nu = 10^{-6}; \tag{A21}$$

the deterministic forcing is given by

$$F(x_1, x_2) = 0.75 \sin(12\pi x_1) \sin(12\pi x_2), \tag{A22}$$

and the space-time discretisation is

$$\delta t = 0.1, \tag{A23}$$

$$\delta x = \frac{L}{P} = \frac{1}{256}, \tag{A24}$$

which yields $N_{\mathrm{x}} = (L/\delta x)^2 = 65536$. Compared to the coarse resolution configuration, this set of parameters yields a vorticity field with more vortices (typically several dozens). The associated DA problem has therefore many more apparent or effective degrees of freedom. The initial true vorticity field for the DA twin experiments is the vorticity obtained after a run of 100 time units starting from a random, spatially correlated field. The system is partially observed on a regular square mesh with one observation site every 4 grid points in each direction, i.e. $N_{\mathrm{y}} = 4096$ observation sites for $N_{\mathrm{x}} = 65536$ grid points. At every cycle $k$, the observation at site $(q_1, q_2) \in \{1 \ldots P/4\}^2$ is given by

$$y_{q_1, q_2} = \zeta_{4q_1-1, 4q_2-1} + v_{q_1, q_2}, \tag{A25}$$

$$v_{q_1, q_2} \sim \mathcal{N}(0, \sigma^2), \tag{A26}$$

and we keep the values $\Delta t = 0.5$ time units and $\sigma = 0.3$ from the coarse resolution configuration. We have checked that the vorticity flow remains stationary over the total simulation time of our DA twin experiments chosen to be $500\Delta t$. Due to the forcing $F$, the flow remains uniformly and stationarily turbulent during the whole simulation.

## Appendix B: Update formulae of Poterjoy's LPF

Following Poterjoy (2016), we derived the following formulas for the $\omega^*$ weights required in the propagation step of Poterjoy's LPF described in Sect. 7.3.2:

$$W = \sum_{i=1}^{N_e} w^i = \sum_{i=1}^{N_e} p\left(y_q \,|\, \boldsymbol{x}^i\right), \tag{B1}$$

$$c_n = \frac{\alpha N_e \left(1 - G\left(\frac{d_{q,n}}{r}\right)\right)}{W G\left(\frac{d_{q,n}}{r}\right)}, \tag{B2}$$

$$\omega_n^{\mathrm{a}} = \sqrt{\frac{\sigma_n^2}{\frac{1}{N_e - 1} \sum_{i=1}^{N_e} \left\{ x_n^{\phi(i)} - \overline{x}_n + c_n \left(x_n^i - \overline{x}_n\right) \right\}^2}}, \tag{B3}$$

$$\omega_n^{\mathrm{f}} = c_n \omega_n^{\mathrm{a}}, \tag{B4}$$

where $W$ and $c_n$ are ancillary variables, $\alpha$ is the constant used for the computation of the local weights (see Eq. (28)), $G$ is the tapering function, $d_{q,n}$ is the distance between the $q$-th observation site and the $n$-th grid point, $r$ is the localisation radius, $\overline{x}_n$ is the mean and $\sigma_n$ the standard deviation of the weighted ensemble $\left\{\left(x_n^i, w^i\right), \, i = 1 \ldots N_e\right\}$. The particles are then updated using Eq. (65).

In Poterjoy (2016), the pdfs are implicitly normalised, such that the constant $\alpha$ is 1. Therefore, our update Eqs. (B1) to (B4) are equivalent to the update Eqs. (A10), (A11), (A5) and (A3) derived by Poterjoy (2016). Note that there is a typing mistake which renders one update equation in Algorithm 1 of Poterjoy (2016) incorrect (last equation on page 66).

## Appendix C: Nonlinear test series with the L96 model

As a complement to the mildly nonlinear test series of Sects. 5, 6, 8 and 9, we provide here a strongly nonlinear test series. We consider the L96 model in the standard configuration described in Appendix A3 with the only difference that the $N_y = N_x$ observations at each assimilation cycle are now given by

$$\forall n \in \{1 \ldots N_x\}, \, y_n = \ln|x_n| + v_n, \quad v_n \sim \mathcal{N}(0, 1). \tag{C1}$$

This strongly nonlinear configuration has been used e.g. by Poterjoy (2016).

Similarly to the mildly nonlinear test series, the distance between the truth and the analysis is measured with the average analysis RMSE. The runs are $9 \times 10^3 \Delta t$ long with an additional $10^3 \Delta t$ spin-up period. Optimal values for the tuning parameters of each algorithms are found using the same method as for the mildly nonlinear test series. Figure C1 shows the evolution of the RMSE as a function of the ensemble size $N_e$ for the LETKF and for the main LPF$^{\mathrm{x}}$ and LPF$^{\mathrm{y}}$ algorithms.

As expected in this strongly nonlinear test series, the EnKF fails at accurately reconstructing the true state. By contrast, all LPFs yield at some point an RMSE under $\sigma = 1$ (the observation standard deviation). Regarding the ranking of the methods,

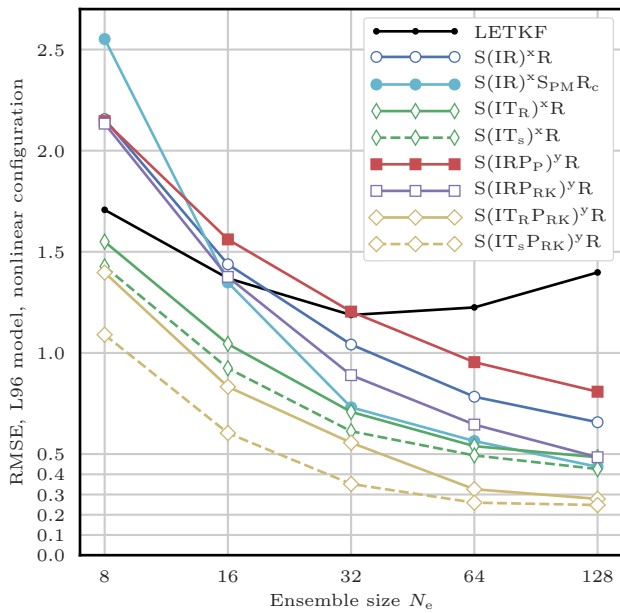

**Figure C1.** RMSE as a function of the ensemble size $N_e$ for the LETKF and the main LPFs with the L96 model in the strongly nonlinear configuration. Note that the ultimate increase of the RMSE of the LETKF with the ensemble size could have been avoided by using random rotations in ensemble space.

most conclusions from the mildly nonlinear case remain true. The best RMSE scores are obtained with algorithms using OT-based resampling methods. Combining the smoothing by weights with the coloured noise regularisation jitter methods yields almost equally good scores as the LPF$^x$ algorithms using OT. Finally, using the second-order propagation method yields the lowest RMSEs despite the non-Gaussian error distributions that result from nonlinearities.

## Appendix D: Rank histograms for the L96 model

As a complement to the RMSE test series, we compute rank histograms of the ensembles (Anderson, 1996; Hamill, 2001). For this experiment, the DA problem is the same as the one in Sects. 5 and 8: the L96 model is used in its standard configuration.

Several algorithms are selected with characteristics detailed in Table D1. The histograms are obtained separately for each state variable by computing the rank of the truth in the unperturbed analysis ensemble (i.e. the analysis ensemble before the regularisation step for the LPFs). To ensure the convergence of the statistical indicators, the runs are $10^5 \Delta t$ long with a $10^3 \Delta t$ spin-up period. The mean histograms (averaged over the state variables) are reported in Fig. D1.

The histogram of the EnKF is quite flat in the middle, its edges reflect a small overdispersion. The histogram of the tuned S(IR)$^x$R algorithm is characterised by a large hump, showing that the ensemble is overdispersive. At the same time, the high frequencies at the edges show that the algorithm yields a poor representation of the distribution tails (as most PF methods). The overdispersion of the ensemble is a consequence of the fact that the parameters have been tuned to yield the best RMSE

**Table D1.** Rank histograms computed with the L96 model in the standard configuration (see Appendix D). All LPFs use zero integration jitter ($q = 0$). The localisation radii are given in number of grid points. For the ETKF, the optimal multiplicative inflation is reported in the regularisation jitter column. The $^*$ asterisk in the RMSE column indicates that the algorithm parameters have been tuned to yield the lowest RMSE score. The first column indicates the corresponding panel in Fig. D1.

| Panel | Algorithm | Ens. size $N_e$ | Loc. radius $r$ | Reg. jitter $s$ | Other parameters | RMSE |
|-------|-----------|-----------------|-----------------|-----------------|------------------|------|
| (a) | ETKF | 20 | $\infty$ | 1.02 | — | $0.188^*$ |
| (b) | $S(IR)^x R$ | 128 | 8 | $10.0 \times 10^{-2}$ | $N_b = 10$ | $0.289^*$ |
| (c) | $S(IT_s)^x R$ | 128 | 20 | $4.5 \times 10^{-2}$ | $h = 1$ | $0.215^*$ |
| (d) | $S(IT_s P_{RK})^y R$ | 128 | 80 | $1.0 \times 10^{-2}$ | $h = 1$ | $0.180^*$ |
| (e) | $S(IR)^x R$ | 128 | 5 | $8.0 \times 10^{-2}$ | $N_b = 40$ | 0.500 |
| (f) | $S(IT_s)^x R$ | 128 | 10 | $3.0 \times 10^{-2}$ | $h = 1$ | 0.228 |

score, regardless of the flatness of the rank histogram. With a different set of parameter, the untuned $S(IR)^x R$ algorithm yields a rank histogram much flatter. In this case, the regularisation jitter is lower (which explains the fact that the ensemble is less overdispersive) and the localisation radius smaller (to avoid the filter divergence). Of course, the RMSE score for the untuned S(IR)$^x$R algorithm is higher than for its tuned version. Similar conclusions can be found with the histograms of the tuned and untuned $S(IT_s)^x R$ algorithm. Note that in this case the histograms are significantly flatter than with the $S(IR)^x R$ algorithm. Finally, the histogram of the (tuned) $S(IT_s P_{RK})^y R$ is remarkably flat.

In summary, the rank histograms of the LPFs are in general rather flat. The ensemble are more or less overdispersive, this is a consequence of the use of regularisation jitter, necessary to avoid the filter divergence. As most PF methods, the LPFs yield a poor representation of the distribution tails.

## Appendix E: The multinomial and the SU sampling algorithms

We describe here the multinomial and the SU sampling algorithms, which are the most common resampling algorithms. In this algorithms, highly probable particles are selected and duplicated while particles with low probability are discarded. Algorithms 9 and 10 describe how to construct the resampling map $\phi$ according to the multinomial resampling and the SU sampling algorithms, respectively. The resampling map $\phi$ is the map such that $\phi(i)$ is the index of the $i$-th particle selected for resampling.

Both algorithms only require the cumulative weights $c^i$ that can easily be obtained from the importance weights $w^i$ using

$$c^i = \sum_{j=1}^{i} w^j, \tag{E1}$$

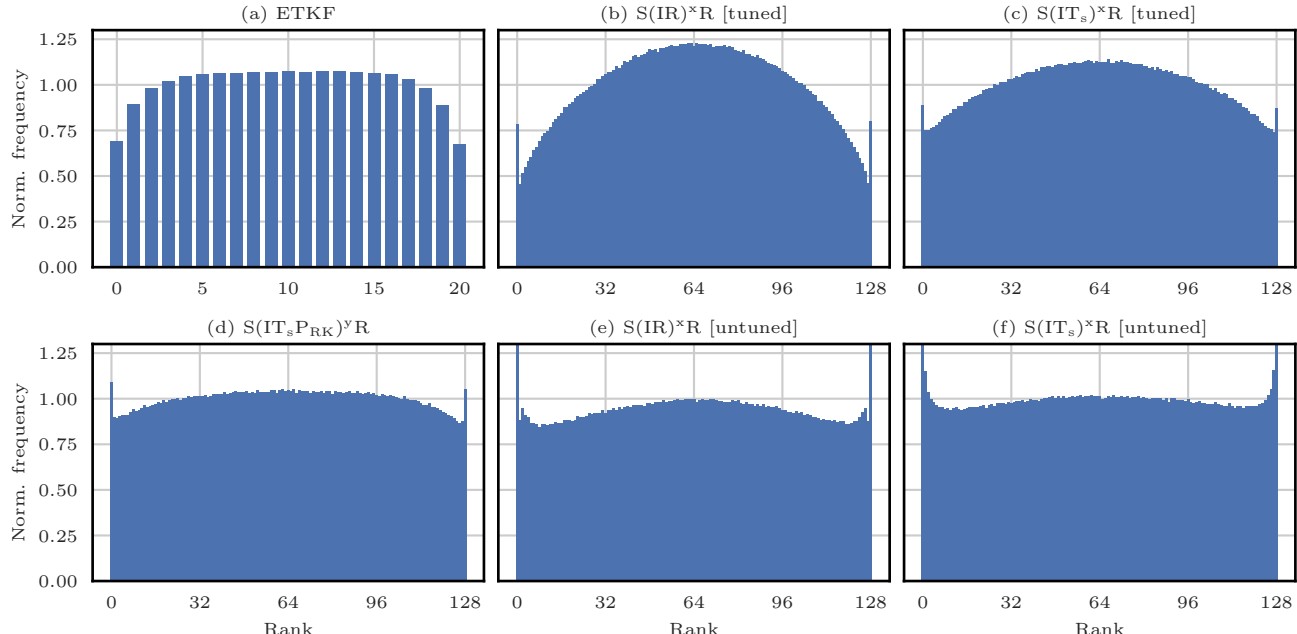

**Figure D1.** Rank histograms for the selection of algorithms detailed in Table D1. The frequency is normalised by $N_e + 1$ (the number of bins).

and both algorithms use random number(s) generated from $\mathcal{U}(0, 1)$, the uniform distribution over the interval $[0, 1]$. Because of these random numbers, both algorithms introduce sampling noise. Moreover, it can be shown that the SU sampling algorithm has the lowest sampling noise (see e.g., van Leeuwen, 2009).

---

**Algorithm 9** Multinomial resampling algorithm

---

**Require:** Weights $w^i$, $i = 1 \ldots N_e$
1: Compute the cumulative weights $c^i$
2: **for** $i = 1$ **to** $N_e$ **do**
3:      Draw a random number $u^i \sim \mathcal{U}(0, 1)$
4:      Set $\phi(i) = \min\left\{ j \in \{1 \ldots N_e\} \setminus u^i \leq c^j \right\}$
5: **end for**
6: **return** Resampling map $\phi$

---

---

**Algorithm 10** SU sampling algorithm

---

**Require:** Weights $w^i$, $i = 1 \ldots N_e$

1: Compute the cumulative weights $c^i$

2: Draw a random number $u \sim \mathcal{U}(0, 1)$

3: **for** $i = 1$ **to** $N_e$ **do**

4:     Compute $u^i = \frac{u+i-1}{N_e}$

5:     Set $\phi(i) = \min\left\{ j \in \{1 \ldots N_e\} \setminus u^i \leq c^j \right\}$

6: **end for**

7: **return** Resampling map $\phi$

---

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
