# Peer review of "Review article: Comparison of local particle filters and new implementations Version 2.1, September 27, 2018"

_Nonlinear Processes in Geophysics, 2018_

## Referee Comment (RC1) · Anonymous Referee #1 · 22 Mar 2018

**1   General comments**

This paper reviews the current research on localisation in particle filters and demonstrates, with results, how some of the current ideas can be developed further. The localisation methods are split in to two types; state domain localisation and sequential-observation localisation. For each type the algorithms are summarised in a general manner, links to previous research are given and any differences noted. The different methods are then explored using the L96 and a baratropic vorticity model. This exploration not only examines the performance of the algorithms themselves but also includes an assessment of the behaviour of more general choices that can be made

with regard to particle filters.

I think this is an excellent paper. It is clearly written and very nicely categorises how localisation fits in to the general literature on solving filter degeneracy in particle filters. Just enough mathematical description and introduction is given to allow an understanding of the different algorithms without overloading the reader with unnecessary detail. I'm not always convinced that there is sufficient detail in the algorithms for someone to reproduce them based on this paper alone, but since it is a review paper and the relevant references are given, I actually think the simplification aids understanding and would be an advantage when read in conjunction with the original papers.

The paper is very long and contains a huge amount of both literature review and actual results. However, again since this is a review paper and the results are all relevant and succinctly summarised I think this is acceptable. If the paper needed to be shortened then for me the sections on the different types of regularisation could perhaps be stated, rather than evidence given, but I found the results interesting and felt they added to the paper.

I think the two slight weaknesses of the paper, the use of the perfect model (not normally used with particle filters and slightly addressed with the discussion on pre-regularisation) and RMSE as the performance indicator, are acknowledged in section 5.1 and I appreciated the inclusion of the rank histograms in the appendix to at least give some indication of the ability of the schemes to correctly represent the posterior. I think a detailed analysis on the ability of the different localisation methods to correctly represent the posterior as a whole is beyond the scope of the paper (and generally very difficult to assess anyway) and the RMSE provides at least a consistent measure of comparison between the different schemes. I would hope that if further research is to be conducted with the best performing localisation schemes then a more detailed analysis of this question would be covered.

I do have a few questions and comments that I have listed below

[Figure]

**2 Specific comments**

Do I understand correctly that your observations in the L96 setup are of every variable at every time step? So your discussion on pre-regularisation in some way is a discussion on whether to use model error or not? Perhaps this could be made more explicit in the main text. If observations are of every variable at every time step then it will strongly constrain the posterior pdfs to be Gaussian and may well be influencing the results seen. This is alluded to in the main text but I think it would add insight to explicitly state this in the discussion in section 5.1 and that it is also addressed through the use of the barotropic vorticity model (which I understand uses a more sparse observational system in space and time).

The paper outlines quite a few different extensions to existing localisation methods. This is stated in the abstract and introduction and when I read the different algorithm descriptions in detail I could find the paragraph were the differences were noted. However, in general I wasn't really left with a strong feeling of where you had introduced new elements and what benefit they had brought i.e. Section 4.4.4 is entirely new work that has the best result for L96 for the state domain localisation but this is only remarked on in the final paragraph of the algorithm description section. If it was possible to include a small summary that highlights the new work and the improvement it brings, either within an already existing concluding section or as something separate, then I think it is a chance to bring your work to the fore. It would also explicitly demonstrate how setting individual schemes in to a general context can bring benefits. This could be split between the state domain localisation and sequential-observation localisation if that was the more natural division.

**3 Technical corrections**

Page 4, line 97. A capital N has slipped in to representatioN
Page 15, line 386. It should be 'in order to preserve part of the spatial structure held in the prior particles'
Page 16, line 404. So $E^r$ has discontinuities?
Page 22, line 543. The sentence doesn't really make grammatical sense.
Page 25, line 625. I assume this is 's' but it would be good to explicitly state it.

---

## Referee Comment (RC2) · S.G. Penny (Referee) · 18 Jun 2018

NPG Farchi and Bocquet 2018

General comments:

This manuscript takes nice steps in consolidating the literature in the field, comparing approaches, and offering new ideas in the development of localized Particle Filters. I do recommend that it be accepted for publication after revisions satisfying the following major points:

1) The relative costs between the methods should be calculated and compared, along with RMSE, in each section. Some discussion of costs is made in passing, but no quantitative analyses are offered until the end where a few methods are compared. I suggest extending this to each of the major direct comparisons at the end of the LPFx and LPFy sections.

2) It would be useful to show the results of the EnKF baseline, both in RMSE and computation time. Since the local PF variants are not outperforming the EnKF baseline, the authors should consider some special case scenarios in which the PF does outperform the EnKF as a motivation for continued development of the local PFs and to show why local PFs may also have advantages over standard EnKFs.

3) In general, I find the algorithm names confusing. The first half of the paper uses a complex coding system, while the second half credits the authors who developed the methods. A more consistent and simpler naming convention would be nice from a reader's point of view, and should be used throughout. In addition, a single table describing every algorithm name, what is does, and what section it can be found in, would also help to add clarity.

From what I can tell, the S(IR)^xSR_a method appears closest to that of Penny and Miyoshi (2016), as it uses smoothing of weights and adaptive "regularisation jitter" based on the ensemble perturbations, and I think this should be given proper credit, as one of the few LPF methods offered in the geophysical literature that has a combination of good performance and low computational cost.

Specific comments:

L28:

Also cite Kalnay and Yang regarding the "Running-in-Place" method

from 2008:

Kalnay, E., S.-C. Yang, 2008: Accelerating the spin-up of Ensemble Kalman Filtering. https://arxiv.org/pdf/0806.0180.pdf

and 2012:

Yang, S.-C., Kalnay, E., and Hunt, B. R.: Handling nonlinearity in Ensemble Kalman Filter: Experiments with the three variable Lorenz model, Mon. Weather Rev., 240, 2628–2646, doi: 10.1175/MWR-D-11-00313.1, 2012a.
https://journals.ametsoc.org/doi/abs/10.1175/MWR-D-11-00313.1

L39:

comma after hybridisation

L43:

[fewer] particles

L60:

degree[s] of freedom

L63:

geophysical system[s]

L85:

I'm assuming y_k:0 is the set of all observations from time t=0 to time t=k, but perhaps you can state that explicitly.

91:

Of course, there are many different goals in data assimilation. This is a typical goal. My immediate reaction here is that the DA filtering problem consists in estimating pi_k+1|k. This is the goal at least, usually, to make a prediction. Perhaps you can say -

"The DA filtering problem consists in estimating pi_k|k and pi_k+1|k with given realizations of y_k:0."

L97:

particle representatio[n]

L111:

What do you mean by "pure ensemble transformations"?

L170:

I'm not sure I understand why this is remarkable. Could you elaborate?

L177:

"… to more elaborate algorithms …"

L182:

"… models [have led] to weight degeneracy…"

L196:

" … it [might seem] surprising that, although MC method[s] have…"

L214/L217:

You switch tenses, first referring to Synder et al. (2008) as a set of authors, and then referring to Snyder et al. (2008) as a paper. Because "et al." means "and others", I prefer the former and recommend changing L217 to:

"Snyder et al. (2008) [do] not illustrate…"

L223:

"…optimal importance proposal [density]…"

L227:

"… does not [primarily] come from …"

L238:

"… [elaborate] models …"

L252:

It seems awkward to begin a sentence with a variable name. Perhaps used instead:

"The quantity tao^2 would then be defined using"

L266:

While I appreciate the implication of calling this a 'discontinuity', there are some complications in defining a concept of continuity on a discrete model grid. Some discussion should be made regarding this point.

L281:

Perhaps you could list some of the past examples of this type via citation.

L285:

Again, I suggest citing a few example of this type as well.

L332:

Not within a circle, but within some general local region. A circle is a common choice.

L367:

"and decrease[s] exponentially"

L371:

The "size" of the blocks using what measure? Number of grid points?

L386:

I think "hold" should be "held"

L390-415:

I'm not sure if the point was adequately made that neighboring weights can be made arbitrarily smooth by letting the radius of the taper function ($r\_s$) get large. I.e. as $r\_s$ goes to infinity, the global PF solution is recovered.

In that sense, I'm not sure why the additional alpha smoothing step is made explicit.

L543:

"only [a] big ensemble"

L561:

"RMSE offers a" to "RMSE offer a"

L562:

I'm not sure it is settled that the RMSE of the mean is an adequate measure of the PF performance, given that the distribution may not have the mean and mode equal.

Further, if we are to adopt a PF solution over an EnKF, then we are acknowledging that the primary data assimilation goal is specifically not mean state estimation, but rather estimation of the state distribution.

L572:

"yield[s]"

L595:

I don't understand what this first sentence means. What does it mean to have more information than the truth?

P24-25:

It would be nice if Figures 4 and 5 were closer to the referencing text. Perhaps you can make that request of the editors.

P25:

It appears here that you are using a fixed parameter for the 'regularization jitter'. Have you compared this the LPF of Penny and Miyoshi (2016) that set this value adaptively based on the analysis ensemble spread?

L651:

I'm confused how the higher RMSEs of the $S(IR\_SU)^x R$ algorithm indicates an efficient approach. Could the authors elaborate. What is the RMSE ratio used in Figure 9? Why does the figure caption say "RMSE" while the y-axis says "RMSE ratio"?

L 675:

I need a reminder at this point - E is the set of ensemble members and X is the set of perturbations around the ensemble mean? Are the $x^i$ in (46) the columns of E?

L 697:

change "as following" to "as follows"

How is equation (48) different than (46)?

Could you instead just say it is defined as in (46) with a new formulation for the Gaussian regularization jitter covariance matrix (49)?

L 702:

Am I interpreting these figures correctly in that the new proposed approaches are all mostly making the RMSE larger relative to the $S(IR)^x R$ (in the small-ensemble size cases of interest)?

L 740:

The smoothing appears to have significant benefits. Are there any strategies for how this could be applied if an exhaustive optimization of the parameters is not possible (e.g. for a large system)?

L 742:

Do you have the baseline RMSE values for the EnKF?

L 743:

"From these results, we conclude that the smoothing by weights step [of Penny and Miyoshi (2016)] is an efficient way of [reducing] the artificial discontinuities [that were] introduced when concatenating the locally updated particles, especially when combined with the coloured noise regularisation jitter method."

I should note that the $S(IR)^x R\_a$ method appears closest to that presented by Penny and Miyoshi (2016), since their inflation is adaptive and using the terminology here is a regularization jitter scaled by the ensemble anomalies.

L 771:

The results look very nice with the OT approach. Do you have an analysis of the relative costs of each of the methods as a function of system size, observation count, and ensemble size?

L 794:

"local LET algorithm"

Is that redundant? Perhaps just say "LET algorithm"

L 804:

I think it would be appropriate at this point to provide a companion plot that shows the relative cost for each method as well.

L 809:

Perhaps you should put the EnKF baseline on the plot as well.

L 816:

"dynamic[s]"

L 828:

"The ETKF requires at least Ne = 12 ensemble members to avoid divergence."

This would imply that the number of positive and neutral Lyapunov exponents of the system is 11.

L 835:

It may not hurt to repeat the definition of each algorithm here.

L 923:

"The SO formalism is elegant."

This seems a strange characterization given that the next few sentences describe legitimate problems with the approach.

L 940:

I suggest either staying consistent with the rest of the paper and defining the section using the algorithm name adopted in the paper - LPF^y - S(IRP_P )^yR, or renaming the rest of the algorithms in the paper based on the authors that introduced them.

L 968:

The terms 'ensemble member' and 'particle' are synonymous - they differentiate the same concept developed in two different fields. The term ensemble does not imply a 2 moment method, so the naming convention shouldn't be used for the purpose stated here.

L 970:

"one first need[s] to"

L 971:

This computation is expensive for large systems. Is this computed in ensemble space or model space?

L 1009:

"any distance that need[s] to be computed relative[] to the observation site…"

Table 2:
The nomenclature table is somewhat helpful, but I'd prefer a full table showing each method for which results are presented, with a description of the method, and the

section where it can be found in the text.

L 1013:

If the block computing is required to make the algorithms computationally scalable to large systems, then these are the results that should be reported.

1073:

"size Ne grows[,] the RMSE decreases"

1075:

Again, I suggest showing the LETKF baseline RMSEs, as well as the computational costs of each method.

1019:

"few but [dis]similar LPF algorithms"

Figure 21:

The better of the white noise and colored noise jitter should be used for each method.

I have to state again that there should be another case presented in which the LETKF fails and the S()R methods produce superior results.

I very much like the promise of the LPF^x OT methods. However, I'd like to see the S(IR)^xSR_a method of Penny and Miyoshi (2016) presented, which should give a nice balance between parallelizable computational costs and accuracy as measured by RMSE - which was the primary goal of the algorithm.

---

## Author Comment (AC1) · 24 Jul 2018

We thank Reviewer 1 for insighful comments and suggestions.

**1 Specific comments**

**1.1 Comment 1**

*Do I understand correctly that your observations in the L96 setup are of every variable at every time step?*

Yes, with the L96 model every variable is observed at every time step.

*So your discussion on pre-regularisation in some way is a discussion on whether to use model error or not? Perhaps this could be made more explicit in the main text.*

Yes, pre-regularisation is in some way similar to the use of model error. This is already explicit in the text ("using a stochastic extension of the model").

*If observations are of every variable at every time step then it will strongly constrain the posterior pdfs to be Gaussian and may well be influencing the results seen. This is alluded to in the main text but I think it would add insight to explicitly state this in the discussion in section 5.1 and that it is also addressed through the use of the barotropic vorticity model (which I understand uses a more sparse observational system in space and time).*

Yes, the posterior pdfs are close to Gaussian pdfs, this is already stated in the last paragraph of Sect. 5.1. In Sects. 6.2 and 9.3, we have followed your suggestion and added a remark.

**1.2 Comment 2**

*The paper outlines quite a few different extensions to existing localisation methods. This is stated in the abstract and introduction and when I read the different algorithm descriptions in detail I could find the paragraph were the differences were noted. However, in general I wasn't really left with a strong feeling of where you had introduced new elements and what benefit they had brought i.e. Section 4.4.4 is entirely new work that has the best result for L96 for the state domain localisation but this is only re-marked on in the final paragraph of the algorithm description section. If it was possible to include a small summary that highlights the new work and the improvement it brings, either within an already existing concluding section or as something separate, then I think it is a chance to bring your work to the fore. It would also explicitly demonstrate how setting individual schemes in to a general context can bring benefits. This could be split between the state domain localisation and sequential-observation localisation if that was the more natural division.*

The outline of the paper has been slightly modified. We have included, at the end of each theoretical section, a sub-section called "Summary for the LPF* algorithms". This sub-section includes highlights where we clarified what already existed, what is new and what was improved. It includes as well as some discussion about the numerical complexity and the asymptotic limit of the algorithms (which was demanded by the other Reviewer and the associate Editor). Thank you for the suggestion, this is a great addition.

**2 Technical corrections**

**2.1 L97**

*A capital N has slipped in to representatioN*

Done.

**2.2 L386**

*It should be 'in order to preserve part of the spatial structure held in the prior particles'*

Done.

**2.3 L404**

*So $\mathbf{E}^r$ has discontinuities?*

Yes, $\mathbf{E}^r$ has discontinuities, which is why we had to improve the resampling step. Remember that $\mathbf{E}^r$ is the matrix implicitly defined by step 5 of Alg. 1.

**2.4 L543**

*The sentence doesn't really make grammatical sense.*

The sentence has been corrected.

**2.5 L625**

*I assume this is 's' but it would be good to explicitly state it.*

Done. Thank you for the suggestion.

---

## Author Comment (AC2) · 24 Jul 2018

We thank Dr. Stephen G. Penny (Reviewer 2) for his insighful comments and suggestions.

In this article, we describe and compare many different localised PF methods. Contrary to many articles in the PF literature, the algorithmic sections are detailed and many explanations are given about potential numerical choices. The numerical illustrations use several models, not only in one dimension, and an exhaustive exploration of the algorithm parameters is performed. Please keep in mind that this goes well beyond most studies on the subject.

**1 General comments**

**1.1 Comment 1)**

*The relative costs between the methods should be calculated and compared, along with RMSE, in each section. Some discussion of costs is made in passing, but no quantitative analyses are offered until the end where a few methods are compared. I suggest extending this to each of the major direct comparisons at the end of the LPFx and LPFy sections.*

The numerical complexity of each method is now discussed in Sects. 4.5.2 and 7.5.2. For the BV model with high resolution, the computation times are reported in Table 5. For the low-order models however (standard L96, BV with coarse resolution), we did not to add the "companion plots" suggested by the reviewer for several reasons:

1. The computation time highly depends on many other factors than numerical complexity: implementation, programming language, processors architecture... With low-order models, these factors may be very important and therefore the computation time is often irrelevant.

2. In our configurations, the parameters are not reprensentative of realistic applications: the number of grid points is very small, the number of particles is very high, and the number of spatial dimensions is limited (to 2 in our case).

3. This would add approximately a dozen figures to an article which is already long.

In the PF literature, the only article in which we have seen some discussion about complexity and computation time is the one by Penny and Miyoshi (2016). Finally, we want to highlight the fact that conclusions regarding the computational cost of a method cannot be based on test series with low-order models. From our experience,

the ranking of an algorithm in computational cost looks very different when using the one-dimensional L96 model or the two-dimensional BV model with high resolution.

**1.2 Comment 2)**

*It would be useful to show the results of the EnKF baseline, both in RMSE and computation time.*

The EnKF scores were given in the text for the L96 model and shown in Fig. 17 for the BV model. We have added a new figure (Fig. 4) that shows the score of the EnKF with the L96 model. We have also added horizontal baselines in most LPF figures (Figs. 8, 13, 14, 15, 16, 18, 20 and 21).

*Since the local PF variants are not outperforming the EnKF baseline, the authors should consider some special case scenarios in which the PF does outperform the EnKF as a motivation for continued development of the local PFs and to show why local PFs may also have advantages over standard EnKFs.*

This is not true, several LPFs do outperfrom the EnKF in the standard configuration for the L96 model (see Fig. 20), which is a première to our knowledge. Besides, the point of this article is not to design LPF algorithms that beat the EnKF in every configuration, but rather to improve the design of LPFs. Contrary to many studies on the PF, we have chosen to use the dynamical models in standard configurations, which allows for a fair comparison with the EnKF.

Following your recommandation, we have added a test series in Appendix C, in a configuration built to make the EnKF fail. In this configuration, we use the same strongly nonlinear observation operator as Poterjoy (2016). However, the interpretation of these results is harder, because some legitimate question can be asked:

1. Is this configuration relevant for realistic models?

2. How good are the score of the LPFs? There is no baseline for comparison (the EnKF does not count since we are outside the scope of its assumptions).

**1.3 Comment 3)**

*In general, I find the algorithm names confusing.*

The coding system for the LPFs looks complex. But please keep in mind that this system was designed to distinguish more than 20 different algorithms. Other studies that focus on a limited number of methods could use a much simpler coding system.

In fact, the coding system follows one simple principle: capital letters refer to the main algorithmic steps and subscripts are used to differentiate the methods. This is now explained in the caption of Tables 1 and 3.

*The first half of the paper uses a complex coding system, while the second half credits the authors who developed the methods.*

We find the criticism unfair, because every time we introduce a new method, we cite the authors that inspired our work if any. Both $LPF^x$ and $LPF^y$ algorithms follow the same convention, with different subscripts to refer to different methods. For the $LPF^y$s, using 'P' and 'RK' as subscripts is a way to distinguish the two different propagation methods and not to credit the author who developed the method. We could have used 'h' and '2' (for 'hybrid' and '2nd order') but obviously this is harder to remember. It serves exactly the same purpose as the subscrits 'e' and 's' (distinction between optimal transport in ensemble space and in state space).

*A more consistent and simpler naming convention would be nice from a reader's point*

*of view, and should be used throughout.*

Following your suggestion, the naming convention has been modified. The new system is consistent and as simple as it can be given the fact that it is used for more than 20 algorithms. It is explained in details in the caption of Tables 1 and 3.

*In addition, a single table describing every algorithm name, what is does, and what section it can be found in, would also help to add clarity.*

Description tables for the $LPF^x$ and $LPF^y$ algorithms have been added (Tables 2 and 4). Thank you for the suggestion; this is a great addition.

*From what I can tell, the $S(IR)\hat{}xSR\_a$ method appears closest to that of Penny and Miyoshi (2016), as it uses smoothing of weights and adaptive "regularisation jitter" based on the ensemble perturbations, and I think this should be given proper credit, as one of the few LPF methods offered in the geophysical literature that has a combination of good performance and low computational cost.*

We did cite Penny and Miyoshi (2016) when presenting the smoothing by weights step in Sect. 4.4.1. As shown by our new results with the high-dimensional BV model, the $S(IR)^xS_{PM}R_c$ (new nomenclature), our generalisation of the LPF of Penny and Miyoshi (2016), does not have a favourable ratio accuracy / computation time.

**2 Specific comments**

**2.1 L28**

*Also cite Kalnay and Yang regarding the "Running-in-Place" method.*

We have added a reference to the RIP as an important precursor method. Note how-
ever that, contrary to the MLEF, the 4D-ETKF and the IEnKS, the RIP is not mathemat-
ically consistent.

**2.2 L39**

*comma after hybridisation*

Done.

**2.3 L43**

*[fewer] particles*

Done.

**2.4 L60**

*degree[s] of freedom*

Done.

**2.5 L63**

*geophysical system[s]*

Done.

**2.6  L85**

*I'm assuming y_k:0 is the set of all observations from time t=0 to time t=k, but perhaps you can state that explicitly.*

Yes, this was stated on the same line.

**2.7  L91**

*Of course, there are many different goals in data assimilation. This is a typical goal. My immediate reaction here is that the DA filtering problem consists in estimating pi_k+1|k. This is the goal at least, usually, to make a prediction. Perhaps you can say - "The DA filtering problem consists in estimating pi_k|k and pi_k+1|k with given realizations of y_k:0."*

The sentence has been changed to reflect the fact that estimating $\pi_{k|k}$ may not be the only goal of DA.

**2.8  L97**

*particle representatio[n]*

Done.

**2.9  L111**

*What do you mean by "pure ensemble transformations"?*

"Pure ensemble transformation" means that this is a transformation that act on the ensemble and that should ideally not alter the density. This is explained by the following

sentence. However, we agree that the term "pure" can be confusing and we removed it.

**2.10   L170**

*I'm not sure I understand why this is remarkable. Could you elaborate?*

This is remarkable that in $w_{k+1}^i$, the dependance on $\mathbf{x}_{k+1}^i$ vanishes. We have added a reference to Doucet (2000) for clarification.

**2.11   L177**

*"... to more elaborate algorithms ..."*
Done.

**2.12   L182**

*"... models [have led] to weight degeneracy..."*
Done.

**2.13   L196**

*" ... it [might seem] surprising that, although MC method[s] have..."*
Done.

[Figure]

**2.14 L214/217**

*You switch tenses, first referring to Synder et al. (2008) as a set of authors, and then referring to Snyder et al. (2008) as a paper. Because "et al." means "and others", I prefer the former and recommend changing L217 to: "Snyder et al. (2008) [do] not illustrate..."*

Done.

**2.15 L223**

*"...optimal importance proposal [density]..."*

Done.

**2.16 L227**

*"... does not [primarily] come from ..."*

Done.

**2.17 L238**

*"... [elaborate] models ..."*

Done.

**2.18 L252**

*It seems awkward to begin a sentence with a variable name. Perhaps used instead: "The quantity tao^ 2 would then be defined using"*

The beginning of the sentence has been changed. Thank you for the suggestion.

**2.19 L266**

*While I appreciate the implication of calling this a 'discontinuity', there are some complications in defining a concept of continuity on a discrete model grid. Some discussion should be made regarding this point.*

Indeed, discontinuity here does not refer to the mathematical notion of continuity. Following your suggestion, we have added some explanation.

**2.20 L281**

*Perhaps you could list some of the past examples of this type via citation.*

Done.

**2.21 L285**

*Again, I suggest citing a few example of this type as well.*

Done.

**2.22 L332**

*Not within a circle, but within some general local region. A circle is a common choice.*

We changed the subsection. Thank you for helping us clarify this point.

**2.23 L367**

*"and decrease[s] exponentially"*

Done.

**2.24 L371**

*The "size" of the blocks using what measure? Number of grid points?*

We clarified this point. Thank you for spotting the imprecision.

**2.25 L386**

*I think "hold" should be "held"*

Done.

**2.26 L390-415**

*I'm not sure if the point was adequately made that neighboring weights can be made arbitrarily smooth by letting the radius of the taper function ($r_s$) get large. I.e. as $r_s$ goes to infinity, the global PF solution is recovered.*
The asymptotic limit of a LPF$^{x}$ algorithm using smoothing by weights is now discussed in Sect. 4.5.4. When $r \rightarrow \infty$, $\mathbf{E}^{r}$ is not necessarily equivalent to the global PF solution (because the resampling is independant at each grid point). When $r \rightarrow \infty$ and $r_{s} \rightarrow \infty$, $\mathbf{E}^{s}$ is not necessarily equivalent to the global PF solution (again because the resampling is performed independantly at each grid point).

*In that sense, I'm not sure why the additional alpha smoothing step is made explicit.*

We do not understand your concern about making the "alpha smoothing step" (we guess you mean Eq. (32)) explicit.

**2.27 L543**

*"only [a] big ensemble"*

The sentence has been changed.

**2.28 L561**

*"RMSE offers a" to "RMSE offer a"*

Done.

**2.29 L562**

*I'm not sure it is settled that the RMSE of the mean is an adequate measure of the PF performance, given that the distribution may not have the mean and mode equal.*

We do not entirely understand the point of this remark. One must keep in mind that PFs are suited to compute the mean state and not the mode. Besides, in these weakly

non-Gaussian configurations, mean and mode should not be far from each other.

*Further, if we are to adopt a PF solution over an EnKF, then we are acknowledging that the primary data assimilation goal is specifically not mean state estimation, but rather estimation of the state distribution.*

We do not agree with this remark. Using a PF does not mean that we are not interested in the mean state. And, again, one must keep in mind that PFs are not suited to "estimat[e] the state distribution". Indeed, with a PF, we primarily have an estimation of

$$\int p(\mathbf{x}) f(\mathbf{x}) \, \mathrm{d}\mathbf{x}, \tag{1}$$

for any test function $f$, but no estimation of $p(\mathbf{x})$.

**2.30  L572**

*"yield[s]"*

The sentence has been removed.

**2.31  L595**

*I don't understand what this first sentence means. What does it mean to have more information than the truth?*

Indeed there was a typo: one should have read "on average more informations than the observations". Sorry about that; it has been corrected.

**2.32 P25-25**

*It would be nice if Figures 4 and 5 were closer to the referencing text. Perhaps you can make that request of the editors.*

We will ask for this in the editing process.

**2.33 P25**

*It appears here that you are using a fixed parameter for the 'regularization jitter'. Have you compared this the LPF of Penny and Miyoshi (2016) that set this value adaptively based on the analysis ensemble spread?*

The discussion on "adaptative" resampling is located in the "coloured noise" section (5.8). In this section, we developed a method that is an extension of the method by Penny and Miyoshi (2016). We compared our method to that of Penny and Miyoshi (2016) (not shown in this article) and always found better accuracy with our extension. We have added a few sentences in section 5.8.2 about this.

**2.34 L651**

*I'm confused how the higher RMSEs of the $S(IR\_SU)\hat{} \; xR$ algorithm indicates an efficient approach. Could the authors elaborate.*

The $S(IR_{su})\times R$ algorithm is the only one that does not use the "adjustment-minimising" property. If it has a higher RMSE, we believe that it means that the "adjustment-minimising" property is efficient. Following your comment, we reformulated the statement.

*What is the RMSE ratio used in Figure 9? Why does the figure caption say "RMSE" while the y-axis says "RMSE ratio"?*

The RMSE ratio used in Fig. 10 (new numbering) is detailed in the second sentence of the caption "The scores are displayed in units of the RMSE of ...". The same kind of ratio is used in Figs. 9, 11 and 12.

**2.35 L675**

*I need a reminder at this point - E is the set of ensemble members and X is the set of perturbations around the ensemble mean? Are the xˆ i in (46) the columns of E?*

$\mathbf{E}$ is the ensemble matrix (defined in Sect. 2), whose columns are the particles $\mathbf{x}^i$, which is a very common notation in ensemble DA. $\mathbf{X}$ is indeed the set of (normalised) perturbations around the ensemble mean, as defined two lines above. Reminders have been added, thank you for the suggestion.

**2.36 L697**

*change "as following" to "as follows"*

Done.

*How is equation (48) different than (46)? Could you instead just say it is defined as in (46) with a new formulation for the Gaussian regularization jitter covariance matrix (49)?*

Equation (51) is different from Eq. (49) (new numbering) because it uses the local weights $w_n^i$ instead of the global weights $w^i$ that do not exist in LPFs. We explained

in the text why Eqs. (47) to (49) cannot be used (see the second subsection of Sect. 5.8.2)

**2.37   L702**

*Am I interpreting these figures correctly in that the new proposed approaches are all mostly making the RMSE larger relative to the S(IR)ˆ xR (in the small-ensemble size cases of interest)?*

Your interpretation is correct. This is discussed in the last subsection of Sect. 5.8.

**2.38   L740**

*The smoothing appears to have significant benefits. Are there any strategies for how this could be applied if an exhaustive optimization of the parameters is not possible (e.g. for a large system)?*

As shown by Fig. 12, $\alpha_s = 1$ is optimal in this configuration for the L96 model. We have checked that this is the case in most situations where we used the smoothing (in particular with the BV model). However, we could not find an obvious relationship between the optimally tuned values of $r$ and $r_s$. Besides, one should keep in mind that, in the "small-ensemble size cases of interest", the benefits of the smoothing are far less impressive than the benefits of OT (this can be seen in Fig. 16).

**2.39   L742**

*Do you have the baseline RMSE values for the EnKF?*

See the new Fig. 4. We have also added a baseline to Fig. 13.

**2.40   L743**

*"From these results, we conclude that the smoothing by weights step [of Penny and Miyoshi (2016)] is an efficient way of [reducing] the artificial discontinuities [that were] introduced when concatenating the locally updated particles, especially when combined with the coloured noise regularisation jitter method." I should note that the $S(IR)xR\_a$ method appears closest to that presented by Penny and Miyoshi (2016), since their inflation is adaptive and using the terminology here is a regularization jitter scaled by the ensemble anomalies.*

Please keep in mind that the work of Penny and Miyoshi (2016) has been cited in Sect. 4.4.1 (where we introduced the method in the first place) and that the $S(IR)^{\times}S_{PM}R_{c}$ algorithm tested in this section is not the LPF of Penny and Miyoshi (2016) but an improvement thereof, which includes: a more general framework that can be applied to different types of resampling, a tapering function, a smoothing radius and a smoothing strength parameters, coloured noise regularisation.

The other corrections ("reducing" and "that were") have been done.

**2.41   L771**

*The results look very nice with the OT approach. Do you have an analysis of the relative costs of each of the methods as a function of system size, observation count, and ensemble size?*

This is detailed in the new section 4.5.2.

**2.42   L794**

*"local LET algorithm" Is that redundant? Perhaps just say "LET algorithm"*

"LET" means linear ensemble transform (introduced in Sect. 2.3 with appropriate citations).

**2.43  L804**

*I think it would be appropriate at this point to provide a companion plot that shows the relative cost for each method as well.*

Please, see the discussion about general comment 1).

**2.44  L809**

*Perhaps you should put the EnKF baseline on the plot as well.*

See the new Fig. 4. We have also added a baseline to Fig. 16.

**2.45  L816**

*"dynamic[s]"*

Done.

**2.46  L828**

*"The ETKF requires at least $Ne = 12$ ensemble members to avoid divergence." This would imply that the number of positive and neutral Lyapunov exponents of the system is 11.*

You are right.

[Figure]

**2.47 L835**

*It may not hurt to repeat the definition of each algorithm here.*

We now refer to the algorithms' list in the (new) Table 2. Thank you for the suggestion.

**2.48 L923**

*"The SO formalism is elegant." This seems a strange characterization given that the next few sentences describe legitimate problems with the approach.*

We wanted to emphasis that the formalism developed in Sects. 7.2.1 and 7.2.2 looks elegant. The next few sentences raise issues that appear when combining the SO formalism with the PF. These issues are not specific to the SO formalism. Following your commment, we mitigated this remark.

**2.49 L940**

*I suggest either staying consistent with the rest of the paper and defining the section using the algorithm name adopted in the paper - LPFˆy - S(IRP_P )ˆyR, or renaming the rest of the algorithms in the paper based on the authors that introduced them.*

The nomenclature for the algorithms has been changed to follow your suggestions (please, see the discussion about general comment 3)). Thank you for spotting the inconsistency in the naming of the subsections. We corrected this point.

**2.50   L968**

*The terms 'ensemble member' and 'particle' are synonymous - they differentiate the same concept developed in two different fields. The term ensemble does not imply a 2 moment method, so the naming convention shouldn't be used for the purpose stated here.*

We now refer to this algorithm as "the second order propagation algorithm".

You are right, the terms "ensemble member" and "particle" designate the same concept but they are used in different contexts. The term "particle" is often used with Bayesian method (or at least with methods based on importance sampling) while the term "ensemble member" is often used with Kalman filters. The naming convention is therefore already commonly being used to distinguish between Bayesian / non-Bayesian methods.

**2.51   L970**

*"one first need[s] to"*

Done.

**2.52   L971**

*This computation is expensive for large systems. Is this computed in ensemble space or model space?*

We stated in Sect. 7.4.3 that only submatrices of $\Sigma$ need to be computed. Therefore this computation is not that expensive for large systems.

How to implement Eq. (65) is beyond the scope of this article. For our numerical experiments, we computed it in state space. Note that because of the schur product, there is no obvious formula for $\Sigma$ in ensemble space.

**2.53 L1009**

*"any distance that need[s] to be computed relative[] to the observation site..."*

Done.

**2.54 Table 2**

*The nomenclature table is somewhat helpful, but I'd prefer a full table showing each method for which results are presented, with a description of the method, and the section where it can be found in the text.*

The caption of this table has now more details, such that it is more explicit. Following your recommendations, we have added new tables (Table 2 and 4). Thanks, this is a great addition.

**2.55 L1013**

*If the block computing is required to make the algorithms computationally scalable to large systems, then these are the results that should be reported.*

This is only a discussion about algorithmic possibilities. The block computation is a way to reduce the computation requirements of LPF$^y$s. It should not be required to make the algorithms computationally scalable to large systems.

**2.56   L1073**

*"size Ne grows[,] the RMSE decreases"*

Done.

**2.57   L1075**

*Again, I suggest showing the LETKF baseline RMSEs, as well as the computational costs of each method.*

The RMSE values for the LETKF are in the dedicated figures. We have also added a baseline to Figs. 20 and 21. For the computational cost, please see the discussion about general comment 1).

**2.58   L1019**

*"few but [dis]similar LPF algorithms"*

Done.

**2.59   Figure 21**

*The better of the white noise and colored noise jitter should be used for each method.*

For very small ensemble sizes, the white noise jitter yields lower RMSEs in most test series so far. This is why we used it in this high-dimensional test series.

*I have to state again that there should be another case presented in which the LETKF fails and the S()R methods produce superior results.*

Please see the discussion about general comment 2).

*I very much like the promise of the LPFˆ x OT methods. However, I'd like to see the S(IR)ˆ xSR_a method of Penny and Miyoshi (2016) presented, which should give a nice balance between parallelizable computational costs and accuracy as measured by RMSE - which was the primary goal of the algorithm.*

We originally did not select the $S(IR)^{\times}S_{PM}R_c$ for this high-dimensional test series for these reasons:

1. Given the results for the BV model in the coarse resolution configuration, with very small ensemble sizes this algorithm is outperformed by the algorithms using OT resampling.

2. This algorithm is slower than the other algorithm, because in two dimensions, computing the smoothing by weights is numerically expensive.

3. Optimal tuning parameters for this method are harder to find (both because there are more parameters and because the simulations are long).

Following your recommendation, we performed the simulation and reported the results in Fig. 22 and Table 5. The ration accuracy / computation time is not in favor of this method.

Finally, please keep in mind that the $S(IR)^{\times}S_{PM}R_c$ algorithm tested here is not the LPF of Penny and Miyoshi (2016) but our improvement thereof, which includes: a more general framework that can be applied to different types of resampling, a tapering function, a smoothing radius and a smoothing strength parameters, coloured noise regularisation.

---

## Editor Decision (ED1)

Bonjour Alban,

I have now received two reviews of the revised version of your paper.

The first review is by the same referee #1 of the previous version. The referee considers your paper can be accepted as it is.

The other review is by a new referee, identified as Referee #3. He/she considers your paper can be accepted subject to minor revisions, and makes a few suggestions. The first two of these have to do with the general conclusions of the paper rather than with specific points.

I have myself read the paper in some detail (mostly for my own instruction), and I make below a few suggestions for edition.

Please correct the paper along referee #3's suggestion, as well as along my own ones. Should you disagree with a particular comment, or decide not to follow a particular suggestion, please state precisely your reasons for that.

I thank you again for having submitted your paper to the NPG Special Issue in tribute to Anna Trevisan, and look forward to receiving a new version.

Olivier Talagrand
Editor, *Nonlinear Processes in Geophysics*

My Editor's suggestions

Line numbers refer to the version of the paper with explicit corrections (file npg-2018-15-author_response-version1.pdf)

1. You repeatedly mention the multinomial resampling and the stochastic universal (SU) sampling algorithms. But, except for a few comments (ll. 104-109), you do not apparently say much about them. It might be useful to say a little more.

2. Eqs (A17) and (A25), rhs's. I understand $x$ should be replaced by $q$ (vorticity)

3. Ll. 2687 and 2729, $(\delta x/L)^2 \rightarrow (L/\delta x)^2$

4. Algorithm 6, p. 28. Point 5 is unclear. Could it be possible to explain more clearly how $x_v^i$ can be updated using Eq. (57). Maybe by introducing an intermediary equation ? And must not $y_q$ be used also at this stage ?

5. You mention repeatedy, and denote $G$, the Gaspari-Cohn function. Is it always the same one (you add the attribute *piecewise rational* in l. 1363).

6. L. 1918. Reference to Fig. 18, rather than 17, seems more appropriate.

7. L. 2254, *this algorithm*. Ambiguous. Which algorithm ?

---

## Author Response (AR2)

**Response to the Editor**

A. Farchi and M. Bocquet

September 25, 2018

We thank Olivier Talagrand (the Editor) for insighful comments and suggestions.

**Comment 1**  *You repeatedly mention the multinomial resampling and the stochastic universal (SU) sampling algorithms. But, except for a few comments (ll. 104- 109), you do not apparently say much about them. It might be useful to say a little more.*

Section 2.3 of our manuscript is dedicated to the resampling step. It is true that the content of this section is rather generic and not specific to the multinomial resampling and SU sampling algorithms. This is why we added, in a new appendix E, a description of the multinomial resampling and the SU sampling algorithms. This appendix is rather short and explicitly refers to the review by van Leeuwen (2009) in which there is a detailed description of several resampling algorithms.

**Comment 2**  *Eqs (A17) and (A25), rhs's. I understand x should be replaced by q (vorticity)*

Indeed, in the right hand side of Eqs. (A17) and (A25), $x$ is the vorticity, which means that it should be replaced by $q$. On the other hand, $q$ is already being used in these equations for the observation site indices. Therefore, in order to avoid any possible confusion, we switch to $\zeta$ for the vorticity. Thank you for spotting this mistake.

**Comment 3**  *Ll. 2687 and 2729, $(\delta x/L)^2 \to (L/\delta x)^2$ .*

This mistake has been corrected.

**Comment 4**  *Algorithm 6, p. 28. Point 5 is unclear. Could it be possible to explain more clearly how $x^i_V$ can be updated using Eq. (57). Maybe by introducing an intermediary equation? And must not $y_q$ be used also at this stage?*

The confusion here comes from the fact the generic Algorithm 6 is built upon Eq. (62) (previouslsy Eq. (59)) and not upon the intermediate Eq. (60) (previously Eq. (57)). With Eq. (62), the three-step update loop for each particle in Algorithm 6 is obvious. Therefore, we removed the reference to Eq. (60) in Algorithm 6.

**Comment 5**  *You mention repeatedy, and denote G, the Gaspari–Cohn function. Is it always the same one (you add the attribute piecewise rational in l. 1363).*

We use the same function through the entire manuscript. It is now always referred to as *the Gaspari–Cohn* function, a notation that we introduce in Sect. 4.2.2. Thank you for helping us clarify this point.

**Comment 6**  *L. 1918. Reference to Fig. 18, rather than 17, seems more appropriate.*

Indeed, at this point it seems more appropriate to refer to Fig. 18. This has been corrected, thank you for the suggestion.

**Comment 7**  *L. 2254, this algorithm. Ambiguous. Which algorithm?*

The ambiguity has been clarified.

**Response to reviewer 3**

A. Farchi and M. Bocquet

September 25, 2018

We thank Reviewer 3 for insighful comments and suggestions.

**Comment 1** *Could you summarize remained problems and challenges of the LPF? Such description is helpful for ones who will start working on the LPF succeeding in this manuscript.*

We have tried to summarise the remaining challenges in a few additional paragraphes in the conclusion. This list, however, reflects our personal opinion and is not meant to be exhaustive. Indeed, introducing localisation in the particle filter is a relatively young topic and anything can happen.

**Comment 2** *Could you discuss the conclusions of this manuscript can be directly applicable to large dimensional problems such as atmospheric models?*

Our successful application with the barotropic vorticity model shows that the LPF algorithms have the potential to work with high-dimensional systems. However, practical difficulties could arise when actually trying to use the LPF algorithms with realistic models. We have tried to summarise these difficulties in the conclusion (new dedicated paragraph). Once again, this list reflects our personal opinion and is not meant to be exhaustive.

**Comment 3** *The optimal localization scale of the LPF is smaller than the LETKF (Section 5.3). Is this the main reason why the LPF did not outperform the LETKF (section 5.12)? Please discuss this point.*

We added in Sect. 5.12 a paragraph in which we discuss potential reasons why the LPF$^x$s do not outperform the LETKF in this configuration. Note that smaller values of the localisation radius is one of the main reasons, although there are other factors.

[revised manuscript text omitted]
^{\rm r}$ is an $N_{\rm x} \times N_{\rm e}$ matrix different from the $N_{\rm x}/N_{\rm b} \times N_{\rm e}$ matrix $\mathbf{E}_{|b}^{\rm r}$ defined in Sect. 4.2.3. We then define the smoothed ensemble matrix $\mathbf{E}^{\rm s}$ by

$$[\mathbf{E}^{\rm s}]_n^i = \frac{\sum_{b=1}^{N_{\rm b}} G\left(\frac{d_{n,b}}{r_{\rm s}}\right) [\mathbf{E}_b^{\rm r}]_n^i}{\sum_{b=1}^{N_{\rm b}} G\left(\frac{d_{n,b}}{r_{\rm s}}\right)}, \tag{30}$$

740 where $d_{n,b}$ is the distance between the $n$-th grid point and the center of the $b$-th block, $r_{\rm s}$ is the *smoothing radius*, a free parameter potentially different from $r$ and $G$ is a taper function, potentially different from the one used to compute the local weights.

If the resampling is performed using a "select and dupli­cate" algorithm (see Sect. 2.3), for example the SU sampling algorithm, then define $\phi_b$ as the resampling map for the $b$-th block, i.e. the map computed with the local weights $w_b^i$ such that $\phi_b(i)$ is the index of the $i$-th selected particle. $\mathbf{E}$ being the prior ensemble matrix, Eq. (30) becomes

$$[\mathbf{E}^{\mathrm{s}}]_n^i = \frac{\sum_{b=1}^{N_{\mathrm{b}}} G\left(\frac{d_{n,b}}{r_{\mathrm{s}}}\right) [\mathbf{E}]_n^{\phi_b(i)}}{\sum_{b=1}^{N_{\mathrm{b}}} G\left(\frac{d_{n,b}}{r_{\mathrm{s}}}\right)}. \tag{31}$$

Finally, the ensemble is updated as

$$\mathbf{E} \leftarrow \alpha_{\mathrm{s}}\mathbf{E}^{\mathrm{s}} + (1 - \alpha_{\mathrm{s}})\mathbf{E}^{\mathrm{r}}, \tag{32}$$

where $\mathbf{E}^{\mathrm{r}}$ is the resampled ensemble matrix implicitly de­fined by step 5 of Algorithm 1 and $\alpha_{\mathrm{s}}$ is the *smoothing strength*, a free parameter in $[0, 1]$ that controls the intensity of the smoothing. When $\alpha_{\mathrm{s}} = 0$, no smoothing is performed and when $\alpha_{\mathrm{s}} = 1$, the analysis ensem­ble is totally replaced by the smoothed ensemble.

Algorithm 2 describes the analysis step for a generic LPF$^{\mathrm{x}}$ with smoothing by weights. The original LPF of Penny and Miyoshi (2016) can be recovered if:

– blocks have size 1 grid point (hence there is no distinc­tion between grid points and blocks);

– the local weights are computed using Eq. (29);

– $G$ is a top hat function;

– the resampling method is the SU sampling algorithm;

– $r_{\mathrm{s}}$ is set to be equal to $r$;

– $\alpha_{\mathrm{s}}$ is set to 0.5.

The method described here is a generalisation of their algo­rithm.

Note that when the resampling method is the SU sampling algorithm, the matrices $\mathbf{E}_b^{\mathrm{r}}$ do not need to be explicitly com­puted. One just has to store in memory the resampling maps $\phi_b$, $b = 1 \ldots N_{\mathrm{b}}$ and then use Eq. (31) to obtain the smoothed ensemble matrix $\mathbf{E}^{\mathrm{s}}$.

The smoothing by weights step is an ad-hoc fix to reduce potential unphysical discontinuities after they have been in­troduced in the local resampling step. Its necessity hints that there is room for improvement in the design of the local re­sampling algorithms.

**4.4.2   Refining the sampling algorithms**

In this section, we study several properties of the local re­sampling algorithm that might help dealing with the discon­tinuity issue: balance, adjustment and random numbers.
* * *

[revised manuscript text omitted]

$$= \frac{p(y_q | \boldsymbol{x}_L) p(\boldsymbol{x}_G, \boldsymbol{x}_L)}{p(\boldsymbol{x}_G, y_q)}. \quad (56)$$

This yields an assimilation method for $y_q$ described by Algorithm 5.
* * *
**Algorithm 5** Single analysis step for a generic LPF$^y$ algorithm using the $LG$ partition
* * *
**Require:** Prior ensemble $\boldsymbol{x}^i$, $i = 1 \ldots N_e$ and observation $y_q$
1: Build the $LG$ partition as described in Sect. 7.1
2: **for** $i = 1$ to $N_e$ **do**
3:     Do not update $\boldsymbol{x}_G^i$
4:     Update $\boldsymbol{x}_L^i$ conditionally to $y_q$ and $\boldsymbol{x}_G^i$ as stated by Eq. (56)
5: **end for**
6: **return** Updated ensemble $\boldsymbol{x}^i$, $i = 1 \ldots N_e$
* * *
**7.2.2   With the $UVW$ partition**

With the $UVW$ partition, the conditional density is factored as

$$p(\boldsymbol{x}|y_q) = p(\boldsymbol{x}_U, \boldsymbol{x}_V, \boldsymbol{x}_W | y_q), \quad (57)$$

$$= \frac{p(\boldsymbol{x}_U, \boldsymbol{x}_V, \boldsymbol{x}_W, y_q)}{p(y_q)}, \quad (58)$$

$$= \frac{p(y_q | \boldsymbol{x}) p(\boldsymbol{x}_V | \boldsymbol{x}_U, \boldsymbol{x}_W) p(\boldsymbol{x}_U, \boldsymbol{x}_W)}{p(y_q)}, \quad (59)$$

$$= \frac{p(y_q | \boldsymbol{x}_U) p(\boldsymbol{x}_V | \boldsymbol{x}_U, \boldsymbol{x}_W) p(\boldsymbol{x}_U, \boldsymbol{x}_W)}{p(y_q)}. \quad (60)$$

If one assumes that the $U$ and $W$ regions are not only uncorrelated but also independent, then one can make the additional factorisation:

$$p(\boldsymbol{x}_U, \boldsymbol{x}_W) = p(\boldsymbol{x}_U) p(\boldsymbol{x}_W). \quad (61)$$

Finally, the conditional density is

[revised manuscript text omitted]